# Evaluating Interpretable Methods via Geometric Alignment of Functional Distortions

**Anna Hedström**[1,6,8,†]   *anna.hedstroem@tu-berlin.de*
**Philine Bommer**[1,8]   *philine.bommer@tu-berlin.de*
**Thomas F Burns**[3,4,5]   *t.f.burns@gmail.com*
**Sebastian Lapuschkin**[6]   *sebastian.lapuschkin@hhi.fraunhofer.de*
**Wojciech Samek**[1,2,6]   *wojciech.samek@hhi.fraunhofer.de*
**Marina M.-C. Höhne**[2,7,8,†]   *mhoehne@atb-potsdam.de*

[1] *Department of Electrical Engineering and Computer Science, Technical University of Berlin*
[2] *BIFOLD – Berlin Institute for the Foundations of Learning and Data*
[3] *Institute for Computational and Experimental Research in Mathematics, Brown University*
[4] *Scientific Artificial Intelligence Center, Cornell University, USA*
[5] *Neural Coding and Brain Computing Unit, Okinawa Insitute of Science and Technology*
[6] *Department of Artificial Intelligence, Fraunhofer Heinrich Hertz Institute*
[7] *Department of Computer Science, University of Potsdam*
[8] *UMI Lab, Leibniz Institute of Agricultural Engineering and Bioeconomy e.V. (ATB)*
[†] *corresponding authors*

**Reviewed on OpenReview:** `https://openreview.net/forum?id=5ceyt8qT4e`

## Abstract

Interpretability researchers face a universal question: without access to ground truth labels, how can the faithfulness of an explanation to its model be determined? Despite immense efforts to develop new evaluation methods, current approaches remain in a pre-paradigmatic state: fragmented, difficult to calibrate, and lacking cohesive theoretical grounding. Observing the lack of a unifying theory, we propose a novel evaluative criterion entitled *Generalised Explanation Faithfulness (GEF)* which is centered on explanation-to-model alignment, and integrates existing perturbation-based evaluations to eliminate the need for singular, task-specific evaluations. Complementing this unifying perspective, from a geometric point of view, we reveal a prevalent yet critical oversight in current evaluation practice: the failure to account for the learned geometry, and non-linear mapping present in the model, and explanation spaces. To solve this, we propose a general-purpose, threshold-free faithfulness evaluator `GEF` that incorporates principles from differential geometry, and facilitates evaluation agnostically across tasks, and interpretability approaches. Through extensive cross-domain benchmarks on natural language processing, vision, and tabular tasks, we provide first-of-its-kind insights into the comparative performance of various interpretable methods. This includes local linear approximators, global feature visualisation methods, large language models as post-hoc explainers, and sparse autoencoders. Our contributions are important to the interpretability and AI safety communities, offering a principled, unified approach for evaluation.

 `https://github.com/annahedstroem/GEF`

## 1 Introduction

Explaining the general behaviour, and predictions of machine learning (ML) models, particularly those functioning as black boxes, is critical, especially in domains such as healthcare, finance, and law. Driven by the urgency to comply with regulations like the EU AI Act, and GDPR, the interpretability (or eXplainable

AI (XAI)) research community has produced a plethora of interpretable methods in recent years (Baehrens et al., 2010; Zeiler & Fergus, 2014a; Lundberg & Lee, 2017; Bykov et al., 2022; Fel et al., 2024; Lieberum et al., 2024). Simultaneously, the rise of large-scale, multi-tasking large language models (LLMs) (or "foundation models") (OpenAI, 2023; Mesnard et al., 2024) has spurred a significant shift in the interpretability landscape, with the mechanistic interpretability community producing a new generation of methods specifically designed to decompose, and reverse-engineer these increasingly black-box models (Elhage et al., 2022; Conmy et al., 2023; Bykov et al., 2023; Bills et al., 2023; Templeton et al., 2024). Despite this immense activity, consensus is lacking whether existing methods are of sufficient quality or trustworthy (Adebayo et al., 2018; Ghassemi et al., 2021; Bordt & von Luxburg, 2024; Bhattacharjee & von Luxburg, 2024). Since black-box models lack ground truth explanation labels (Bellido & Fiesler, 1993; Benitez et al., 1997), the universal question: "how faithful is the explanation to the model it seeks to explain?" remains difficult to answer. The prevalence of disagreements within the interpretability community about which methods[1] work, and under what conditions (Neely et al., 2021; Watson et al., 2022; Krishna et al., 2022; Koenen & Wright, 2024) signals that the challenge of evaluation is still unsolved.

To *approximate* explanation quality (Agarwal et al., 2022b; Hedström et al., 2023b), researchers commonly use perturbation-based evaluations, where *robustness* (Montavon et al., 2018; Alvarez-Melis & Jaakkola, 2018b; Yeh et al., 2019; Nguyen & Martinez, 2020; Dasgupta et al., 2022), *sensitivity* (Adebayo et al., 2018; Hedström et al., 2024), and *faithfulness* (Bach et al., 2015; Samek et al., 2017; Ancona et al., 2018; Rieger & Hansen, 2020; Dasgupta et al., 2022; Bhatt et al., 2020; Rong et al., 2022) are well-embraced criteria to examine the relationship between explanation, and model outputs under perturbation, albeit with different emphases. Here, robustness, and sensitivity criteria refer to making small or large perturbations (*e.g.,* adding noise to the input or randomising model parameters), and then measuring corresponding changes in the explanation output. Faithfulness criterion generally measures how much the model's performance degrades when inputs, such as tokens or pixels, are cumulatively perturbed according to the explanation values. Significant changes in model behaviour are interpreted as indicators of explanation faithfulness.

**Lack of Cohesive, Unified Theory.**  Despite repeated attempts to define, and measure faithfulness (Montavon et al., 2018; Jacovi & Goldberg, 2020; Bhatt et al., 2020; Turpin et al., 2023; Lanham et al., 2023; Agarwal et al., 2024), fragmented mathematical terminology makes it an ongoing, and unresolved matter (Bordt & von Luxburg, 2024). What exactly is explanation faithfulness, and how do robustness, and sensitivity evaluations differ from it? From a conceptual standpoint, although these evaluations share common steps—such as perturbing the inputs or the model parameters, measuring the effects, and interpreting the functional outcomes—the overwhelming number of evaluation methods under these distinct criteria (Lakkaraju et al., 2022), and the absence of a cohesive, unified theory makes it difficult to answer such seemingly straightforward questions. To better understand these evaluations' shared attributes, assumptions, and outcomes, a mathematical discussion is required. In Section 2, we propose a *unifying* perspective that formalises robustness, sensitivity, and faithfulness evaluations, providing a principled *Generalised Explanation Faithfulness* (GEF) definition in Section 3 to substitute singular evaluations.

**Ignoring the Impact of Geometry.**  Alongside the lack of a cohesive, unified theory, most perturbation-based evaluations (Section 2.1.2)—while well-intended, and intuitive—often rely on overly simplistic assumptions about the underlying geometry of both model, and explanation spaces. When perturbations are introduced, the functional outcomes of models, and explanations are frequently compared using direct distance measures or correlation coefficients (Alvarez-Melis & Jaakkola, 2018b; Yeh et al., 2019; Ancona et al., 2018; Bhatt et al., 2020; Nguyen & Martinez, 2020; Agarwal et al., 2022a). From a *geometric* perspective, this overlooks a simple yet critical fact: that a uniform perturbation such as input noise or parameter shifts can affect non-linear systems in highly non-uniform ways. Only in a linear system, the perturbation effects would be uniform. By neglecting the geometric differences (*e.g.,* differences in curvatures) between the model, and explanation spaces, current evaluations risk misjudging how faithful the explanation is *w.r.t.* its underlying model. For fair measurements across non-linear systems, perturbation effects must be measured in the context of the distinct geometric structures of the respective manifolds (Lee, 2012). In Section 4, we

---

[1]Throughout this work, we use the terms "interpretable methods" and "explanation methods" interchangeably, without implying a difference in their scope or function.

examine these geometric factors, and introduce a solution that accounts for the intrinsic geometry of each space, thereby improving current evaluation practice.

To address the research gaps in unified theory (Section 3), and the neglected impact of geometry (Section 4), our work offers a fourfold contribution.

**(C1)** In the absence of cohesive theory, we systematise common steps in numerous perturbation-based evaluation algorithms (Section 2), and provide a unifying evaluative criterion for robustness, sensitivity, and faithfulness evaluations (Section 3).

**(C2)** To account for geometric discrepancies in many evaluation methods, we propose a solution based on differential geometry that ensures fair measurements across non-linear mappings (Section 4).

**(C3)** Recognising the need for a general-purpose, threshold-free, task-agnostic faithfulness evaluators, we provide `GEF` and `Fast-GEF` methods, serving distinct compute budgets (Section 5).

**(C4)** Observing the lack of cross-domain insights on the faithfulness across distinct explanation approaches such as local, global, LLM as an explainer, and sparse autoencoders (SAEs), we perform extensive experiments across vision, tabular, and natural language processing (NLP) tasks (Section 6).

Our contributions carry substantial importance to the interpretability (and related) communities. The reliability of individual explanation methods, and XAI as a field is already under hot debate, thus it is not only timely but relevant to provide clarity on the matter of explanation faithfulness. As we enter a new era of interpretability, it is of utmost importance to revisit, and revise existing evaluation approaches. We hope this work will clarify how best to approach and perform faithfulness evaluation, ultimately empowering researchers to confidently select and develop new interpretable methods.

## 2 Interpretability Evaluation: Where Are We Now?

In this section, we present the scope of this work. We begin by outlining preliminaries to estimate explanation quality, followed by a description of the general workflow of perturbation-based evaluation. Finally, we mathematically formalise robustness, sensitivity, and faithfulness evaluation, revealing critical assumptions that are essential for their validity. Complete notation tables are provided in Appendix A.9.

### 2.1 Preliminaries

Let $f_\theta : \mathcal{X} \to \mathcal{Y}$ be a differentiable neural network (NN) that maps inputs $\boldsymbol{x} \in \mathbb{R}^D$ to predictions $\boldsymbol{y} \in \mathbb{R}^C$ of $C$ classes. By functionally mapping $\boldsymbol{x} \in \mathcal{X}$ to $\boldsymbol{y} \in \mathcal{Y}$ with parameters $\theta$ such that $\boldsymbol{y} = f(\boldsymbol{x}; \theta)$, a trained model $f_\theta$ is obtained, which we refer to as $f$. Here, $\theta$ includes weights, and biases, and exists in parameter space $\Theta \in \mathbb{R}^W$ for a fixed architecture in function space $f_\theta \in \mathcal{F}$. The model $f$ may represent NN architectures ranging from simple feedforward MLPs, CNNs to highly parameterised transformer-based models.

**Local Explanations.** To interpret a specific model prediction (*i.e.,* logit) $y := y_c$ of a class $c \in [1, 2, \ldots C]$, we may employ a *local* method. Let $\phi_L : \mathcal{F} \times \mathcal{X} \times \mathcal{Y} \to \mathbb{R}^V$ be a local explanation function that takes an input, and logit pair, and assigns importance scores to a subset (or all) of its input features such that

$$\boldsymbol{e} = \phi_L(f, \boldsymbol{x}, y; \lambda), \tag{1}$$

where $\boldsymbol{e} \in \mathbb{R}^V$ is the explanation output, parameterised by $\lambda$.

A broad variety of local explanation approaches fall within the scope of our work, *e.g.,* gradient-based (Simonyan & Zisserman, 2015; Smilkov et al., 2017; Sundararajan et al., 2017; Bykov et al., 2022; Krishna et al., 2023; Selvaraju et al., 2020), back-propagation-based (Bach et al., 2015; Shrikumar et al., 2017), model-agnostic (Zeiler & Fergus, 2014a; Lundberg & Lee, 2017), local surrogate (Ribeiro et al., 2016a), attention-based (Chefer et al., 2021; Covert et al., 2022), or prototypical explanation methods (Simonyan & Zisserman, 2015). More recent approaches (Krishna et al., 2023; Kroeger et al., 2023) that leverage separate LLMs as the explanation function $\phi$ to interpret local predictions in a post-hoc manner, are also within the scope of this work.

**Global Explanations.** To study the model $f$ from a *global* point of view, an explanation is produced independent of a specific instance, *i.e.,* $\boldsymbol{x}$. Here, a global explanation method $\phi_G : \mathcal{F} \times \mathcal{Y} \to \mathbb{R}^V$ takes a trained model $f$, and generates an explanation $\boldsymbol{e} \in \mathbb{R}^V$ for specific neural activation associated with a target class $c$. Here, $c$ us represented by logit $y$ such that

$$\boldsymbol{e} = \phi_G(f, y; \kappa), \tag{2}$$

where $\phi_G$ is parameterised by $\kappa$. Here $\phi_G$ may be variants of activation-maximisation (or "feature visualisation") which provide either natural, or synthetic data points of maximal activation (Berkes & Wiskott, 2006; Erhan et al., 2009; Olah et al., 2017; Nguyen, 2020; Fel et al., 2024) or concept-based explanations Bykov et al. (2023). Recently, trained SAEs Bricken et al. (2023); Lieberum et al. (2024); Huben et al. (2024) have emerged as an alternative formulation for $\phi_G$, aiming to produce interpretable "monosemantic" feature encodings at the layer level, providing insight into a model's intermediate representations.

For convenience, we let $\phi \in \mathcal{E}$ denote $\phi_L$, and $\phi_G$ although they formally reside in different spaces. To avoid label leakage (Jethani et al., 2023), we use the predicted class (and not the true class) to generate the explanation $\boldsymbol{e}$.

### 2.1.1 Estimate Explanation Quality

Without ground truth explanation labels, the task of estimating the quality of an explanation $\phi$ is non-trivial. To approximate explanation quality, researchers rely on metric-based heuristics (or "metrics"). Following Hedström et al. (2023a), we define a general evaluation function $\Psi_\tau : \mathcal{E} \times \mathcal{X} \times \mathcal{F} \times \mathcal{Y} \to \mathbb{R}$

$$q = \Psi(\phi, \boldsymbol{x}, f, y; \tau) \tag{3}$$

which returns a quality estimate $q \in \mathbb{R}$, indicating the quality of a given explanation, parameterised by $\tau$. When global explanations $\phi_G$ are evaluated, $\boldsymbol{x}$ is omitted from Equation 3. Unless required, we omit hyperparameters $\tau, \lambda, \kappa, \zeta$ for notational convenience.

### 2.1.2 Related Works

Within approaches that evaluate explanation quality by approximation, we concentrate on those that examine the *functional relationship* between the explanation, and the model through means of perturbation, *i.e.,* assessing qualities such as robustness, sensitivity, and faithfulness. These are briefly introduced below, and mathematically formalised in Section 2.3.

**Robustness.** Robustness (also referred to as "continuity", and "stability") methods evaluate the explanation function's resilience to infinitesimal input noise, and is a widely used evaluation technique (Yeh et al., 2019; Montavon et al., 2018; Alvarez-Melis & Jaakkola, 2018b; Nguyen & Martinez, 2020; Agarwal et al., 2022a; Dasgupta et al., 2022). Most commonly, robustness is evaluated by first perturbing an input sample, then generating the explanation for the perturbed input, and finally comparing this explanation to the original explanation. Higher similarity between the original, and perturbed explanation indicates higher quality. Existing robustness measures differ in how noise is applied to the input (*e.g.,* using a Gaussian (Alvarez-Melis & Jaakkola, 2018b; Yeh et al., 2019) or a uniform distribution (Agarwal et al., 2022a)), and how explanation similarity is measured (*e.g.,* Yeh et al. (2019) computes difference with *Monte-Carlo* sampling, and Alvarez-Melis & Jaakkola (2018b); Agarwal et al. (2022a) rely on variants of a *Lipschitz* constant).

**Sensitivity.** Sensitivity (or "randomisation") methods (Adebayo et al., 2018; Hedström et al., 2024) act complementary to robustness, and assesses a critical, and perhaps indisputable evaluative quality: that the explanation function $\phi$ should be sensitive to a randomisation of model parameters. Existing sensitivity measures differ in how the change in the explanation outputs is measured (*e.g.,* Adebayo et al. (2018) relies on *Structural Similarity Index* (SSIM), and Hedström et al. (2024) uses discrete entropy calculations), and how perturbation is applied (*e.g.,* Adebayo et al. (2018) randomises model parameters layer-by-layer in a *top-down* fashion, and Hedström et al. (2024) uses *bottom-up* or *full* parameter randomisation). The sensitivity criterion asks that the explanation should change significantly when the model parameters are randomised, whether layer-by-layer (Adebayo et al., 2018) or entirely (Hedström et al., 2024).

**Faithfulness.** Faithfulness (or "fidelity") methods (Bach et al., 2015; Samek et al., 2017; Montavon et al., 2018; Ancona et al., 2018; Rieger & Hansen, 2020; Dasgupta et al., 2022; Bhatt et al., 2020; Rong et al., 2022; Atanasova et al., 2023; Blücher et al., 2024; Chuang et al., 2024) evaluate explanations by gradually perturbing the input based on the importance of pixels or tokens indicated by the explanation, and observing the resulting degradation in model performance. Mehthods differ in how model responses are reported (with logits (Alvarez-Melis & Jaakkola, 2018a; Yeh et al., 2019; Bhatt et al., 2020) or softmax probabilities (Montavon et al., 2018; Ancona et al., 2018; Rieger & Hansen, 2020; Nguyen & Martinez, 2020; Dasgupta et al., 2022; Rong et al., 2022)), how perturbations are ordered (ascending (Arya et al., 2019; Nguyen & Martinez, 2020) or descending (Bach et al., 2015; Samek et al., 2017; Rong et al., 2022)), and in the general approach to perturbation (whether using single-pixel changes (Bach et al., 2015), patch-based masking with a constant value (Samek et al., 2017), or linear interpolation (Rong et al., 2022)). Faithfulness methods typically aggregate model responses into a single quality estimate, such as AUC (Bach et al., 2015; Samek et al., 2017; Rong et al., 2022). For faithfulness to be considered fulfilled, the model's performance should rapidly decrease as perturbations are applied—the steeper the degradation, the higher the explanation quality.

Beyond approximation techniques, interpretability researchers have explored alternative ways to evaluate explanation quality, such as using human judgment (Zeiler & Fergus, 2014b; Ribeiro et al., 2016b), and restricting tasks to synthetic or toy environments (Guidotti, 2021; Carmichael & Scheirer, 2023). While such approaches complement evaluation methods that approximate explanation quality, they lack scalability, and generalisability to real-world scenarios, and are not covered in this work.

## 2.2 Perturbation-based Evaluation

A key observation is that robustness, sensitivity, and faithfulness evaluations generally rely on three common steps. First, a perturbation is applied to either the input (*e.g.,* by adding infinitesimal noise) or the model parameters (*e.g.,* by randomisation). Second, the effect of the perturbation is measured on the output of either the explanation function $\phi$ or the model $f$. Third, an interpretation is made to assess whether this change in functional outputs is acceptable given a criterion, such as requiring the distance in explanation outputs to be small when the perturbation is small. We refer to Figure 1 for an illustration.

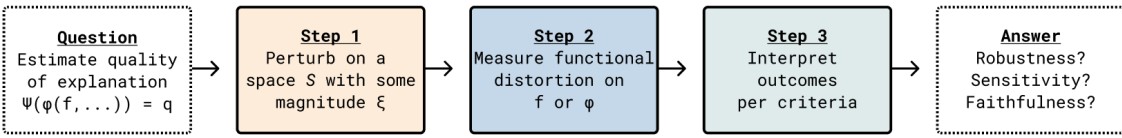

Figure 1: An overview of the "perturb, measure, and interpret" evaluation methodology (Section 2.3).

To facilitate mathematical unification (Section 4), and further insights (Section 5), we next formalise the three steps of perturbation-based evaluation. Therefore, some general notation for perturbation (Eqs 4-5), and measurement (Definition 1) is introduced. By systematising evaluation, we can advance our conceptual understanding, especially in clarifying how existing methods differ, and what attributes are shared.

### 2.2.1 Step 1. Perturbation

First, a perturbation is initiated. This is typically done either on the model parameter space in large magnitudes, *e.g.,* by randomising weights, or on the input in small magnitudes, *e.g.,* by adding Gaussian noise. Alternatively, perturbations can be applied cumulatively, such as by masking pixels or regions of pixels, or by replacing tokens in textual inputs. To accommodate diverse evaluation methods across different data modalities, we follow Hedström et al. (2023a), and define a general perturbation function that can be applied on any real-valued space $\mathcal{S} \subseteq \{\mathcal{X}, \Theta, \mathcal{Y}\}$. Let $\mathcal{P}_{\mathcal{S}} : \mathcal{S} \to \mathcal{S}$ be a perturbation function of $\boldsymbol{s} \in \mathcal{S}$ with parameters $\omega \in \mathbb{R}$ such that

$$\mathcal{P}_{\mathcal{S}}(\boldsymbol{s};\omega) = \hat{\boldsymbol{s}}, \tag{4}$$

where $\forall \hat{\boldsymbol{s}}, \boldsymbol{s} \in \mathcal{S}$, and $\hat{\boldsymbol{s}} \neq \boldsymbol{s}$. For brevity, we may omit $\omega$ such that $\mathcal{P}_{\mathcal{S}}(\boldsymbol{s}) := \mathcal{P}_{\mathcal{S}}(\boldsymbol{s};\omega)$. With Equation 4, we may, *e.g.,* generate a perturbed instance $\hat{\boldsymbol{s}}$ with input perturbation, *i.e.,* $\hat{\boldsymbol{x}} = \mathcal{P}_{\mathcal{X}}(\boldsymbol{x})$ or model parameter

randomisation, *i.e.,* $\hat{\theta} = \mathcal{P}_\Theta(\theta)$. Since robustness, sensitivity, and faithfulness evaluations require distinct perturbation magnitudes, we let $\xi$ denote the difference between $\boldsymbol{s}$, and $\hat{\boldsymbol{s}}$ as follows

$$\delta(\boldsymbol{s}, \hat{\boldsymbol{s}}) = \xi, \tag{5}$$

where $\delta : \mathcal{S} \times \mathcal{S} \to \mathbb{R}$ is a general discrepancy function, *e.g.,* an $\ell_p$-norm, cosine distance or Pearson correlation.

### 2.2.2  Step 2. Measurement

As a second step, the impact of the perturbation is measured on relevant functions. Common approaches include measuring the distance between explanation outputs or recording the change in model responses under random or cumulative masking guided by the explanation output. We define a general approach to measure the perturbation impact on a separate function (*e.g.,* the impact of input perturbation on the model function) below.

**Definition 1 (Functional Distortion)** *Let $\boldsymbol{s}, \hat{\boldsymbol{s}} \in \mathcal{S}$ denote instances in space $\mathcal{S} \subseteq \{\mathcal{X}, \Theta, \mathcal{Y}\}$, before, and after perturbation, respectively. Let $k : \mathcal{S} \to \mathcal{H}$ denote a separate function that maps $\boldsymbol{s}, \hat{\boldsymbol{s}}$ to a distinct space $\mathcal{H} \subseteq \{\mathcal{F}, \mathcal{E}\}$ from $\mathcal{S}$. Then, perturbation impact in function $k$ is measured by functional distortion $\boldsymbol{D}_k : \mathcal{S} \times \mathcal{S} \to \mathbb{R}$ as follows*

$$\boldsymbol{D}_k(\boldsymbol{s}, \hat{\boldsymbol{s}}) = \delta(k(\boldsymbol{s}), k(\hat{\boldsymbol{s}})), \tag{6}$$

*where $k(\boldsymbol{s}) = h$ with $h \in \mathcal{H}$, and $\delta : \mathcal{H} \times \mathcal{H} \to \mathbb{R}$.*

**Model, and Explanation Distortion.** With Definition 1, we can flexibly apply perturbation in one space, and then evaluate the effect in a different space[2]. For example, assume we have applied perturbation on the input space, *i.e.,* $\hat{\boldsymbol{x}} = \mathcal{P}_\mathcal{X}(\boldsymbol{x})$ (Equation 4), and therefore have two instances $\boldsymbol{x}$, and $\hat{\boldsymbol{x}}$. Then, to measure the perturbation impact on the model function $f$, we follow Definition 1, and set $k = f$ where $h = \boldsymbol{y}$. Evaluating $\boldsymbol{D}_f(\boldsymbol{x}, \hat{\boldsymbol{x}})$ from Equation 6 effectively means that we compare model evaluations on perturbed, and non-perturbed inputs, *i.e.,* $\delta(y, \hat{y})$ with $\hat{y} = f_c(\hat{\boldsymbol{x}}; \theta)$ for the same class $c$. Alternatively, to measure perturbation impacts on the explanation function $\phi$, we set $k = \phi$ where $h = \boldsymbol{e}$. Evaluating $\boldsymbol{D}_\phi(\boldsymbol{x}, \hat{\boldsymbol{x}})$, practically means that we compute $\delta(\boldsymbol{e}, \hat{\boldsymbol{e}})$ where $\hat{\boldsymbol{e}} = \phi(\hat{\boldsymbol{x}}, \dots)$ is the explanation *w.r.t.* perturbed input $\hat{\boldsymbol{x}}$. For comparability, $\hat{\boldsymbol{e}}$ is generated *w.r.t.* the same class $c$ as its non-perturbed counterpart $\boldsymbol{e}$. Similarly, to compute functional distortion after parameter perturbation, *i.e.,* $\hat{\theta} = \mathcal{P}_\Theta(\theta)$, we compute $\boldsymbol{D}_f(\theta, \hat{\theta})$, and $\boldsymbol{D}_\phi(\theta, \hat{\theta})$ using logit $\hat{y} = f(\boldsymbol{x}; \hat{\theta})$, and explanation $\hat{\boldsymbol{e}} = \phi(f_{\hat{\theta}}, \dots)$, respectively. To generalise the notation across different perturbation types, we let $\boldsymbol{D}_f$, and $\boldsymbol{D}_\phi$ denote the model, and explanation distortion quantities, respectively.

### 2.2.3  Step 3. Interpretation

In the final step of the evaluation workflow (Figure 1), the distortion quantities are examined separately according to their evaluative criteria. For example, if robustness is evaluated, generally low values for $\boldsymbol{D}_\phi$ are expected, assuming perturbation magnitude $\xi$ is small. Conversely, if sensitivity is evaluated, high values for $\boldsymbol{D}_\phi$ are expected, assuming perturbation magnitude $\xi$ is large. If faithfulness is evaluated, model distortion $\boldsymbol{D}_f$ is anticipated to increase as perturbation is cumulatively applied according to the explanation function output. Notably, a key limitation of this step is the need for thresholds to be set by researchers in order to distinguish between low-, and high-quality evaluation outcomes, which has shown could be adversarially manipulated (Wickstrøm et al., 2024).

### 2.3 Formalising Robustness, Sensitivity, and Faithfulness

Equipped with a general perturbation function $\mathcal{P}_\mathcal{S}$ (Equation 4), and its magnitude $\xi$ (Equation 5) as well as a measure to compute functional distortion of the explanation, and model functions (Definition 1), we can combine a wide variety of existing evaluation techniques into general formalisations of robustness, sensitivity,

---

[2]While both the perturbation magnitude $\xi$ (Equation 5), and the distortion $\boldsymbol{D}_k$ (Equation 6) use the discrepancy function $\delta(\cdot, \cdot)$, their outputs differ. Notably, $\xi$ expresses the discrepancy between the original, and perturbed instance, and $\boldsymbol{D}_k$ measures the discrepancy in a distinct space from the perturbation space.

and faithfulness evaluation methods (*cf.* Section 2.1.1). Based on these three main criteria (Definitions 2-4), we show that the validity of each explanation criterion critically depends on fulfilling a separate, implicit model assumption (Assumptions 1-3).

We proceed by presenting a definition of explanation robustness, absorbing the spirit of numerous existing robustness methods[3] (Yeh et al., 2019; Montavon et al., 2018; Alvarez-Melis & Jaakkola, 2018b; Nguyen & Martinez, 2020; Agarwal et al., 2022a).

**Definition 2 (Explanation Robustness)** *Let $\hat{\boldsymbol{x}} = \mathcal{P}_{\mathcal{X}}(\boldsymbol{x})$ be a perturbed input, and $\Psi^{RO}$ be a quality estimator to yield robustness estimates $q^{RO} \in \mathbb{R}$ such that $q^{RO} = \boldsymbol{D}_{\phi}(\boldsymbol{x}, \hat{\boldsymbol{x}})$. Given thresholds $\alpha, \varepsilon_{\boldsymbol{D}_{\phi}}^{RO} \in \mathbb{R}^{+}$, an explanation function $\phi$ is robust if the perturbation magnitude $\xi^{RO} \leq \alpha$:*

$$q^{RO} \leq \varepsilon_{\boldsymbol{D}_{\phi}}^{RO}. \tag{7}$$

For an explanation function $\phi$ to be considered robust, the estimator $\Psi^{RO}$ should yield low values, *i.e.,* $q^{RO} \leq \varepsilon_{\boldsymbol{D}\phi}^{RO}$, reflecting minor differences between the original explanation $\boldsymbol{e}$, and the perturbed explanation $\hat{\boldsymbol{e}}$. Since the stability expectations of the explanation function $\phi$ are dictated by the robustness of $f$ (Yeh et al., 2019; Chalasani et al., 2020; Agarwal et al., 2022a; Tan & Tian, 2023), it would be false to expect $\phi$ to exhibit robustness if its underlying model is not robust. Consequently, the validity of Equation 7 depends on the fulfillment of model robustness (Assumption 1).

**Assumption 1 (Model Robustness)** *Given an input perturbation $\mathcal{P}_{\mathcal{X}}$ of magnitude $\xi^{RO}$, and thresholds $\alpha, \varepsilon_{\boldsymbol{D}_{f}}^{RO} \in \mathbb{R}^{+}$, $\xi^{RO} \leq \alpha$, the model distortion (Equation 6) is bounded by $\boldsymbol{D}_{f}(\boldsymbol{x}, \hat{\boldsymbol{x}}) \leq \varepsilon_{\boldsymbol{D}_{f}}^{RO}$.*

In line with works of Adebayo et al. (2018); Hedström et al. (2024), we define explanation sensitivity in the following.

**Definition 3 (Explanation Sensitivity)** *Let $\hat{\theta} = \mathcal{P}_{\Theta}(\theta)$ create a model $f_{\hat{\theta}}$ with perturbed parameters, and $\Psi^{SE}$ be a quality estimator that yields sensitivity estimates $q^{SE} \in \mathbb{R}$ such that $q^{SE} = \boldsymbol{D}_{\phi}(\theta, \hat{\theta})$. Given thresholds $\alpha, \varepsilon_{\boldsymbol{D}_{\phi}}^{SE} \in \mathbb{R}^{+}$, an explanation function $\phi$ is sensitive if the perturbation magnitude $\xi^{SE} > \alpha$:*

$$q^{SE} > \varepsilon_{\boldsymbol{D}_{\phi}}^{SE}. \tag{8}$$

For $\phi$ to be considered sensitive to randomness, the differences between explanations should be substantial, meaning $\Psi^{SE}$ yields high estimates, *i.e.,* $q^{SE} > \varepsilon_{\boldsymbol{D}_{f}}^{SE}$, reflecting significant discrepancies between $\boldsymbol{e}$, and $\hat{\boldsymbol{e}}$. This expectation that $q^{SE}$ should be large is based on the assumption that the model responded strongly to the perturbation. Similar to how explanation robustness depends on the stability of $f$, the emphasis on a large $q^{SE}$ assumes a different model response. Therefore, the validity of the sensitivity evaluation (Equation 8) depends on model sensitivity (Assumption 2).

**Assumption 2 (Model Sensitivity)** *Given a parameter perturbation $\mathcal{P}_{\Theta}$ of magnitude $\xi^{SE}$, and thresholds $\alpha, \varepsilon_{\boldsymbol{D}_{f}}^{SE} \in \mathbb{R}^{+}$, $\xi^{SE} > \alpha$, the model distortion (Equation 6) is bounded by $\boldsymbol{D}_{f}(\theta, \hat{\theta}) > \varepsilon_{\boldsymbol{D}_{f}}^{SE}$.*

With various existing interpretations of explanation faithfulness (Section 2.1.2), we focus on common criteria to combine these interpretations into a single definition below.

**Definition 4 (Explanation Faithfulness)** *Let $\hat{\boldsymbol{x}}^{z} = \mathcal{P}_{\mathcal{X}}(\boldsymbol{x}; z)$ denote the input after the $z^{th}$ perturbation for $z \in [1, Z]$, where $\mathcal{P}_{\mathcal{X}}$ progressively masks the top-z features according to the indices given by argmax($\boldsymbol{e}$), with perturbation magnitudes $\xi_{z}$ satisfying $\xi_{1} \leq \xi_{2} \leq \ldots \leq \xi_{Z}$. A quality estimator $\Psi^{FA}$ yields a vector of faithfulness estimates $\boldsymbol{q}^{FA} \in \mathbb{R}^{Z}$ with entries $q_{z}^{FA} = f(\hat{\boldsymbol{x}}^{z}, \theta)$, where the overall faithfulness score $q^{FA} \in \mathbb{R}$ is obtained by aggregating these estimates via a function $\nu : \mathbb{R}^{Z} \to \mathbb{R}$:*

$$q^{FA} = \nu(\hat{\boldsymbol{q}}^{FA}). \tag{9}$$

---

[3]Some algorithmic details are omitted in the definition. For completeness, mathematical definitions are provided for each evaluation method in Appendix A.4.5.

When $\nu$ is defined using AUC, a faithful explanation is expected to produce low aggregated scores $q^{FA}$ (Equation 9). The conventional expectation in faithfulness evaluation (Bach et al., 2015; Samek et al., 2017; Rong et al., 2022) is that significant distortions should occur early, as the "more important features" are removed first. To ensure that the faithfulness score is solely driven by the quality of $\phi$, and not by other factors, such as out-of-distribution samples (OOD) (Hase et al., 2021; Hesse et al., 2024), non-linear feature effects or artifacts introduced by cumulative perturbations (Hooker et al., 2019; Brunke et al., 2020; Hase et al., 2021; Rong et al., 2022; Brocki & Chung, 2022), the model distortion to these perturbations should be monotonically non-decreasing, *i.e.,* satisfy model faithfulness (Assumption 3).

**Assumption 3 (Model Faithfulness)** *Given $Z$ cumulative perturbations $\mathcal{P}_\mathcal{X}$ of magnitudes $\xi_z$ with $\xi_1 \leq \xi_2 \leq \cdots \leq \xi_Z$ the corresponding model distortions (Equation 6) are: $\boldsymbol{D}_f^1 \leq \boldsymbol{D}_f^2 \leq \cdots \leq \boldsymbol{D}_f^Z$ with $\boldsymbol{D}_f^z = \boldsymbol{D}_f(\boldsymbol{x}, \hat{\boldsymbol{x}}^z)$.*

### 2.4  Model Assumptions in Practice

Evaluations under Definitions 2-4 typically assume that model distortions are proportional to perturbation magnitudes, *i.e.,* that larger perturbations lead to greater distortions, and smaller perturbations result in lesser distortions. This naturally raises the question: with commonly used perturbation techniques for evaluating robustness (*e.g.,* additive Gaussian noise), sensitivity (*e.g.,* layer-wise randomisation), and faithfulness (*e.g.,* cumulative input masking, does this assumption hold in practice? In Appendix A.5, we extensively analyse the extent to which Assumptions 1-3 hold versus fail across various explanation methods, and NN models.

Notably, we find that Assumptions 1-3 are systematically violated in practice. While this is expected due to the inherent non-linearity embedded in $f$, it has significant consequences for the validity of existing evaluations (Definitions 2-4). Evaluation outcomes may be misleading when explanation robustness is enforced for models that fundamentally lack it (Chalasani et al., 2020; Tan & Tian, 2023; Agarwal et al., 2022a), or when faithfulness scores are attributed to explanation quality without considering OOD scenarios (Hase et al., 2021; Hesse et al., 2024). In Section 3.3, we propose a mitigation strategy to address this issue.

## 3  A Unifying Perspective

With clear definitions of robustness, sensitivity, and faithfulness evaluations (Section 2.3), we may now explore their shared attributes, and outcomes. In the following, we discuss the unifying aspects of these evaluations, and introduce a novel definition to evaluate faithfulness, which integrates these distinct evaluations into a single criterion of explanation quality.

### 3.1  Unifying Attributes

Upon formalising the evaluation criteria (Definitions 2-4), a notable observation is that robustness, sensitivity, and faithfulness exhibit common attributes. Each of the evaluative criteria (1) introduces a *perturbation* of a specific magnitude $\xi$, (2) *measures* the functional effect, and (3) *interprets* the results, *i.e.,* the quality estimate $q$. Also, the evaluation is performed under distinct *model assumptions* about its distortion $\boldsymbol{D}_f$. We refer to Table 1 for a summary of these findings.

In Figure 2 (A), we illustrate these theoretical similarities on a graph, with axes corresponding to the shared attributes $\xi$, and $q$. Here, we can observe that robustness evaluation (*green*) involves minimal perturbation with a small difference in expected explanation output (or low $q$). Sensitivity (*red*) employs substantial perturbation, expecting a significant difference in explanation output (or high $q$). Faithfulness (*blue*) uses cumulative perturbation of $Z$ steps, evaluating the corresponding variations in model output. By placing the different perspective of explanation quality onto Figure 2 (A), and thereafter examining the positions of the post-perturbed instances $\hat{\boldsymbol{s}} \in \mathcal{S}$, we can advance our understanding of how the criteria relate to one another: specifically, that diverse evaluation methods can be unified under a shared conceptual framework.

Table 1: A concise overview of the attributes of the *robustness*, *sensitivity*, and *faithfulness* evaluations. The last row presents our *unified* evaluation proposal (Def 5), whose theory and practical implementation are described in Section3 and Section5, respectively.

| EVALUATION ($\Psi$) (DEFINITIONS 2-4) | **Step 1.** PERTURBATION; MAGNITUDE (EQUATIONS 4-5) | **Step 2.** MEASUREMENT (DEFINITION 6) | **Step 3.** INTERPRETATION (EQUATION 7-9) | MODEL ASSUMPTIONS (ASSUMPTIONS 1-3) |
|---|---|---|---|---|
| ROBUSTNESS ($\Psi^{RO}$) SENSITIVITY ($\Psi^{SE}$) FAITHFULNESS ($\Psi^{FA}$) | $\mathcal{P}_\mathcal{X}(\boldsymbol{x}); \xi^{RO} \leq \alpha$ 
 $\mathcal{P}_\Theta(\theta); \xi^{SE} > \alpha$ 
 $\mathcal{P}_\mathcal{X}(\boldsymbol{x},z); \xi_1^{FA} \leq \xi_2^{FA} \leq \cdots \leq \xi_Z^{FA}$ | $\boldsymbol{D}_\phi(\boldsymbol{x},\hat{\boldsymbol{x}})$ 
 $\boldsymbol{D}_\phi(\theta,\hat{\theta})$ 
 $f(\hat{\boldsymbol{x}}^z,\theta)$ | $q^{RO} \leq \varepsilon_{\boldsymbol{D}_\phi}^{RO}$ 
 $q^{SE} > \varepsilon_{\boldsymbol{D}_\phi}^{SE}$ 
 $q^{FA} = \nu(\hat{\boldsymbol{q}}^{FA})$ | $\boldsymbol{D}_f \leq \varepsilon_{\boldsymbol{D}_f}^{RO}$ 
 $\boldsymbol{D}_f > \varepsilon_{\boldsymbol{D}_f}^{SE}$ 
 $\boldsymbol{D}_f^1 \leq \boldsymbol{D}_f^2 \leq \cdots \leq \boldsymbol{D}_f^Z$ |
| UNIFIED ($\Psi^{GEF}$) | $\mathcal{P}_\Theta(\theta,z); \xi_1^{GEF} \leq \xi_2^{GEF} \leq \cdots \leq \xi_Z^{GEF}$ | $\boldsymbol{D}_\phi(\theta,\hat{\theta})$, AND $\boldsymbol{D}_f(\theta,\hat{\theta})$ | $\rho(\boldsymbol{d}_f,\boldsymbol{d}_\phi) \approx 1$ | NONE |

## 3.2 Unifying Outcomes

Another point of unification emerges when considering the outcomes of these evaluation criteria, and how they interact in practice. In Figure 2 (B), similar to the traditional confusion matrix (in ML) or contingency table (in statistics), we provide a visual representation of the possible model, and explanation outcomes, post-perturbation. Although model, and explanation outcomes are typically continuous in reality—for conceptual clarity, we classify them into four distinct quadrants: true positive (TP), true negative (TN), false positive (FP), and false negative (FN). A key benefit of *discretising* evaluation outcomes in this way, is that we can distinguish between *aligned*, and *misaligned* explanation behaviour:

- *Aligned outcomes (TP + TN).* The green quadrant represents outcomes where the explanation, and model agree, indicating explanation robustness, *i.e.,* $\boldsymbol{e} = \hat{\boldsymbol{e}}$, and $y = \hat{y}$, and satisfying Assumption 1. Conversely, the red quadrant contains outcomes where both explanation, and model outputs differ, reflecting explanation sensitivity, *i.e.,* $\boldsymbol{e} \neq \hat{\boldsymbol{e}}$, and $y \neq \hat{y}$, and satisfying Assumption 2. Explanation faithfulness is achieved when evaluation outcomes are aligned over $Z$ steps (Assumption 3).

- *Misaligned outcomes (FP + FN).* The orange quadrants highlight misalignment between $\phi$, and $f$. The top-left quadrant shows explanation dissimilarity despite prediction stability (*i.e.,* $y = \hat{y}$, and $\boldsymbol{e} \neq \hat{\boldsymbol{e}}$), failing Assumption 2. The bottom-right quadrant shows explanation similarity despite a prediction change (*i.e.,* $y \neq \hat{y}$, and $\boldsymbol{e} = \hat{\boldsymbol{e}}$), failing Assumption 1.

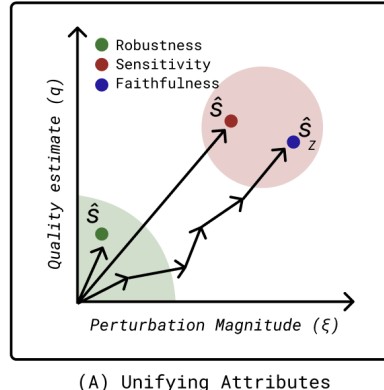
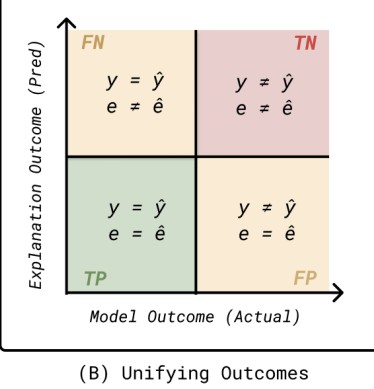
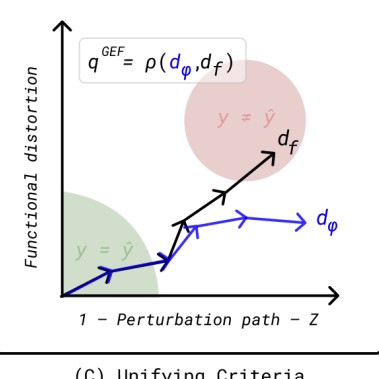

(A) Unifying Attributes  (B) Unifying Outcomes  (C) Unifying Criteria

Figure 2: Intuition behind the relationship between robustness, sensitivity, and faithfulness evaluations. (A) illustrates the shared attributes, *i.e.,* perturbation magnitude $\xi$, and quality estimate $q$ that unifies robustness (*green*), sensitivity (*red*), and faithfulness (*blue*) evaluations. (B) displays a confusion matrix of *discretised* model, and explanation outcomes, with green, and red quadrants indicating aligned behaviour, and orange quadrants showing misalignment. (C) shows our proposed GEF criteria (Definition 5) which measures explanation to model alignment over diverse evaluation perspectives.

**Explanation Faithfulness is Alignment.** Our analysis reveals that evaluation according to Definitions 2-4, fundamentally concerns the *alignment* between the explanation, and the model's behaviour, whether across single (Definitions 2-3) or multiple (Definition 4) perturbation steps. A key observation is that

existing robustness, and sensitivity measures provide a limited view of isolated model conditions: robustness evaluates alignment when the model's predictions remain stable (TP quadrant), while sensitivity evaluates alignment when predictions change (TN quadrant). Faithfulness (Definition 4) evaluates alignment over $Z$ steps, assuming non-decreasing, monotonic model responses under cumulative perturbations (Assumption 3). These *singular* perspectives require strict adherence to specific model conditions, and consequently fail to evaluate the full behaviour of the explanation function. Next, we propose a unifying criterion for faithfulness evaluation. We refer to Figure 2 (C) for an illustration.

### 3.3 Unifying Criteria

We extend, and generalise the current faithfulness criterion (Definition 4) by integrating the robustness, and sensitivity evaluations into a combined criterion, that is free from restrictive model assumptions. Using a series of $Z$ perturbations, we measure explanation alignment across a spectrum of model outcomes—from cases where model predictions remain consistent, *i.e.,* $y = \hat{y}$, to those where predictions diverge, *i.e.,* $y \neq \hat{y}$. In this way, a generalised definition of explanation faithfulness is obtained.

**Definition 5 (Generalised Explanation Faithfulness)** *Let* $\boldsymbol{d}_f = [\boldsymbol{D}_f^1, \boldsymbol{D}_f^2, \ldots, \boldsymbol{D}_f^Z]$ *and* $\boldsymbol{d}_\phi = [\boldsymbol{D}_\phi^1, \boldsymbol{D}_\phi^2, \ldots, \boldsymbol{D}_\phi^Z]$ *be the model, and explanation distortion vectors, where* $\boldsymbol{D}_f^z$*, and* $\boldsymbol{D}_\phi^z$ *are distortion quantities of the* $z^{th}$ *step along a perturbation path* $z \in [1, Z]$*, from robustness at* $z = 1$ *to sensitivity at* $z = Z$ *such that* $\forall y, \hat{y} \in Y$:

$$(z = 1 : y = \hat{y}) \quad and \quad (z = Z : y \neq \hat{y}),$$

*where* $\hat{y}$*, and* $y$ *are perturbed versus unperturbed model outputs, respectively. Let* $\Psi^{GEF}$ *be a quality estimator that yields estimates* $q^{GEF} \in \mathbb{R}$ *via the correlation coefficient* $\rho : \mathbb{R}^Z \times \mathbb{R}^Z \to \mathbb{R}$ *such that* $q^{GEF} = \rho(\boldsymbol{d}_{f,\phi})$*. An explanation function* $\phi \in \mathcal{E}$ *is faithful to* $f \in \mathcal{F}$ *if:*

$$q^{GEF} \approx 1. \tag{10}$$

With Equation 10, we define a quality estimator $\Psi^{GEF}$ that yields values ranging between $[-1, 1]$, with a value of 1 implying perfect generalised faithfulness, 0 suggesting an absence of it, and $-1$ an inverse relationship. GEF estimation is, therefore, *threshold-free* in the sense that the correlation coefficient directly indicates the quality of the explanation, eliminating the need for arbitrary cut-offs. Note that in Definition 5, we implicitly rely on predicted class $c$ to generate the perturbed logit $\hat{y}$ as the target for the explanation, and model distortion. In Appendix A.1.2, we discuss a broader application of GEF where the targets $\hat{y}$, and $y$ are replaced by any $c^{th}$ neuron within a layer $l \in [1, L]$ of a feed-forward model. Moreover, Definition 5 applies to a wide range of explanation functions, as discussed in Section 2.1. The choice of $\rho$, and perturbation applied to construct the distortion vectors depends on the practical implementation (Section 5.2).

**Remarks.** Our definition shares similarities with faithfulness estimation (Definition 4) in that it assesses explanation quality along a perturbation path. However, it fundamentally differs by focusing on *general alignment* rather than a specific scenario of measuring the *magnitude* of model response to *cumulative input* perturbation. A key benefit of our proposal is that we use the model distortion to *anchor* the expectations of the explanation distortion, and as such, eliminate the need to rely on arbitrary thresholds. In this way, the evaluation will be grounded in the exact functional response of the model, and thus resilient to OOD scenarios: expecting small explanation distortions only when model distortions are small, and vice-versa.

**Theoretical Benefits.** A good faithfulness measure should assign low scores to unfaithful explanations, and high scores to faithful explanations. In Appendix A.1.3, we prove that a linear model $f = \theta\boldsymbol{x} + c$ where $\theta$ acts as the explanation, attains a perfect faithfulness score, *i.e.,* $q^{GEF} = 1$ with GEF. Conversely, unfaithful explanations are penalised by GEF. For instance, constant explanations that generate no distortion, *i.e.,* $\boldsymbol{D}_\phi(\boldsymbol{e}, \hat{\boldsymbol{e}}) = 0$, pass the conventional robustness test $q^{RO} \leq \varepsilon_{\boldsymbol{D}_\phi}^{RO}$ (Definition 2), but correctly fails in the GEF formulation. Similarly, random explanations (*e.g.,* generated by uniform sampling, *i.e.,* $\hat{\boldsymbol{e}}_i \sim \mathcal{U}(0, 1)$) produce maximal distortion (Binder et al., 2022), and thus generally pass the sensitivity test, *i.e.,* $q^{SE} > \varepsilon_{\boldsymbol{D}_\phi}^{SE}$ (Definition 3) but fails in our definition. We provide proofs for both cases in Appendix A.1.1, and outline empirical evidence in (Section 4.2).

# 4 A Geometric Perspective

With an advanced understanding of explanation faithfulness (Section 3), we can more systematically study the behaviour of the explanation function. Without assuming the restricted model conditions (Assumptions 1-3) are met, which are often violated in practice (Appendix A.5), we can more objectively measure the *true* explanation function behaviour. We can formalise questions such as: do explanation functions in real-world evaluation scenarios align or misalign with their model? How do different types of perturbation impact model and explanation functions? In the following, and in Appendix A.6, we empirically examine such questions across common explanation methods (Section 2.1) for different NN models. Guided by differential geometry, we provide theoretical considerations on the impact of geometry (Section 4.2).

## 4.1 Explanation Alignment Patterns

To empirically analyse whether explanation functions are aligned with their model, we study how the distortions of various local, and global explanation functions, and models change under perturbation. Here, we use additive Gaussian noise, *i.e.,* $\nu_i \sim \mathcal{N}(0, \sigma)$ to generate perturbed inputs $\hat{\boldsymbol{x}}_i = \boldsymbol{x} + \nu_i$, with a standard deviation $\sigma$ increasing until the model behaves randomly (*i.e.,* with an accuracy equal to $1/C$) using $Z = 10$ perturbation steps. We refer to Table 2 in Section 6 for details regarding datasets, and models, and to Appendix A.6 for extended results.

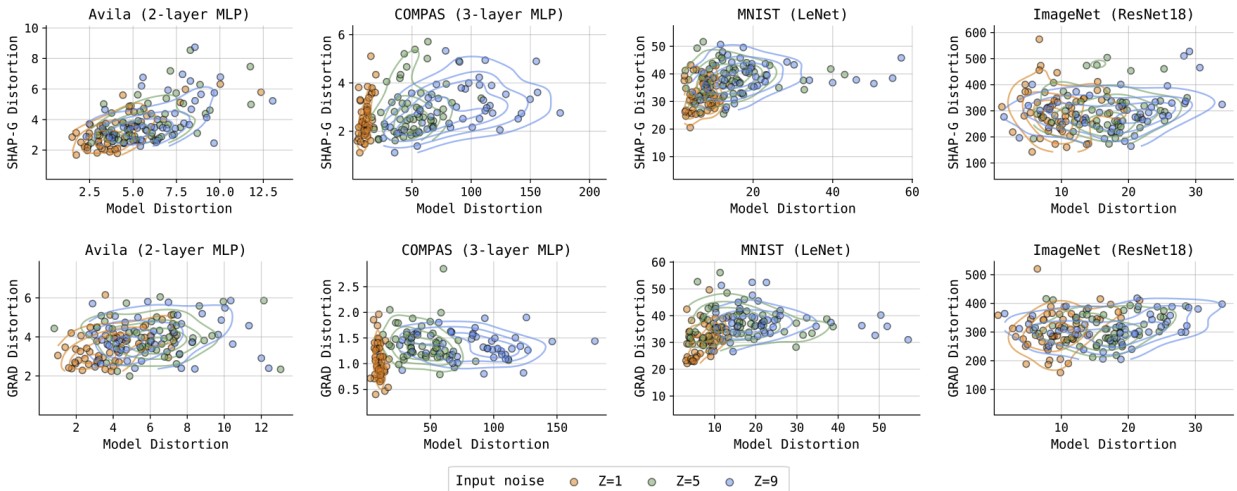

Figure 3: Model (x-axis), and explanation distortions (y-axis) under varying levels of additive Gaussian input noise for vision, and tabular tasks. The scatter points represent individual samples, coloured by perturbation magnitude (z=1, z=5, z=9), with overlapping contours highlighting the relative alignment patterns. The top, and bottom rows represent *GradientSHAP* (SHAP-G), and *Gradient* (GRAD) explanations, respectively.

Figure 3 (top, and bottom) presents the results, which can be interpreted as a continuous analogue of the confusion matrix presented in Section 3.2. The scatter points, coloured by perturbation magnitude, reveal that $\phi$, and $f$ rarely align fully. Instead, the relative alignment varies with both the model, and the explanation functions. Here, we include *GradientSHAP* (SHAP-G) (Lundberg & Lee, 2017), and *Gradient* (GRAD) (Morch et al., 1995; Baehrens et al., 2010)) on top, and bottom rows in Figure 3, respectively, with more results in Appendix A.6). The overlapping contours (*e.g.,* Avila results in Figure 3) underscore a simple but nonetheless systematically overlooked aspect of perturbation-based evaluation (Section 2.3): that a uniform perturbation of its inputs may affect highly non-linear systems in a non-uniform way. If the effects were uniform, the system would likely be linear.

## 4.2 The Impact of Geometry

By considering the geometric nature of the spaces these functions inhabit, we can understand the observed misalignment better. In differential geometry, each space—whether it is the model output space $\mathcal{Y}$ or the

explanation output space $\mathcal{E}$—can be viewed as a manifold with its unique geometric characteristics (Lee, 2012). When a perturbation is applied, a new point on these manifolds may be accessed, and then, when functional distortion (Definition 1) is computed in each space, we are effectively computing a distance between two points on each manifold. For example, with model parameter perturbation, *i.e.*, $\hat{\theta} = \mathcal{P}_\Theta(\theta)$ (Equation 4) we obtain perturbed model outputs $\hat{\boldsymbol{y}}$ (or, a logit $\hat{y}$) given $\hat{\boldsymbol{y}} = f_{\hat{\theta}}(\boldsymbol{x})$. From this, model distortion (Definition 1) is calculated using, *e.g.,* Euclidean distance between the original, and the perturbed instance.

A key observation is that, when distances in two different spaces are directly compared, we ignore the fact that manifolds have their own separate geometric characteristics which are distinct, where distances in one space not necessarily reflect equivalent distances in another. In direct comparisons such as correlation (Ancona et al., 2018; Bhatt et al., 2020) or Lipschitz calculations (Alvarez-Melis & Jaakkola, 2018a; Agarwal et al., 2022a), a global flat metric is assumed. We refer to Figure 4 (A), and (B) for an illustration of the problem of ignoring the impact of geometry. As a result, the quality estimation of the explanation may be misleading.

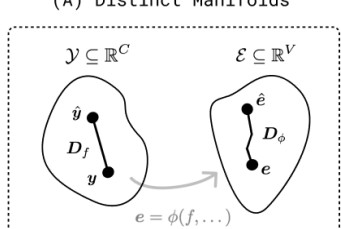 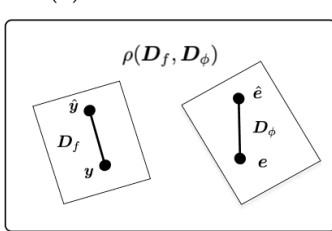 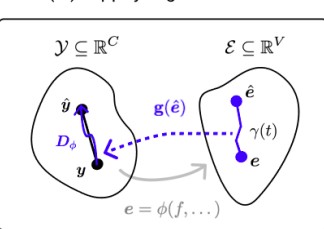

Figure 4: An illustration of the relationship between the manifolds of the model, and explanation. (A) shows how the explanation function maps between the model, and explanation spaces, $\mathcal{Y}$, and $\mathcal{E}$. (B) displays the problem with directly comparing distortions across spaces, assuming a global flat metric. (C) illustrates the pullback operation using metric tensor $\mathbf{g}$ to adjust distortions in $\mathcal{E}$ for comparison in $\mathcal{Y}$.

### 4.3 Reconciling Geometric Discrepancies

To enable a geometrically sound comparison between explanation, and model distortion, the aim is to recompute $\boldsymbol{D}_\phi$ to incorporate the non-linear mappings used in generating explanations. This can be achieved by mapping the distortion from the explanation space $\mathcal{E}$ to the model space $\mathcal{Y}$, effectively "pulling back" the measured distance into $\mathcal{Y}$ (see Ch. 11 of Lee (2012) for further details). Guided by differential geometry, we create a metric tensor $\mathbf{g}$ that serves as this pullback onto $\mathcal{Y}$. This process is illustrated as $\mathbf{g}(\hat{\boldsymbol{e}})$ in Figure 4 (C).

To construct the metric tensor $\mathbf{g}$, we consider an infinitesimal neighbourhood around the parameter perturbation $\theta + du$, for a fixed $\boldsymbol{x}$, and $y$. By applying a first-order Taylor expansion in this neighbourhood, we obtain

$$\phi(f_{\theta+du},\ldots) \approx \boldsymbol{e} + J_f du, \tag{11}$$

where $J_f \in \mathbb{R}^{V \times C}$ is the Jacobian for fixed input $\boldsymbol{x}$, with elements $J_{i,j} = \frac{\partial e_i}{\partial f_j}$. We use $f_j$ as shorthand for $f_j(\boldsymbol{x})$. Effectively, $\theta + du$ yields a new perturbed model $f_{\hat{\theta}}$ which is computed with model parameter perturbation (Section 5.1). With Equation 11, we can compute the elements of the pullback tensor $\mathbf{g} \in \mathbb{R}^{V \times V}$ as the sum of the resulting changes in each explanation element $e_v$ *w.r.t.* the changes in each model element $f_j$

$$g_{i,j}(\boldsymbol{e}) = \sum_{v=1}^{V} \frac{\partial e_v}{\partial f_i} \frac{\partial e_v}{\partial f_j}. \tag{12}$$

Thus, Equation 12 captures the sensitivity of $\phi$ to model output changes, with $\mathbf{g}$ corresponding to the squared Jacobian $\mathbf{g} = J_f^\top J_f$. In this way, we can obtain a more reliable measurement of distances in the *pseudo-Riemannian manifold* $(\mathcal{Y}, \mathbf{g})$ of space $\mathcal{Y}$.

With the pullback metric tensor **g** in place, we can measure explanation distortion that is equivalent to computing the path length under the induced parameter changes in the "pulled-back" space

$$\boldsymbol{D}_\phi := L(\gamma) = \int_0^1 \frac{d\gamma(t)}{dt}^\top g_{\gamma(t)} \frac{d\gamma(t)}{dt} dt, \tag{13}$$

where $\gamma(t)$ is a path between endpoints $\boldsymbol{e}, \hat{\boldsymbol{e}} \in \mathcal{E}$ derived from the original, and perturbed models, respectively. Here, $t$ denotes the step size. Now, we replace our original definition of explanation distortion $\boldsymbol{D}_\phi = \delta(\boldsymbol{e}, \hat{\boldsymbol{e}})$ (Definition 1) with the *total accumulated* distortion along the path, *i.e.,* Equation 13. Here, longer paths correspond to greater distortions. Upon taking this geometric perspective, we can study $\mathcal{Y}$ using extrinsically-defined geometry, and contrast it with the simpler assumption of a flat, intrinsic Euclidean metric. As a result, $\boldsymbol{D}_\phi$, and $\boldsymbol{D}_f$ are more fairly compared in the same space.

## 5 Method: From Theory to Practice

While our unified theory (Section 3), and solution to reconcile geometric discrepancies in measurement (Section 4), provide first steps towards resolving issues in perturbation-based evaluation, many practical concerns remain regarding the choice of perturbation. In this section, describe how to reliably translate our theory (Definition 5) to practice—we propose a general-purpose, task-agnostic perturbation technique based on model parameter scaling (Section 5.1), and introduce the full evaluation algorithms, *i.e.,* `GEF` and `Fast-GEF`(Section 5.2).

### 5.1 Selecting Perturbation Strategy

While all perturbation-based evaluations inherently require parameterisation, input-based perturbation (Definitions 2 and 4), has proven particularly challenging to calibrate (Sturmfels et al., 2020; Haug et al., 2021). Without ground truth labels, selecting parameters such as patch size, pixel, or token replacement strategies is typically based on researchers' judgment. Small changes to input parameters have been shown to significantly impact evaluation outcomes (Brunke et al., 2020; Brocki & Chung, 2022; Rong et al., 2022; Blücher et al., 2024), raising concerns about reliability.

Moreover, perturbing on the input space is not only impractical from a practitioner's standpoint but also compromises impartiality—if parameters must be adjusted for each model, and dataset, how can task-specific confounds be controlled? In Appendix A.5.1, we provide empirical evidence for the existence of confounds in faithfulness evaluations (Definition 4).

Researchers need a general-purpose, dataset, and architecture-agnostic perturbation strategy that facilitates evaluation across distinct explanation approaches (*e.g.,* local, and global methods), and magnitudes, *i.e.,* $\xi$. Following Bykov et al. (2022), we propose a simple perturbation strategy in the following.

**Model Parameter Scaling.** *Introduce perturbations* $\forall z \in [1, Z]$ *by scaling parameters* $\theta \in \mathbb{R}^W$ *with Gaussian noise* $\eta_i \sim \mathcal{N}(\mathbf{1}, \sigma_z^2 \mathbb{1})$, *and* $\sigma_z^2 \in \mathbb{R}^+$ *such that* $\hat{\theta}_z = \theta \cdot \eta_i$, *yielding a perturbed model* $f_{\hat{\theta}_z}$.

By systematically perturbing model parameters instead of the input, from low to high magnitudes with incremental increases of $\sigma_z^2$, ranging from robustness at $z = 1$ to sensitivity at $z = Z$, explanation behaviour is evaluated comprehensively, and agnostically across tasks. With $\xi := \delta(f(\boldsymbol{x}), f_{\hat{\theta}_z}(\boldsymbol{x}))$ (Equation 5), we can measure the perturbation impact at each $z^{th}$ step so that robustness, *i.e.,* $y = \hat{y}$, and sensitivity, *i.e.,* $y \neq \hat{y}$ criteria are fulfilled (Definition 5). Our approach contrasts with the model parameter randomisation procedure of Adebayo et al. (2018), which proposes layer-wise *randomisation* in a top-down order, an approach that faces methodological concerns (Sundararajan & Taly, 2018; Binder et al., 2022; Kokhlikyan et al., 2021; Yona & Greenfeld, 2021). For an illustration of how model parameter scaling affects the classifier's decision boundary, we refer to Fig. 1 of Bykov et al. (2022).

### 5.2 Introducing `GEF` Evaluator

From an algorithmic perspective, three steps are necessary to perform the evaluation. First, given a model, and a test set of input-output pairs, we generate perturbed models $f_{\hat{\theta}_1}, \ldots, f_{\hat{\theta}_Z}$ given $Z$ sets of parameters $\hat{\theta}_1, \ldots, \hat{\theta}_Z$ along a perturbation path (see Algorithm 1, line 6). Then, for each model $f_{\hat{\theta}_z}$, we compute the model, and explanation distortion quantities, *i.e.,* $\boldsymbol{D}^z_f$, and $\boldsymbol{D}^z_\phi$, using the pullback tensor **g** (lines 7, 9, and 10). Finally, distortion vectors are constructed, and correlated using $\rho(\boldsymbol{d}_f, \boldsymbol{d}_\phi)$ (lines 14, and 15). Due to the stochastic nature of model perturbation, we repeat this process $M$ times to average out the effects. We refer to Figure 5, and Algorithm 1 for an overview of the steps involved. An ablation study on hyperparameter choices is provided in Appendix A.7.

**GEF Evaluator (Algo. 5)**

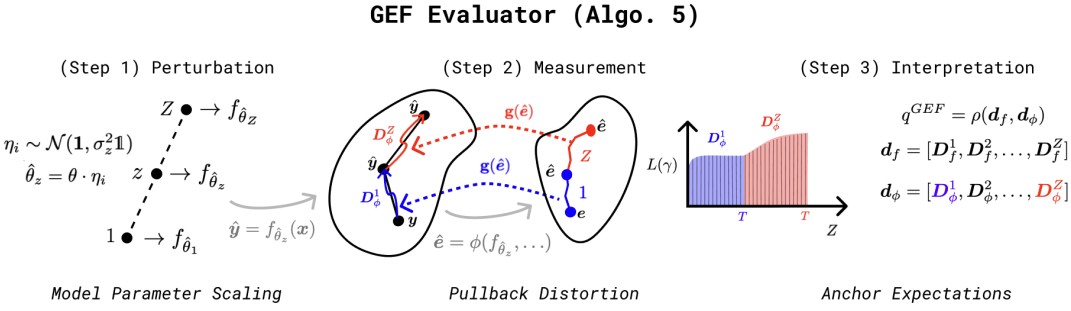

Figure 5: The three steps of `GEF` evaluation (Algorithm 1) to estimate generalised explanation faithfulness (Definition 5).

**Practical Benefits.** Our proposed evaluation (Algorithm 1) provides several practical benefits. First, *anchoring*, negates the need to rely on arbitrary thresholds in evaluation, *e.g.,* when determining a permissible value for the evaluations themselves (Equations 7, 8, and 9) or what perturbation magnitude leads to model alignment for a particular task. Second, perturbing via *model parameter scaling*, at varying intensities combines distinct criteria of explanation quality into a single unified evaluation metric (Section 3.3) that is agnostic to the data, model, and explanation approach. Third, the pullback metric calculation provides a geometrically grounded faithfulness measurement, capturing the true functional impacts of the explanation *w.r.t.* its model.

---

**Algorithm 1** `GEF` Evaluator

---

1: **Require:** Model $f$, explanation function $\phi$, input-prediction pairs $\boldsymbol{x}, y \in \boldsymbol{X}, \boldsymbol{Y}$ with $\boldsymbol{X} \subseteq \mathcal{X}, \boldsymbol{Y} \subseteq \mathcal{Y}$
2: **Parameters:** Integers $Z, M, T, K$, correlation measure $\rho$
3: **for** $\boldsymbol{x}, y$ in range($\boldsymbol{X}, \boldsymbol{Y}$) **do**
4:     $\boldsymbol{e} \leftarrow \phi(f, y, \ldots)$
5:     **for** $z$ in range($Z$) **do**
6:         $\hat{y} \leftarrow f_{\hat{\theta}_z}(\boldsymbol{x})$
7:         $\boldsymbol{D}^z_f \leftarrow \delta(y, \hat{y})$ // Equation (6)
8:         **if** *Fast-GEF* **then**
9:             $\boldsymbol{D}^z_\phi \leftarrow \delta(\boldsymbol{e}, \hat{\boldsymbol{e}})$ with $\hat{\boldsymbol{e}} \leftarrow \phi(f_{\hat{\theta}_z}, \ldots)$ // Equation (6)
10:        **else**
11:            $\boldsymbol{D}^z_\phi \leftarrow$ `compute_path_length`$(f_{\hat{\theta}_z}, \boldsymbol{x}, y, T, K)$ // Equation (13)
12:        **end if**
13:     **end for**
14:     **Construct:** $\boldsymbol{d}_f \leftarrow [\boldsymbol{D}^1_f, \boldsymbol{D}^2_f, \ldots, \boldsymbol{D}^Z_f]$, and $_\phi \leftarrow [\boldsymbol{D}^1_\phi, \boldsymbol{D}^2_\phi, \ldots, \boldsymbol{D}^Z_\phi]$
15:     **Calculate:** $q^{GEF} \leftarrow \rho(\boldsymbol{d}_{f,\phi})$
16:     **Return:** $q^{GEF}$
17: **end for**

---

**Implementation Details.** Unless stated otherwise, we use *Euclidean distance* for $\delta$ in the functional distortion calculations (Definition 1), and define $\rho$ using *Spearman Rank Correlation*, assessing the degree of

monotonic relationship between the distortion quantities. For the experiments, we set $Z = 5$ (see discussion of the influence of $Z$ in Appendix A.1.4) but it is a choice that can be flexibly updated in the open-source implementation. In Appendix A.2, we provide further details on the implementation, including how to generate the perturbation path (line 2), and how to tune parameters (line 2). This also includes information on how we compute the path length (line 11); where we follow an approximation procedure outlined in Equations 18, and 19.

### 5.3 Balancing Computational Constraints

While the pullback operation ensures a fair geometric comparison of distortion quantities, its use of high-dimensional Jacobian calculations, and integral steps (Equation 12) also increases computational demands. To accommodate evaluation contexts involving large model architectures or high-dimensional explanations, we offer an alternative method. For a faster yet *naive* approximation of explanation quality, we omit the pullback operation, and instead define $D_\phi$ according to Equation 6. This approach, entitled `Fast-GEF`, is less computationally demanding, and complements the *exact* approach with pullback, entitled `GEF`, providing a geometrically sound quality estimate.

**Choosing between `GEF` or `Fast-GEF`.** Users can choose between these methods based on their specific computational constraints and demands for accurate quality estimates. We recommend using `GEF` wherever possible due to its ability to account for manifold-specific distortions. However, `Fast-GEF` provides a computationally efficient alternative that is suitable for large-scale tasks or resource-constrained environments. Empirical results show that while the `GEF` or `Fast-GEF` may diverge in individual estimates (Appendix A.6), they often share categorical rankings of explanation methods (Appendix A.8.2).

## 6 Experiments

Our experiments aim to answer the following questions:

- **(Q1)** Are unified, `GEF` and `Fast-GEF` evaluations more empirically reliable than competitive singular approaches?

- **(Q2)** How does generalised faithfulness of local, and global explanation methods compare across distinct data domains?

- **(Q3)** How faithful are LLMs as a top-$K$ token post-hoc explainer for NLP classifications?

- **(Q4)** Are SAEs generally faithful, and does more capacity in their width improve their faithfulness?

To answer these questions, we select a diverse set of datasets, model architectures on tabular, vision, and NLP classification tasks. See Table 2 for an overview. Our experiments evaluate the faithfulness of various explanation approaches, as detailed below.

**Global, and Local Methods.** For global methods, we include feature visualisation techniques with different regularisation, and optimization procedures: *Deep-Viz* (DV) (Yosinski et al., 2015), *Magnitude Constrained Optimization* (MACO) (Fel et al., 2024), and *Fourier preconditioning* (FO) (Olah et al., 2017). Optimization steps are set to 50, 100, and 250, otherwise, default values are used as provided in the respective publications ((Fel et al., 2024), and (Nguyen, 2020)). For local methods, two variants of *Layer-wise Relevance Propagation* (LRP), the $\varepsilon$-*rule* (LRP-$\varepsilon$) (Bach et al., 2015) with $\varepsilon = 1e^{-6}$, and the $z^+$-*rule* (LRP-$z^+$) (Montavon et al., 2017) are employed. Also, we include several gradient-based approaches such as *Gradient* (GRAD) (Morch et al., 1995; Baehrens et al., 2010), *Saliency* (SAL) (Simonyan et al., 2014), *Input×Gradient* (IXG) (Shrikumar et al., 2016), *GradCAM* (G-CAM) (Selvaraju et al., 2020), *Guided Backpropagation* (GBPG) (Springenberg et al., 2015), *SmoothGrad* (SMG) (Smilkov et al., 2017) with 10 noisy samples, and noise level $0.1/(x_{\max} - x_{\min})$, *Integrated Gradients* (INT-G) (Sundararajan et al., 2017) with 10 iterations, and zero baseline. For NLP tasks, we evaluate *LayerIntegratedGradients* (L-INTG) explanations *w.r.t.* the first embedding layer. Two Shapley-based algorithms (Lundberg & Lee, 2017) are included: *GradientSHAP* (SHAP-G) with 10 samples, and *PartitionShap* (SHAP-P) for NLP tasks.

**LLM-x Methods.** An emerging research area in explainability uses separate LLMs to generate post-hoc attributions for important features of a given model (Bills et al., 2023; Kroeger et al., 2023; Krishna et al., 2023; Amara et al., 2024). We create LLM-x explanations by prompting Gemma-2B-IT (Mesnard et al., 2024) to rank the top-$K$ most important tokens given a textual input, which is then parsed, decoded, and mapped to input tokens, producing binary attribution vectors. LLM prompts describe the model's classification task, and prediction certainty before, and after model perturbation (Section 5.1). The temperature is set to 0 for deterministic outputs. Varying synonyms, the order of tokens, and the number of top-$K$ values to $\{5, 10\}$ contribute to the robustness of our findings. The full explanation methodology is described in Appendix A.4.4 with an illustration in Figure A.2.

**Sparse Autoencoders.** SAEs have lately come forth as an interpretability method for understanding the internal representations of LLMs (Templeton et al., 2024; Huben et al., 2024). In our work, we generate SAE explanations using `Gemma-Scope` (Lieberum et al., 2024), pretrained on the residual block representations of the Gemma-2-2B model. Explanations are saved for all 26 layers at both 16K, and 65K widths. Given the sparsity of the explanation vectors, we use cosine distance to compute explanation distortion, defined as $1 - \frac{\mathbf{u} \cdot \mathbf{v}}{|\mathbf{u}||\mathbf{v}|}$, as it effectively measures similarity regardless of magnitude. Appendix A.4.4 provides a detailed description of the generation process for each SAE explanation.

**Control Variants** We also evaluate the faithfulness of two control variants: a random explanation (RAN) sampled from a uniform distribution, $\hat{\boldsymbol{e}}_i \sim \mathcal{U}(1, 0)$, and a top-$K$ control variant (RAN-$K$) with $K$ non-zero attributions, each equal to 1. Unless specified, all experiments evaluate 250 explanations for the logit of the predicted class. For comparability, global, and local explanations are normalised by dividing the attribution map by the square root of its average second-moment estimate (Equation 21) (Binder et al., 2022), with further explanation preprocessing details provided in Appendix A.4.4. For metric implementation, and meta-evaluation, we use the `Quantus` (Hedström et al., 2023b), and `MetaQuantus` (Hedström et al., 2023a) libraries, respectively. Further experimental details for Q1, Q2, and Q3 are provided in Appendix A.8.1, A.8.4, and A.8.5, respectively.

Table 2: An overview of datasets, and models, with references in Appendix A.4. A semicolon separates models used per dataset.

| Modality | Dataset (n. classes) | Model (size) | Acc. % | Source | Expl. dim | Task |
|---|---|---|---|---|---|---|
| Text | SMS Spam (2) | BERT-TINY FT (4.4M) | 98.0 | HF | 128 | Spam |
| | IMDb (2) | Pythia FT (7.6M); Gemma-2 (2B) | 86.4; 95.6 | HF | 512 | Sentiment |
| | SST-2 (2) | BERT-tiny FT (4.4M) | 98.0 | HF | 59 | Sentiment |
| Vision | ImageNet-1K (1000) | ResNet18 (11.7M) | 89.1 | Torchvision | 50176 | Object |
| | PATH (9) | MedCNN (235.2K) | 84.3 | Local | 784 | Pathology |
| | Derma (7) | MedCNN (234.9K) | 73.2 | Local | 784 | Dermatology |
| | MNIST (10) | LeNet (61.7K) | 97.7 | Local | 784 | Digit |
| | fMNIST (10) | LeNet (61.7K) | 87.7 | Local | 784 | Fashion |
| Tabular | Adult (2) | 3-layer MLP (11.7K); LR (28) | 84.6; 83.3 | OpenXAI | 13 | Income |
| | Compas (2) | 3-layer MLP (11.1K); LR (16) | 85.0; 85.3 | OpenXAI | 7 | Recidivism |
| | Avila (12) | 2-layer MLP (3.5K) | 80.8 | Local | 10 | Letter |

## 6.1 Measuring Empirical Reliability

To investigate the empirical reliability of `GEF` and `Fast-GEF` evaluations compared to singular approaches, we perform meta-evaluation, which is the practice of evaluating the evaluation method itself. To this end, we adopt the meta-evaluation methodology from Hedström et al. (2023a), which bypasses the lack of ground truth labels by focusing on *metric consistency* ("does this evaluation method produce similar results under consistent conditions?"). For this, two practical meta-evaluative tests are performed: the Input Perturbation Test (IPT), and the Model Perturbation Test (MPT). Each test returns a meta-consistency (MC) score (see Equation 20), which ranges between $[0, 1]$. Higher values indicate greater reliability. Full meta-evaluation scoring methodology is provided in Appendix A.3. As a sanity check, we also show in Appendix A.8.3 that our proposed evaluators assign low scores to different random control variants, where other metrics fail to do so.

**Setup.** We benchmark three evaluation methods per criterion. In the robustness category, we include *Relative Input Stability* (RIS), *Relative Representation Stability* (RRS), *Relative Ouput Stability* (ROS) (Agarwal et al., 2022a). In the sensitivity category, we include *Model Parameter Randomisation Test* (MPRT) (Adebayo et al., 2018), *Smooth MPRT* (sMPRT), and *Efficient MPRT* (EMPRT) (Hedström et al., 2024). In the faithfulness category, we include *Faithfulness Correlation* (FC) (Bhatt et al., 2020), *Pixel-Flipping* (PF) (Bach et al., 2015), and *Region-Perturbation* (Samek et al., 2017). All metrics are mathematically described in Appendix A.4.5. To ensure comparability with the original publication (Hedström et al., 2023a), we run meta-evaluation on the same set of tasks, which includes ImageNet (Russakovsky et al., 2015), MNIST (Le-Cun et al., 2010) and fMNIST (Xiao et al., 2017) datasets with architectures such as ResNets (He et al., 2016) and LeNets (LeCun et al., 1998) architectures. Each metric evaluates GRAD, SAL, G-CAM, SHAP-G explanations. Further results, and details are provided in Tables A.1, and A.2, and Appendix A.8.5.

**Results.** Figure 6 (A) shows that our proposed unified methods (`GEF` and `Fast-GEF`) achieve the highest *overall* MC scores, averaged over both MPT, and IPT tests. Our unified methods significantly outperform the most comparable evaluation approach, the faithfulness metrics, which also use $Z$ perturbation steps, with average MC scores of 0.733 compared to 0.601. Although no evaluation method achieves a perfect score (*i.e.,* MC=1), the unified methods still perform comparably to robustness metrics, and surpass sensitivity metrics, with average scores of 0.727, and 0.673, respectively. These results are encouraging as they show that unified methods can achieve high reliability, even when explanation behaviour is evaluated under multiple model conditions, unlike robustness, and sensitivity metrics that focus on a single perspective. While the ROS metric has the highest individual score, this is not statistically significant, and it only offers a limited view of explanation quality. Figure 6 (B) shows that unified metrics excel in MPT, while robustness metrics perform slightly better in IPT. These score differences correspond to robustness metrics using input perturbations, and unified metrics relying on model perturbations. Further details are provided in Appendix A.8.4.

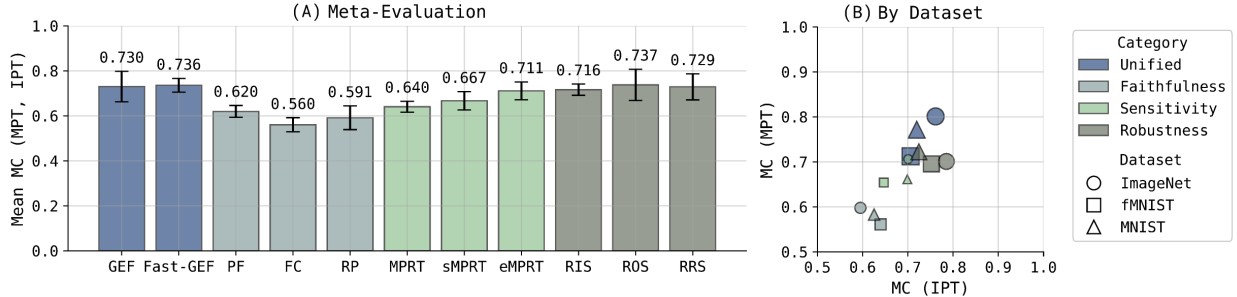

Figure 6: Meta-evaluation, and comparison to established explanation evaluation methods. (A) shows the mean MC scores across MPT, and IPT, aggregated over all datasets, with the error bars showing the standard deviation. (B) displays MC scores aggregated by the test type, and dataset, where the size of the scatter point denotes the standard deviation. `GEF` scores are computed for fMNIST, and MNIST datasets due to computational constraints.

## 6.2 Cross-Evaluating Local, and Global Methods

While local, and global explanations serve distinct purposes, and provide different insights *w.r.t.* their model, it is beneficial to compare them side-by-side in a unified view, as they often rely on similar methodological components, such as network gradients (LeCun et al., 1998; Olah et al., 2017). The absence of general-purpose evaluations has however so far prevented such comparison. `GEF` and `Fast-GEF` effectively fill this gap, facilitating a first, cross-domain comparative faithfulness benchmarking between global, and local methods. Extended results are provided in Appendix A.8.4.

Figures 7 and 8 provide an overview of cross-domain results for tabular, and vision tasks. For all tabular tasks, `GEF` estimates are computed. For vision tasks, due to the high computational cost of global methods, `Fast-GEF` is used to allow for a fair comparison to local methods. As shown in Figure 7, no explanation method is perfectly faithful to its model (*i.e.,* no score equals 1) nor consistently outperforms others across tested tasks. This variation aligns with most benchmarking studies of local linear approximation methods, which rarely identify a single winning method (Hedström et al., 2024; Hesse et al., 2024). Among tested

global feature visualisation methods, MACO generally outperforms FO variants, consistent with Fel et al. (2024). Comparing the faithfulness scores of DV, MACO, and FO reveals that more optimisation steps do not necessarily result in higher explanation faithfulness. All tested methods significantly outperform the random baseline (RAN), which serves as the theoretical lower bound. As expected, RAN produces faithfulness scores centered around zero. In Figure 8 (A), and (B), we observe that RAN explanation distortion quantities are flat, *i.e.,* independent of the model distortion. Tables A.4, and A.5 in Appendix A.8 present the result of Figure 7.

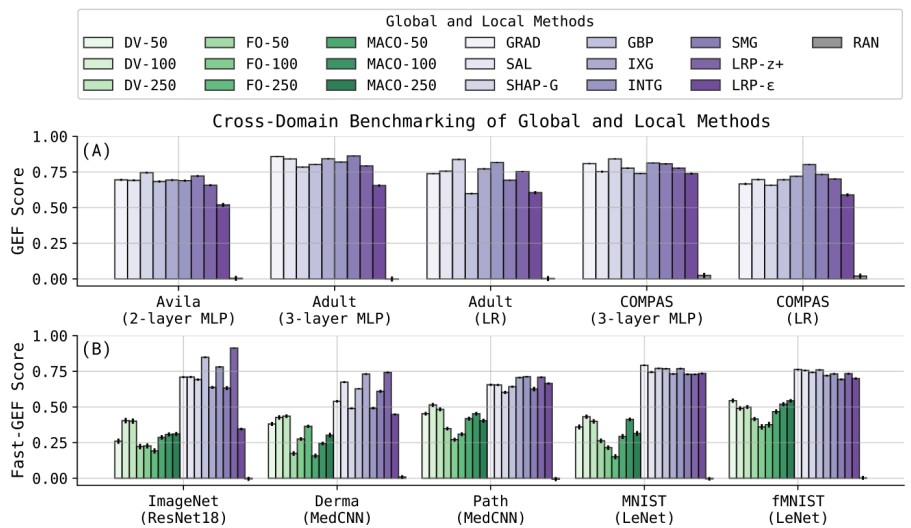

Figure 7: `GEF` and `Fast-GEF` results on (A) local across tabular, and (B) local versus global methods across vision tasks. The error bar shows the standard error, *i.e.,* $\frac{\sigma}{\sqrt{N}}$, where $\sigma$ is the standard deviation, and $N$ is the sample size.

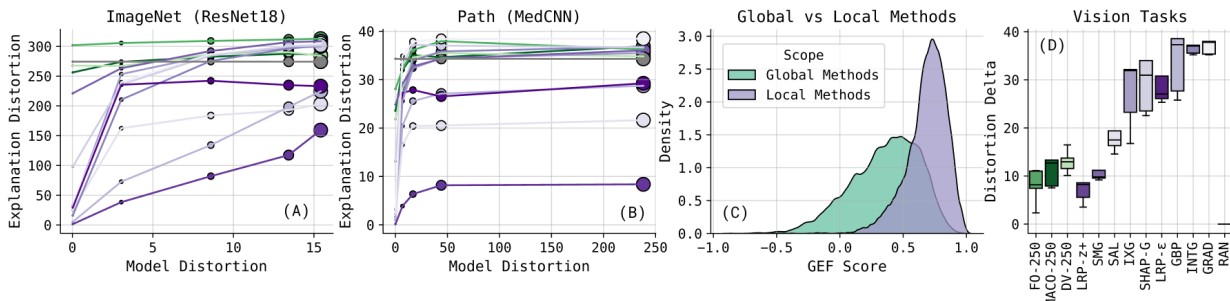

Figure 8: `Fast-GEF` results for vision tasks. (A), and (B) plot the model, and explanation distortion for ImageNet (ResNet18), and Path (MedCNN) along the perturbation path with $Z = 5$ perturbation steps. Here, global methods (DV, MACO, FO) are selected with 250 optimisation steps. (C) displays the distribution of `Fast-GEF` scores for local, and global methods, aggregated over all vision tasks. (D) reports the aggregated difference in explanation distortion between start $z = 1$, and end $z = 5$.

**Local Methods are Moderately Aligned.** Despite local methods showing imperfect, and highly varying scores across tested models, and datasets, most `GEF` estimates in tabular tasks, and `Fast-GEF` estimates in vision exceed 0.5, suggesting that the explanation retains some alignment with its model. This is not surprising given that parameter scaling directly effects the model's curvature, to which local gradient-based methods are highly sensitive (Dombrowski et al., 2019), thereby instantaneously influencing their responsiveness to perturbation.

Figure 7 (A) shows that some local methods produce distortion outputs nearly monotonically related to its model, particularly at lower magnitudes (*i.e.,* a $z \leq 3$). This finding nuances studies by Adebayo et al. (2018), which provide single-point sensitivity estimates, conclusively reporting low reactivity to parameter randomisation in local methods. Corroborating recent rebuttal works (Yona & Greenfeld, 2021; Sundararajan

& Taly, 2018; Binder et al., 2022) that challenges stark claims of method failure (Adebayo et al., 2018), we find that gradient-based methods are moderately faithful.

**Global Methods are Constrained by Regulariser.** Figure 8 (C) shows aggregate `Fast-GEF` scores, indicating that global feature visualisation methods typically are less faithful compared to local linear approximation methods. These differences in faithfulness estimates may be attributed to the global methods' inherent reliance on optimisation procedure (Olah et al., 2017), and NN's ability to retain its learned features despite perturbation via parameter perturbation (Binder et al., 2022). For reference, DV applies multiple regularisation techniques directly to the image, such as Gaussian blur, and cropping regions based on norm, and pixel contribution, while MACO, and FO regularise the frequency domain representation, with MACO adding an extra layer of regularisation via a predefined magnitude template. As observed in Figure 8 (A), and (B), despite model perturbation, explanation distortions stay relatively flat, with lower distortion deltas compared to most local methods, as displayed in Figure 7 (D). A strongly regularised optimisation procedure may inherently limit the faithfulness of global methods, in favour of a maximally activated neuron response.

### 6.3 Evaluating LLMs as Post-hoc Explainers

While researchers have recently begun exploring the potential of using LLMs as post-hoc explainers, there is still limited theoretical understanding, and empirical evidence on the general faithfulness of such approach. Can an LLM which is inherently decoupled from the model it seeks to explain, provide faithful outcomes? In our evaluation, we prompt Gemma-2B-IT for a top-$K$ token explanation for a given input, and prediction pair for datasets characterised by short tokenized lengths, *i.e.,* 59 for SST-2, and 128 for SMS Spam. The post-processed binary explanation vectors are then evaluated with `GEF` and `Fast-GEF`. See Appendix A.8.5 for further details, and extended results.

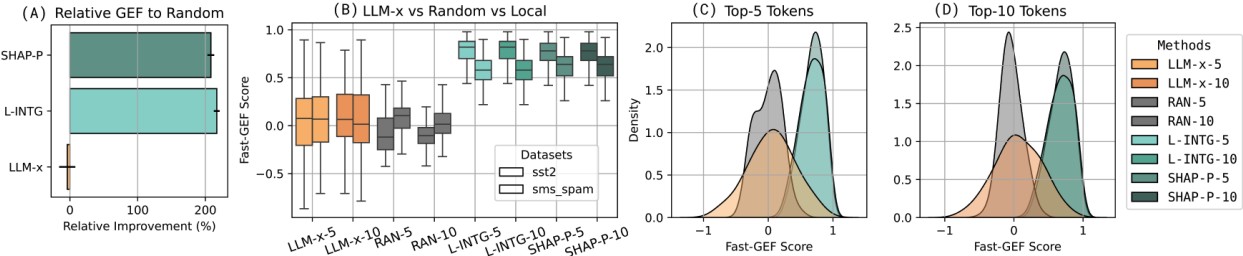

Figure 9: `GEF` (with $M{=}3$), and `Fast-GEF` results on different top-$K$ explanation NLP tasks. (A) shows the percentage improvement in `GEF` scores relative to RAN, aggregated over all tasks, with error bars showing the standard error. (B) shows the results in the form of box plots for the two datasets with SST-2 (*left*), and SMS Spam (*right*). (C), and (D) show the distribution of `Fast-GEF` scores for top-5, and top-10 explanations respectively, both aggregated over all tasks.

**LLM-x Explanations Comparable to Random.** Our `GEF` and `Fast-GEF` results in Figure 9 (A), and (B) show that Gemma-2B-IT as an explainer is (i) significantly less faithful than local methods such as SHAP-P, and L-INTG, and (ii) similarly unfaithful as random explainers RAN-5 or RAN-10, on both SST-2, and SMS Spam classification tasks. Figure 9 (C), and (D) demonstrate that these findings generalise over both top-5, and top-10 tokens tasks, aggregated over both datasets. Our results, showing that LLM-x explanations are not more faithful than random, differ from the encouraging results reported by Kroeger et al. (2023), who found `GPT-4` to be as faithful as local methods in identifying top-$K$ tokens for tabular tasks. This divergence may naturally stem from variations in the experimental setup, including the specific explanation task, LLM used, methodology to evaluate faithfulness, and prompting strategies, however, it also underscores that the faithfulness of LLM-x is still an open research question. To fully understand the potential of LLMs as explainers, further research with additional LLMs would be beneficial.

### 6.4 Measuring Faithfulness of Sparse Autoencoders

SAEs are gaining attention for their claimed ability to construct interpretable "monosemantic" features of a given layer of an LLM. Yet, their general faithfulness remains underexplored (Makelov et al., 2024; Mallen

& Belrose, 2024). To this end, we evaluate SAE explanations for Gemma-2-2B model on $N = 250$ samples on the IMDb dataset (Maas et al., 2011). Here, the Gemma-2-2B model is repurposed as a binary classifier by extracting the logits of the "positive" or "negative" classes at the final token position of the prompt. Appendix A.4.4 provides more details.

**High Faithfulness Independent of SAE Width.** Figure 10 (A) demonstrate that the SAE explanations generally are faithful *w.r.t.* the model's intermediate representations. `Fast-GEF` scores are consistently above 0.75 except for fluctuations in layers 1, and $16 - 19$. No significant difference is observed between 16K, and 65K widths, suggesting that the width of the encoding, *i.e.,* the capacity of the SAE, does not correlate with explanation faithfulness. Moreover, Figure 10 (B) shows that although sparsity of the latent activations decreases in later layers, it does not influence its faithfulness (*i.e.,* $\rho = 0.023$, computed with Spearman Rank correlation). This suggests that a higher activation in SAE latents does not necessarily relate to its measured faithfulness. This raises the question of whether the faithfulness of SAE explanations is inherently scalable within different statistical or qualitative contexts where SAEs are studied. Figure 10 (C)-(G) further illustrates how explanation distortions vary across layers, with values (*y-axis*) increasing in the middle layers.

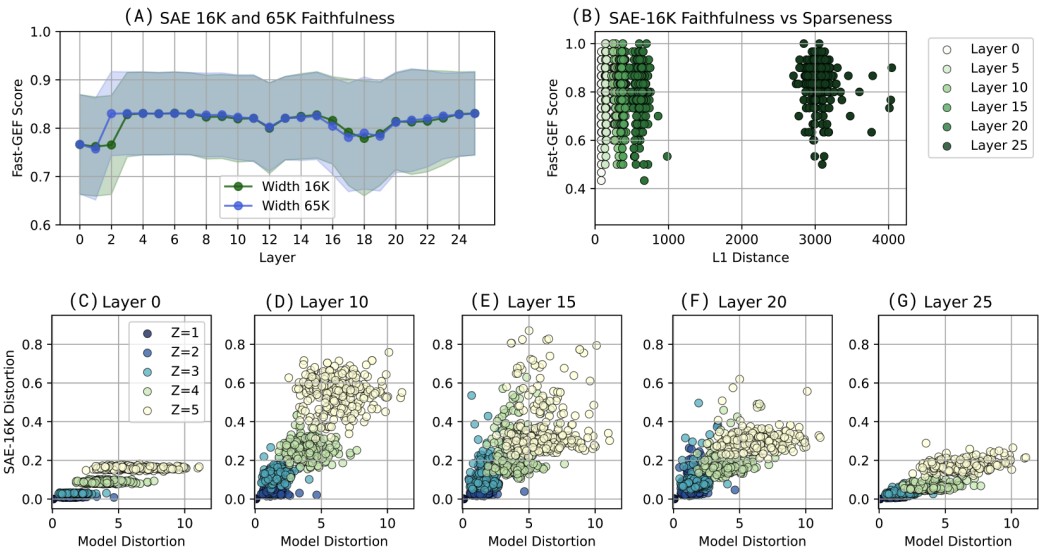

Figure 10: `Fast-GEF` results for SAE explanations on IMDb dataset, using $Z = 5$ perturbation levels, and $M = 3$ models. (A) shows `Fast-GEF` faithfulness scores across layers. (B) shows the sparsity as defined by L1 distance against the `Fast-GEF` indicating no relationship. (C)-(G) illustrates how SAE distortions develop across model layers, coloured by perturbation level.

# 7   Discussion: Where Are We Going?

With the evolving landscape of interpretability, redefining both the conceptual framework, and the geometric foundations of explanation faithfulness is important. Our work puts forward a long-overdue unification of robustness, sensitivity, and faithfulness evaluations, providing a novel, and urgently needed, revised approach (Definition 5) to evaluate the direct alignment between explanation, and model functions (Section 3). In this work, we address the fundamental flaws of many existing evaluations: a systematic overlook of the intrinsic geometry of non-linear spaces (Section 4). Our solution offers a threshold-free, fair comparison of functional distortions, making our approach not just another evaluation method but a necessary foundation for future interpretability research (Section 5).

**Novel, Empirical Insights.** In a first-ever cross-domain faithfulness benchmarking of global, and local explanations on vision, tabular, and NLP tasks (Section 6), we learn that tested local explanation methods generally are moderately faithful. We find that global feature visualisation methods are comparatively less faithful, which is an important understanding considering the recent evidence pointing to their general

susceptibility to adversarial manipulation (Geirhos et al., 2023; Bareeva et al., 2024a). While it would be valuable to compare our findings with existing studies, to our knowledge, there is no direct study on the faithfulness of feature visualisations. Existing evaluations focus on the alignment with human preferences or improvement on a downstream task (Borowski et al., 2021; Zimmermann et al., 2021; Krishna et al., 2023; Bareeva et al., 2024b) or similarity to natural samples of the explained class (Fel et al., 2024). Our findings on generalised faithfulness thus provide complementary insights into the quality of feature visualisation as *model* explainers.

Additionally, due to the recent interest LLMs as potential post-hoc explainers (Krishna et al., 2023; Kroeger et al., 2023), we study their faithfulness. We find no improved faithfulness compared to random explanations, and encourage more investigation on this question. Finally, we observe that residual stream SAEs on Gemma-2-2B exhibit generally high faithfulness, with the width having limited influence (*i.e.,* 16K or 65K). Further investigation is required to fully understand the potential of SAEs, and LLM-x as *generally faithful* explainers.

## 7.1 Limitations

While the results in our paper allow us to claim that our proposed method is more sound geometrically (Section 4), more reliable empirically (Section 6.1), and easier to use practically (Section 5), our evaluation alone does not imply that the explanation quality is sufficient. Without ground truth labels, we cannot assess the statistical validity of an explanation function. An explanation may be estimated to be *generally faithful* but still lack intrinsic value (Bhattacharjee & von Luxburg, 2024) or interpretable qualities (Bordt & von Luxburg, 2024). The need for a thorough, application-grounded assessment of explanation quality that asserts value on a downstream task (Krishna et al., 2023; Lanham et al., 2023) is not eliminated when using `GEF`. Evaluation using synthetic models with known ground truth (Carmichael & Scheirer, 2023) could complement our proposal.

## 7.2 Future Work

There are several exciting geometric, and empirical questions worth exploring. The geometric considerations in `GEF` suggest a deeper examination of the computational trade-offs of computing accurate pullbacks on individual explanation functions, specifically in comparing global versus local methods. In future work, there is opportunity to build on the growing body of research in ML that draws from geometry, and related topics in higher mathematics to deepen our understanding of NNs, and problems to which they are applied (Stephenson et al., 2021; Burns & Tang, 2023; Papamarkou et al., 2024). Recent theoretical studies on LLMs, and transformer models (Hoogland et al., 2024; Burns, 2024) have illustrated how neural activations may arrive at, and utilise "superpositional" encoding strategies (Elhage et al., 2022), which prominently feature considerations or findings of a geometric or topological nature. Continued development of general frameworks, and theories that conceptualise NNs in terms of geometry, and topology (Bianchini & Scarselli, 2014; Hauser & Ray, 2017; Naitzat et al., 2020; Benfenati & Marta, 2023a;b; Burns & Fukai, 2023) will likely facilitate a deeper understanding of both explanations, and evaluations, particularly in relation to the underlying mathematical characteristics of data, optimisation processes, and learned functions.

Recent advances in manifold geometry have introduced tools to analyse how input data modulates internal processing through perturbations (Kvinge et al., 2023). Exploring how explanation faithfulness varies with training data, and how it intersects with the geometric characteristics of the model presents an exciting direction. We also expect models optimised with non-Euclidean methods (Fei et al., 2023) to reveal stronger differences between `GEF` and `Fast-GEF`, providing new opportunities to study the interplay between geometry, and faithfulness in explainability.

Lastly, we plan to expand our benchmarking scope to include natural activation-maximisation explanations (Borowski et al., 2021), concept-based explanations like INVERT (Bykov et al., 2023), and non-classification tasks. Given that pullback calculations can be computationally prohibitive for high-dimensional explanations, and highly parameterised models, exploring ways to speed up the Jacobian calculation (Equation 19), and employ adaptive noise schedules would be valuable.

**Broader Impact Statement**

Interpretability, or XAI, is widely acknowledged as essential for responsible ML. This paper critically examines current evaluation methods from unifying, and geometric perspectives, and proposes improvements. While negative societal impacts are improbable, overreliance on any single evaluation method is not advised.

**Acknowledgments**

This work was partly funded by the German Ministry for Education and Research (BMBF) through the project Explaining 4.0 (ref. 01IS200551). Additionally, this work was supported by the European Union's Horizon Europe research and innovation programme (EU Horizon Europe) as grant TEMA (101093003); the European Union's Horizon 2020 research and innovation programme (EU Horizon 2020) as grant iToBoS (965221); the German Research Foundation (DFG) as research unit KI-FOR 5363 (project ID: 459422098); the state of Berlin within the innovation support programme ProFIT (IBB) as grant BerDiBa (10174498); and BIFOLD (refs. 01IS18025A, 01IS18037A); T.B. thanks Vasiliki Liontou for helpful discussions.

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

## Appendix

The Appendix is organised as follows theoretical considerations (Section A.1), implementation notes for `GEF` and `Fast-GEF` (Section A.2), details on the `MetaQuantus` framework (Section A.3), the general experimental setup (Section A.4), analysis of model assumptions (Section A.5), alignment patterns extended results (Section A.6), ablation experimental results (Section A.7), extended results from individual experiments (Section A.8), and notation tables (Section A.9).

### A.1 Theoretical Considerations

The following subsections provide detailed proofs, extensions, and discussions surrounding the GEF criterion.

#### A.1.1 GEF: Penalising Random Explanations

Following the discussion in Section 3, a quality estimator should be able to identify unfaithful explanations. In the following, we show that our proposed GEF criterion (Definition 5) recognise two specific types of unfaithful explanations: constant, and random explanations, which is independent of its model, by design.

**Corollary 1 (Penalising Unfaithful Explanations)** *Let $\Psi^{GEF}$ be a quality estimator that yields estimates $q^{GEF} \in \mathbb{R}$, with $q^{GEF} = 0$ indicating a lack of generalised faithfulness. To be a valid measure of explanation quality, $\Psi^{GEF}$ should assign low scores to both (I) constant, and (II) random explanations*

$$Constant \ (I): \forall \hat{e} : e = \hat{e} \Rightarrow q^{GEF} = 0$$

$$Random \ (II): \forall \hat{e} : \hat{e} \sim \mathcal{U}(0,1) \Rightarrow q^{GEF} = 0$$

*where $\hat{e}$, and $e$ are perturbed, and unperturbed explanations, and $\mathcal{U}(0,1)$ denotes a uniform distribution. The GEF estimate $q^{GEF} = \rho(\boldsymbol{d}_f, \boldsymbol{d}_\phi)$ (Definition 5) assigns low scores in the first, and the second case.*

**Proof.** In case (I), the explanation does not change across perturbations, leading to an explanation distortion vector $\boldsymbol{d}_\phi$ that contains only zeros

$$\forall \hat{e}, z \in [1, Z] : e = \hat{e} \Rightarrow \ \boldsymbol{D}_\phi^z = 0,$$

whereas the model's distortion vector $\boldsymbol{d}_f$ will contain non-zero values due to perturbations

$$\forall z \in [1, Z] : \boldsymbol{D}_f^z \neq 0.$$

Consequently, the correlation coefficient $\rho(\boldsymbol{d}_f, \boldsymbol{d}_\phi)$ will be zero with $q^{GEF} = 0$.

In case (II), the explanation distortion $\boldsymbol{D}_\phi^z$ will be approximately uniform across all perturbation steps since each perturbation is independently drawn from the same distribution:

$$\forall \hat{e}, z, j \in [1, Z], z \neq j : \hat{e} \sim \mathcal{U}(0,1) \Rightarrow \boldsymbol{D}_\phi^z \approx \boldsymbol{D}_\phi^j,$$

whereas the model distortion $\boldsymbol{D}_f^z$ will vary according to the degree of the perturbation

$$\forall \hat{e}, z, j \in [1, Z], z \geq j : \hat{e} \sim \mathcal{U}(0,1) \Rightarrow \boldsymbol{D}_f^j \geq \boldsymbol{D}_\phi^z.$$

The lack of correlation between $\boldsymbol{d}_f$, and $\boldsymbol{d}_\phi$ results in a quality measure $q^{GEF}$ that is equal to zero. This completes the proof.

#### A.1.2 GEF: Extension

To extend the applicability of GEF (Definition 5) to global methods that explain *any* neuron within a model, we adopt Kopf et al. (2024), and view the model $f$ as a composition of two functions, $F : \mathcal{X} \to \mathcal{G}$, and $L : \mathcal{G} \to \mathcal{Y}$, such that $f = L \circ F$. Here $\mathcal{G} \subset \mathbb{R}^{c \times w^* \times h^*}$, where $c \in \mathbb{N}$ is the number of neurons in the layer, and $w^*, h^* \in \mathbb{N}$ represent the width, and height of the feature map, respectively. The function $F$, is referred to as the *feature extractor*. We redefine the model function as a chosen feature extractor, and replace $y$ in Definition 5 with the activation of the $c^{th}$ neuron such that *i.e.,* $y = F_c(\boldsymbol{x}, \theta) : \mathcal{X} \to \mathbb{R}^{w^* \times h^*}$. While the model's output space $\mathcal{Y}$ is replaced by $\mathcal{G}$, we similarly define the perturbed instance $\hat{y}$.

### A.1.3  GEF: Derivation of Linear Case

Our definition of GEF is based on the observation that any distortion present in the model output space $\mathcal{Y}$, should be mirrored in the explanation space $\mathcal{E}$. Since neural networks are non-linear functions, a fair distortion in $\mathcal{Y}$, and $\mathcal{E}$, requires the introduction of the pullback (Section 4).

In the case of a linear model, however, the relationship between the distortion quantities $\boldsymbol{D}_f$, and $\boldsymbol{D}_\phi$ can be derived analytically. Here, the explanation is based on the first-order Taylor term, which is a linear approximation of the model's behaviour, forming the foundation of many established explanation methods (*e.g.,* Montavon et al. (2017)). We proceed to derive this relationship explicitly below.

**Proof.** Consider $f$ to be a linear model of the form $f(\boldsymbol{x};\theta) = \theta\boldsymbol{x} + c$. The explanation is the parameter vector $\theta$. We can derive the expected distortion $\boldsymbol{D}_f := \mathbb{E}_{\hat{\theta}_m}[(f(\boldsymbol{x};\theta) - f(\boldsymbol{x};\hat{\theta}_m))^2]$ (see Equation 6) where $m \in [1, M]$ denotes the number of perturbed models for a fixed perturbation magnitude $\xi$, *i.e.,* a step $z$.

$$\boldsymbol{D}_f^z = (\theta\boldsymbol{x} + c)^2 - 2(\theta\boldsymbol{x} + c)\mathbb{E}_{\hat{\theta}_m}\left[(\hat{\theta}_m\boldsymbol{x} + c)\right] + \mathbb{E}_{\hat{\theta}_m}\left[(\hat{\theta}_m\boldsymbol{x} + c)^2\right]$$

$$\boldsymbol{D}_f^z = \theta^2\boldsymbol{x}^2 + 2c\theta\boldsymbol{x} + 2c^2 - 2\boldsymbol{x}(\theta\boldsymbol{x} + c)\mathbb{E}_{\hat{\theta}_m}\left[\hat{\theta}_m\right] - 2c(\theta\boldsymbol{x} + c) + \mathbb{E}_{\hat{\theta}_m}\left[\hat{\theta}_m^2\boldsymbol{x}^2 + 2c\hat{\theta}_m\boldsymbol{x}\right]$$

$$\boldsymbol{D}_f^z = \theta^2\boldsymbol{x}^2 - 2\boldsymbol{x}(\theta\boldsymbol{x} + c)\mathbb{E}_{\hat{\theta}_m}\left[\hat{\theta}_m\right] + 2c\boldsymbol{x}\mathbb{E}_{\hat{\theta}_m}\left[\hat{\theta}_m\right] + \boldsymbol{x}^2\mathbb{E}_{\hat{\theta}_m}\left[\hat{\theta}_m^2\right],$$

$$\boldsymbol{D}_f^z = \theta^2\boldsymbol{x}^2 - 2\theta\boldsymbol{x}^2\mathbb{E}_{\hat{\theta}_m}\left[\hat{\theta}_m\right] + \boldsymbol{x}^2\mathbb{E}_{\hat{\theta}_m}\left[\hat{\theta}_m^2\right]. \tag{14}$$

For the explanation distortion, a similar decomposition can be performed

$$\boldsymbol{D}_\phi^z = \theta^2 - 2\theta\mathbb{E}_{\hat{\theta}_m}\left[\hat{\theta}_m\right] + \mathbb{E}_{\hat{\theta}_m}\left[\hat{\theta}_m^2\right]. \tag{15}$$

By combining Equation 14 and 15, we arrive at

$$\boldsymbol{D}_\phi^z = \frac{1}{\boldsymbol{x}^2}\boldsymbol{D}_f^z, \tag{16}$$

We can construct the distortion vectors $\mathbf{d}_\phi$, and $\mathbf{d}_f$, and for each entry Equation 16 holds. When $\rho$ is defined as the Pearson correlation coefficient, we find the distortion of the model $\mathbf{d}_f$, and the distortion of the explanation function $\mathbf{d}_\phi$ to be perfectly correlated

$$\rho(\mathbf{d}_\phi, \mathbf{d}_f) = \frac{\mathrm{cov}_\xi(\mathbf{d}_f, \mathbf{d}_\phi)}{\sqrt{\mathrm{Var}_\xi(\mathbf{d}_f)}\sqrt{\mathrm{Var}_\xi(\mathbf{d}_\phi)}},$$

which is equal to

$$\rho(\mathbf{d}_\phi, \mathbf{d}_f) = \frac{1/\boldsymbol{x}^2\mathrm{Var}_\xi(\mathbf{d}_f)}{1/\boldsymbol{x}^2\mathrm{Var}_\xi(\mathbf{d}_f)} = 1. \tag{17}$$

This proves that in a simplified scenario, the key assumption of correlated distortion quantities holds, *i.e.,* the model parameters $\theta$ provide a *perfectly faithful* explanation. Since monotonicity is a weaker condition than linearity, Equation 17 also holds when $\rho$ is defined as the *Spearman Rank correlation* coefficient.

### A.1.4  GEF: Influence of Z

The parameter $Z$ represents the number of steps in the perturbation path, and consequently dictates how finely the model's response will be captured by the GEF criterion (Definition 5). As such, selecting an appropriate value for $Z$ is critical because it affects the interpretation of the results. A higher $Z$ allows for a finer evaluation of how well an explanation aligns with the model's behaviour under varying conditions. When using Spearman's rank correlation coefficient as our measure of $\rho$, a larger $Z$ generally stabilises the faithfulness score due to the reduction in confidence intervals with more samples (*i.e.,* $CI \sim \frac{1}{Z}$) (Bonett &

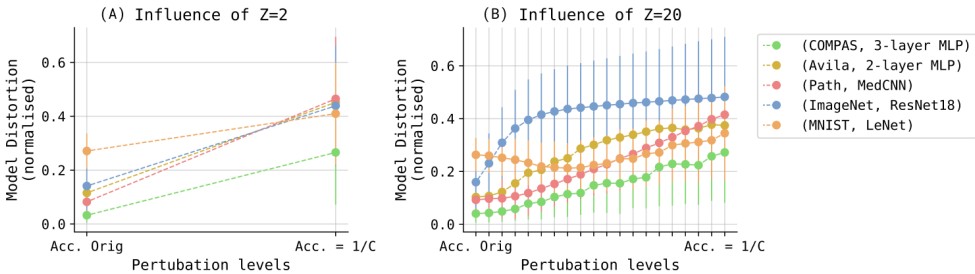

Figure A.1: Model distortion (normalised by its maximum value) with (A) showing $Z = 2$ perturbation steps, and (B) showing $Z = 20$ perturbation steps.

Wright, 2000). Nonetheless, this assumes a monotonic response from both the model, and the explanation, which may not be realistic (Section 4). If the model itself is not monotonic across perturbations, expecting the explanation to behave monotonically is also unrealistic.

Figure A.1 (A) ($Z = 2$), and (B) ($Z = 20$) demonstrate the violation of the monotonicity assumption, as we observe large error bars, and divergent behaviour for $Z = 20$, indicating non-monotonic responses. Accordingly, a moderate value of $Z$ (Zar, 2005) is advised for meaningful measurement.

### A.2 Notes on `GEF` and `Fast-GEF` Implementation

In the following, we provide details on the `GEF` algorithm.

### A.2.1 Generate Perturbation Path

To generate the perturbation path of length $Z$, satisfying $y \neq \hat{y}$, $\forall y, \hat{y} \in \mathcal{Y}$, we computationally find the minimum noise level $\sigma_z^2$ at $z = Z$, such that the perturbed model's accuracy (ACC) approximates $\frac{1}{C}$, where $C$ is the number of classes, within a threshold, $i.e.$, $\epsilon \ll 1$. Here, $\text{ACC} = \frac{1}{N} \sum_{i=1}^{N} (f(\boldsymbol{X}_i; \theta) = \boldsymbol{Y}_i)$ where $N$ is the number of samples in the test set, denoted $\boldsymbol{X}$. This is achieved by progressively increasing $\sigma^2$, and applying it to the model according to Section 5.1, which process concludes when the model's accuracy satisfies the condition $|\text{ACC} - \frac{1}{C}| < \epsilon$, thereby determining perturbation level for subsequent evaluation.

**Compute Path Length.** For a more faithful estimate of explanation distortion, for each step $z \in [0, Z]$, we compute $\boldsymbol{D}_\phi$, defined as the path length $L_{(\gamma)}$. We replace the integral in Equation 13 with a sum over $T$ steps:

$$L(\gamma) = \sum_{t=1}^{T} de_t^T (J_f(\hat{\boldsymbol{e}}_t)^T J_f(\hat{\boldsymbol{e}}_t)) de_t, \tag{18}$$

where $de_t \in \mathbb{R}^V$ denotes the feature-wise difference in explanations $i.e.$, $(\boldsymbol{e} - \hat{\boldsymbol{e}}_t)$ with $\phi(f_{\hat{\theta}_t}, \ldots) = \hat{\boldsymbol{e}}_t$, and $J_f(\hat{\boldsymbol{e}}_t) \in \mathbb{R}^{V \times C}$ is the Jacobian for fixed $\boldsymbol{x}$, and $f_{\hat{\theta}_t}$. To numerically approximate this Jacobian, for each step $t \in [0, T]$, we perturb the neural activations ($i.e.$, logits $\hat{\boldsymbol{y}}$) by adding infinitesimal noise. In practice, we sample from a Gaussian distribution $v_k \sim \mathcal{N}(0, 0.001)$ such that $\hat{y}_k = \hat{\boldsymbol{y}} + v_k$, $k \in [1, K]$ times. After each perturbation, we recalculate the corresponding explanation $\phi(\hat{y}_k, \ldots) = \hat{\boldsymbol{e}}_k$. Elements of the Jacobian $J_f(\hat{\boldsymbol{e}}_t)$ are then computed as feature-wise difference between $\boldsymbol{e}$, and $\hat{\boldsymbol{e}}_k$:

$$\frac{\partial e_i}{\partial f_j} \approx \lim_{K \to \infty} \frac{1}{K} \sum_{k=1}^{K} (e_j - \hat{e}_{j,k}) \nu_k^{-1}. \tag{19}$$

where $i, j$ refers to the indices of the Jacobian $J_f(\hat{\boldsymbol{e}}_t)$.

Unless specified otherwise, we set $M$, $Z$, $T$, and $K$ to 5 in all experiments. Please find Appendix A.7 for an ablation study motivating these hyperparameters.

### A.3 Notes on `MetaQuantus` framework

For meta-evaluation, a two-step process is employed. First, two types of controlled perturbations are introduced: minor, and disruptive. These are designed to evaluate the metric's resilience to noise ($NR$), and its sensitivity to adversarial conditions ($AR$), respectively. Specifically, these perturbations are applied in both the input, and model spaces, resulting in two distinct tests: the Input Perturbation Test (IPT), and the Model Perturbation Test (MPT)[4]. Second, the effects of the perturbations are measured in two meta-evaluative criteria: intra-consistency (**IAC**), and inter-consistency (**IEC**). Here, **IAC** refers to measuring the similarity in score distributions post-perturbation, and **IEC** refers to the occurrence of categorical ranking changes within a set of distinct explanation methods[5]. Each metric is then assigned a summarised meta-consistency score, denoted as MC $\in [0, 1]$:

$$
\text{MC} = \left( \frac{1}{|\mathbf{m}^*|} \right) \mathbf{m}^{*T} \mathbf{m} \quad \text{where} \quad \mathbf{m} = \begin{bmatrix} \mathbf{IAC}_{NR} \\ \mathbf{IAC}_{AR} \\ \mathbf{IEC}_{NR} \\ \mathbf{IEC}_{AR} \end{bmatrix}, \tag{20}
$$

with $\boldsymbol{m}^* \in \mathbb{R}^4$ representing an ideal quality estimator, essentially a vector of ones. A higher MC score, approaching 1, indicates superior reliability according to the defined evaluation criteria. Metrics that demonstrate both resilience to minor perturbations, and reactivity to disruptive changes achieve higher MC scores. We refer to the original publication (Hedström et al., 2023a) for further details on the elements in the meta-evaluation vector **m** (Equation 20), and the framework in general.

### A.4 General Experimental Setup

Here, we describe the models, datasets, tooling, hardware, explanation, and evaluation methods in this work.

#### A.4.1 Models, and Datasets

We employ various models for vision, text, and tabular tasks in our experiments. See Table 2.

- For vision classification, we use ImageNet-1K for object recognition (Russakovsky et al., 2015) with ResNet18 (He et al., 2016); Pathology, and Derma for medical image analysis with proposed Med-CNN architecture (Yang et al., 2023); and MNIST (LeCun et al., 2010), and fMNIST, (Xiao et al., 2017) for digit, and fashion recognition with LeNet (LeCun et al., 1998).

- For text classification, we use SMS Spam (Almeida et al., 2011) with a tiny, fine-tuned BERT model (Romero, 2024); IMDb (Maas et al., 2011) with Pythia (AlignmentResearch, 2024); and SST-2 (Socher et al., 2013) with a tiny, fine-tuned BERT model (VityaVitalich, 2023).

- For tabular classification, we use Adult (Becker & Kohavi, 1996) and, COMPAS (ProPublica, 2016), with 3-layer MLP; and Avila (Stefano et al., 2018) with 2-layer MLP.

All models that are not publicly accessible are released at GitHub repository at `https://github.com/annahedstroem/GEF`.

#### A.4.2 Tooling

Several libraries, and open-source implementations enabled this work, including `transformers` (Wolf et al., 2020), `OpenXAI` (Agarwal et al., 2022b), `Captum` (Kokhlikyan et al., 2020), `Zennit` (Anders et al., 2021),

---

[4]For the IPT, independent, and identically distributed (i.i.d.) additive uniform noise is applied, defined as $\hat{x}_i = x + \nu_i$, where $\nu i \sim \mathcal{U}(\alpha, \beta)$. For the MPT, multiplicative Gaussian noise is applied to all network weights, represented as $\hat{\theta} i = \theta \cdot \nu i$ with $\nu i \sim \mathcal{N}(\mu, \sigma^2)$. The hyperparameters $\alpha, \beta, \mu, \sigma^2 a$ follow the specifications of the original study (Hedström et al., 2023a).

[5]**IAC** provides a normalised p-value derived from the non-parametric *Wilcoxon signed-rank test* (Wilcoxon, 1945), comparing the original, and perturbed score distributions. For $NR$, similar distributions are expected, whereas for $AR$, the distributions are anticipated to differ. **IEC** counts ranking changes within explanation methods post-perturbation, with an ideal metric showing consistent rankings under minor noise ($NR$), and altered rankings under disruptive noise ($AR$).

`Shap` (Lundberg & Lee, 2017), `Activation-Maximization` (Nguyen, 2020), and `Horama` (Fel et al., 2024). For metric implementation, and meta-evaluation, we use the `Quantus` (Hedström et al., 2023b), and `MetaQuantus` (Hedström et al., 2023a) libraries, respectively.

### A.4.3 Hardware

The experiments were conducted using two hardware configurations: a cluster with four Tesla V100S-PCIE-32GB GPUs, each offering 32 GB of memory, and a DGX-2 system featuring eight NVIDIA A100-SXM4-40GB GPUs, each with 40 GB of memory. Both setups support the NVIDIA driver version 535.161.07, and CUDA 12.2.

### A.4.4 Explanation Methods

All the hyperparameters of the individual explanations methods, are listed in the main manuscript. Concerning the preprocessing, the signs of the attributions are maintained, unless the method algorithmically relies on it such as SAL. Note, that not every explanation method is suitable or intended to be used for all data modalities, and/ or model architectures. For example, GradCAM explanations are primarily designed for convolutional neural networks (CNN) models, and global feature visualisation methods are generally applied to vision tasks. We only report `GEF` and `Fast-GEF` results where appropriate.

**Normalisation.** We perform normalisation using the square root of the mean of the squared values (as detailed in the Appendix of (Binder et al., 2022)). This approach introduces less variance compared to normalisation techniques like scaling by the maximum value. It is defined as follows

$$\text{norm}(\boldsymbol{e}) = \frac{\boldsymbol{e}_{h,w}}{\left(\frac{1}{HW}\sum_{h',w'}\boldsymbol{e}_{h',w'}^2\right)^{1/2}} \ , \tag{21}$$

where $H$, and $W$ represent the height, and width, respectively, and $\hat{\boldsymbol{e}}_{h,w}$ denotes the explanation value at the pixel location $(h, w)$[6].

**LLM-x Methodology.** In the following, we describe the methodology used to produce LLM-x explanations. An illustration is provided in Figure A.2.

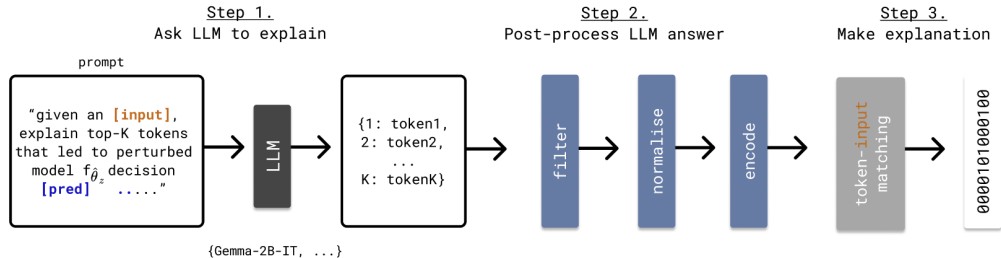

Figure A.2: A high-level overview of the three-step LLM-x methodology.

To generate LLM-x explanations, we use Gemma-2B-IT (Mesnard et al., 2024) as the explainer. For each instance, we create a prompt describing the task, softmax confidence before, and after perturbation, and the class labels. The prompt template introduces the task (*e.g.,* classifying `sms messages` or `sentiment analysis`), and uses synonyms for model descriptions (*e.g.,* `"AI"`, `"machine learning"`), and perturbation types (*e.g.,* `"adversarially manipulated"`, `perturbed with noise"`) to vary language. The softmax change is calculated, and added to the template and is described in the context of the model getting `"more"`

---

[6]This normalisation method ensures that the mean squared distance from zero of each explanation score equals one. Unlike other normalisation techniques that constrain attribution values to a predefined range—making them suitable for visualisation—this method retains a metric useful for comparing the distances across different explanation methods.

or `"less"` certain of a class label. The LLM is asked to return the top-$K$ important tokens in a structured JSON format, ranking tokens from `1` to `K`. The temperature is set to 0 for deterministic outputs.

After prompting, invalid or non-JSON outputs are removed. The LLM-ranked tokens are normalised by lowercasing, removing punctuation. Then, these tokens (or words) are encoded with the original model's tokeniser. Binary explanation vectors are created by matching the LLM-ranked tokens to the original input tokens, with a value of 1 for matching tokens, and 0 otherwise.

For full details, including the code, and prompt template, we refer to our GitHub repository at `https://github.com/annahedstroem/GEF`.

**SAE Methodology.** SAEs are designed to create sparse, interpretable representations of the internal activations of a LLM, while preserving their reconstruction. The activations of a given layer $l$, *i.e.,* $f_\theta(\boldsymbol{x})$ are encoded into a sparse latent vector $z(\boldsymbol{x})$, with latent dimensions larger than the internal representation,. Then, these decoded representations are reconstructed such that $\boldsymbol{g}(z) \approx f_\theta(\boldsymbol{x})$. This process is defined by the encoder and decoder functions

$$\boldsymbol{z}(f_\theta(\boldsymbol{x})) := \sigma\left(\boldsymbol{W}_{enc}f_\theta(\boldsymbol{x}) + \boldsymbol{b}_{enc}\right), \tag{22}$$

$$\boldsymbol{g}(z) := \boldsymbol{W}_{dec}\boldsymbol{z} + \boldsymbol{b}_{dec}, \tag{23}$$

where $\sigma$ enforces sparsity through activation functions like ReLU or JumpReLU (Lieberum et al., 2024), using $L1$ or $L0$ regularisation during training. The SAE explanations are generated by performing a forward pass through the SAE encodings, and storing the activated values of $\boldsymbol{z}$.

### A.4.5 Evaluation Methods

Next, we mathematically define the evaluation methods (or "metrics") used in this work (Section 6.1).

**Faithfulness.** Within the faithfulness category, we evaluate three metrics, including, *Faithfulness Correlation* (FC) (Bhatt et al., 2020), *Pixel-Flipping* (PF) (Bach et al., 2015), and *Region-Perturbation* (RP) (Samek et al., 2017). FC is defined as follows

$$\Psi_{\mathrm{FC}} = \operatorname*{corr}_{S \in |S| \subseteq d}\left(\sum_{i \in S}\phi(\boldsymbol{x}, f, \hat{y}; \lambda)_i, f(\boldsymbol{x}) - f\left(\boldsymbol{x}_{[\boldsymbol{x}_s=\overline{\boldsymbol{x}}_s]}\right)\right), \tag{24}$$

where $|S| \subseteq D$ is a subset of indices of a sample $\boldsymbol{x}$, $\overline{\boldsymbol{x}}$ is the chosen baseline value, and $\boldsymbol{x}_{[\boldsymbol{x}_s=\overline{\boldsymbol{x}}_s]}$ are the masked input, with randomly chosen indices.

PF returns a vector of prediction scores $p_i$ corresponding to pixel replacements $i \in n$, which are sorted in descending order by the highest relevant pixel in the explanation $\phi(\boldsymbol{x}, f, \hat{y}; \lambda)$. To return one evaluation score per input sample, we calculate the area under the curve (AUC) as follows

$$\Psi_{\mathrm{PF}} = \sum_{i=1}^{n}(\hat{y}_i + \hat{y}_{i+1}) \cdot \frac{p_{i+1} - p_i}{2} \tag{25}$$

where $p_i$, and $p_{i+1}$ are the prediction values of the $i^{th}$, and $(i+1)^{th}$ perturbation step, and $\hat{y}_i$, and $\hat{y}_{i+1}$ the corresponding network prediction.

RP follows the most-relevant-first perturbation strategy, creating consecutive perturbed samples $\hat{y}_i, \hat{y}_{i+1}$ such that for $\hat{y}_i$ perturbed pixels correspond to larger respective explanation values than the pixel perturbed in $\hat{y}_{i+1}$. Across each perturbation curve, the area over the curve (AOC) is calculated, and averaged across multiple masked inputs $\hat{\boldsymbol{x}}$ as follows

$$\Psi_{\mathrm{RP}} = \frac{1}{L+1}\mathbb{E}_{(\hat{\boldsymbol{x}})}\left(\sum_{k=1}^{L}(\hat{y}_0 + \hat{y}_k)\right), \tag{26}$$

where $L$ is the number of perturbed features in the input.

**Robustness.** Within the robustness category, we evaluate three metrics, including, *Relative Input Stability* (RIS), *Relative Representation Stability* (RRS), *Relative Ouput Stability* (ROS) (Agarwal et al., 2022a). RIS extends (Alvarez-Melis & Jaakkola, 2018b), which is a measure of how much the explanation changes *w.r.t.* the input under slight perturbation $\hat{\boldsymbol{x}} = \boldsymbol{x} + \boldsymbol{u}_i$. The change is measured as the $l_p$ norm, and the RIS metric only considers perturbations that result in the same model prediction, *i.e.,* $f(\boldsymbol{x}) = f(\hat{\boldsymbol{x}})$. It is defined as follows

$$\Psi_{\text{RIS}} = \max_{\hat{\boldsymbol{x}}} \frac{\left\| \frac{\phi(\boldsymbol{x},f,\hat{y};\lambda) - \phi(\hat{\boldsymbol{x}},f,\hat{y};\lambda)}{\phi(\boldsymbol{x},f,\hat{y};\lambda)} \right\|_p}{\max\left( \left\| \frac{\boldsymbol{x}-\hat{\boldsymbol{x}}}{\boldsymbol{x}} \right\|_p, \epsilon_{min} \right)}, \ \forall \hat{\boldsymbol{x}} \in \mathcal{N}_\epsilon; f(\boldsymbol{x}) = f(\hat{\boldsymbol{x}}) \tag{27}$$

where $\epsilon_{min} > 0$ ensures a non-zero denominator.

In contrast to the RIS metric, RRS considers the internal representation of the model $\mathcal{L}(\cdot)$ (*e.g.,* an output embedding), while maintaining similar perturbation conditions

$$\Psi_{\text{RRS}} = \max_{\hat{\boldsymbol{x}}} \frac{\left\| \frac{\phi(\boldsymbol{x},f,\hat{y};\lambda) - \phi(\hat{\boldsymbol{x}},f,\hat{y};\lambda)}{\phi(\boldsymbol{x},f,\hat{y};\lambda)} \right\|_p}{\max\left( \left\| \frac{\mathcal{L}_{\boldsymbol{x}}-\mathcal{L}_{\hat{\boldsymbol{x}}}}{\mathcal{L}_{\boldsymbol{x}}} \right\|_p, \epsilon_{min} \right)}, \ \forall \hat{\boldsymbol{x}} \in \mathcal{N}_\epsilon; f(\boldsymbol{x}) = f(\hat{\boldsymbol{x}}) \tag{28}$$

where $\epsilon_{min} > 0$ ensures a non-zero denominator.

ROS makes similar adaptations as the RRS metric, assumes however that the model's internal representations are not accessible. Instead the output logits $h(\boldsymbol{x})$, and $h(\hat{\boldsymbol{x}})$ are assessed

$$\Psi_{\text{ROS}} = \max_{\hat{\boldsymbol{x}}} \frac{\left\| \frac{\phi(\boldsymbol{x},f,\hat{y};\lambda) - \phi(\hat{\boldsymbol{x}},f,\hat{y};\lambda)}{\phi(\boldsymbol{x},f,\hat{y};\lambda)} \right\|_p}{\max\left( \left\| h(\boldsymbol{x}) - h(\hat{\boldsymbol{x}}) \right\|_p, \epsilon_{min} \right)}, \ \forall \hat{\boldsymbol{x}} \in \mathcal{N}_\epsilon; f(\boldsymbol{x}) = f(\hat{\boldsymbol{x}}) \tag{29}$$

where $\epsilon_{min} > 0$ ensures a non-zero denominator.

**Sensitivity.** Within the sensitivity category, we evaluate three metrics, including, *Model Parameter Randomisation Test* (MPRT) (Adebayo et al., 2018), *Smooth Model Parameter Randomisation Test* (sMPRT), *Efficient Model Parameter Randomisation* (eMPRT) (Hedström et al., 2024). MPRT measures the similarity between the original explanation $\boldsymbol{e}_l$, and the explanation $\hat{\boldsymbol{e}} := \phi(\boldsymbol{x}, \hat{f}_l^t, y)$ of the perturbed model $\hat{f}_l^t$ randomised in a top-down fashion up to layer $l \in [L, L-1, \dots, 1]$

$$\hat{q}^{\text{MPRT}} = \rho(\boldsymbol{e}, \hat{\boldsymbol{e}}_l), \tag{30}$$

with similarity function $\rho : \mathbb{R}^D \times \mathbb{R}^D \mapsto \mathbb{R}$.

sMPRT computes a quality estimate $\hat{q} \in \mathbb{R}$ between explanations $\boldsymbol{e}_i := \phi(\hat{\boldsymbol{x}}_i, f, y; \lambda)$, and $\hat{\boldsymbol{e}}_{l,i} := \phi(\hat{\boldsymbol{x}}_i, \hat{f}_l^b, y; \lambda)$ averaged over $i \in [1, N]$ where $\hat{\boldsymbol{e}}_{l,i}$ corresponds to the perturbed model $\hat{f}_l^b$ randomised in a bottom-down fashion up to layer $l \in [1, 2, \dots, L]$

$$\hat{q}^{\text{sMPRT}} = \rho\left( \frac{1}{N} \sum_{i=1}^N \boldsymbol{e}_i, \frac{1}{N} \sum_{i=1}^N \hat{\boldsymbol{e}}_{l,i} \right), \tag{31}$$

with $\hat{\boldsymbol{x}}_i = \boldsymbol{x} + \eta_i$, and $\eta_i \sim \mathcal{N}(0, \sigma)$ with $||\eta_i||_p \le \epsilon$ holding with high probability, for $\sigma, \epsilon \in \mathbb{R}$.

eMPRT measures the relative rise in the complexity of the explanation from a fully randomised model $\hat{f}$ such that $\hat{\boldsymbol{e}} := \phi(\boldsymbol{x}, \hat{f}, y; \lambda)$:

$$\hat{q}^{\text{eMPRT}} = \frac{c(\hat{\boldsymbol{e}}) - c(\boldsymbol{e})}{c(\boldsymbol{e})} \tag{32}$$

where $c : \mathbb{R}^D \mapsto \mathbb{R}$ is a complexity function, *e.g.,* discrete entropy.

## A.5 Analysing Violations of Model Assumptions

To understand whether perturbation techniques commonly employed for robustness, sensitivity, and faithfulness evaluations generally fulfill the critical assumptions of model distortion (Assumptions 1-3), we performed several experiments. To investigate how often model robustness, sensitivity, and faithfulness (Assumptions 1-3) hold versus fail in practice, we set up a simple experiment that tracks model, and explanation distortions, *i.e.,* $\boldsymbol{D}_f$, and $\boldsymbol{D}_\phi$, while applying perturbation commonly used in evaluation such as additive Gaussian noise for robustness evaluation, top-down, and bottom-up layer-by-layer parameter randomisation for sensitivity evaluation, and cumulative masking for faithfulness evaluation.

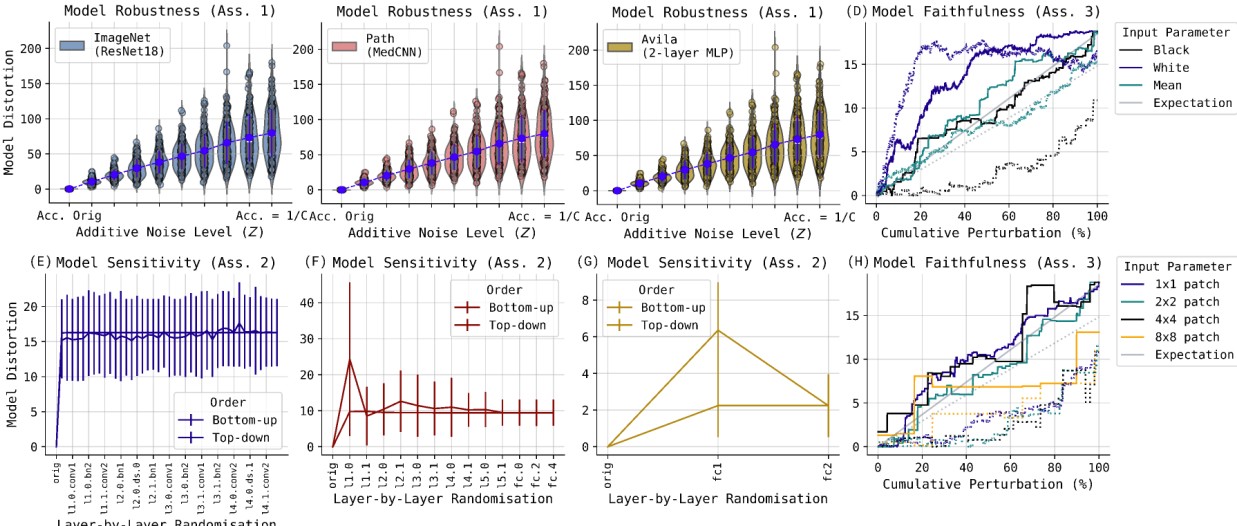

Figure A.3: Impact of model distortion (y-axis) over common perturbation types in robustness, sensitivity, and faithfulness evaluations, across different datasets, and NN architectures. (A), (B), and (C) depict the distribution of model distortions across different perturbation magnitudes of additive Gaussian noise for ImageNet (ResNet18), Path (MedCNN), and Avila (2-layer MLP), respectively. (D), (E), and (F) show the average, and standard deviation of model distortions over layer-wise top-down, and bottom-up randomisation for the same datasets (as indicated by colour). (G), and (H) display model distortions for randomly chosen MNIST (*solid* line), and fMNIST (*dashed* line) samples (LeNet) under cumulative perturbations using different patch sizes ($1 \times 1$, $2 \times 2$, $4 \times 4$, $8 \times 8$), and baseline replacement strategies (*black, white, mean*).

**Model Robustness Under Additive Noise.** To understand the extent to which model robustness (Assumption 1) is generally satisfied for robustness evaluation (Definition 2), we examine Figure A.3 (A), (B), and (C), and Figure A.4 (A) and (B). Here, the distribution of $\boldsymbol{D}_f$ is visualised over $Z = 10$ input perturbation steps, showing how model distortion varies with increasing input perturbation magnitude, using additive Gaussian noise, *i.e., , $\nu_i \sim \mathcal{N}(0, \sigma)$ to generate perturbed inputs $\hat{\boldsymbol{x}}_i = \boldsymbol{x} + \nu_i$, with $\sigma$ increasing until the model behaves randomly (*i.e.,* accuracy $= 1/C$). While the average trend (*blue* line) indicates that larger perturbation causes higher model distortion, sample-wise exceptions frequently appear. In Figure A.5, random sample trajectories reveal both correlated and uncorrelated patterns between perturbation levels and model distortions. This is a key observation, as it implies that model robustness cannot be assured by a general threshold without inspecting each evaluation sample individually.

**Model Sensitivity Under Layer-by-Layer Randomisation.** For sensitivity evaluations (Definition 3) to be meaningful, the model distortion caused by perturbation must be significant (Assumption 2). To test this practice, we perform consecutive layer-wise model parameter randomisation; in both a top-down (Adebayo et al., 2018), and bottom-up (Hedström et al., 2024) manner. From Figure A.3 (E), (F), and (G), and Figure A.4 (D) and (E), we observe that, although model distortion generally increases with layer-wise randomisation, there are exceptions of non-monotonicity (see, *e.g.,* Path, and Avila results in Figure A.3 (E), and (F), respectively). The high standard deviation (see the error bars) suggests that layer-wise randomisa-

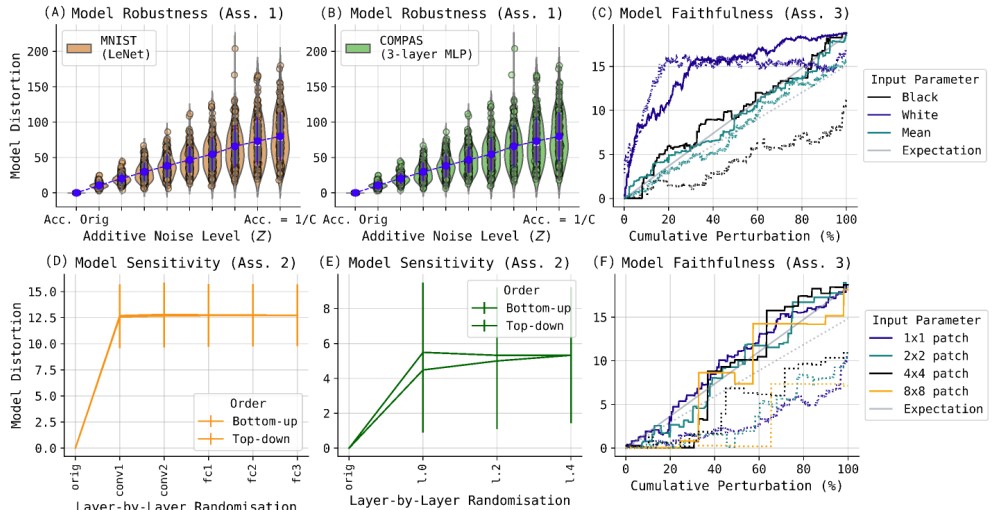

Figure A.4: Impact of model distortion (y-axis) over common perturbation types in robustness, sensitivity, and faithfulness evaluations, across different datasets, and NN architectures. (A), and (B) depict the distribution of model distortions across different perturbation magnitudes of additive Gaussian noise for MNIST (LeNet), and COMPAS (2-layer MLP), respectively. (D), and (E) show the average, and standard deviation of model distortions over layer-wise top-down, and bottom-up randomisation for the same datasets (as indicated by colour). (C), and (F) display model distortions for randomly chosen MNIST (*solid* line), and fMNIST (*dashed* line) samples (LeNet) under cumulative perturbations using different patch sizes (*1x1, 2x2, 4x4, 8x8*), and baseline replacement strategies (*black, white, mean*).

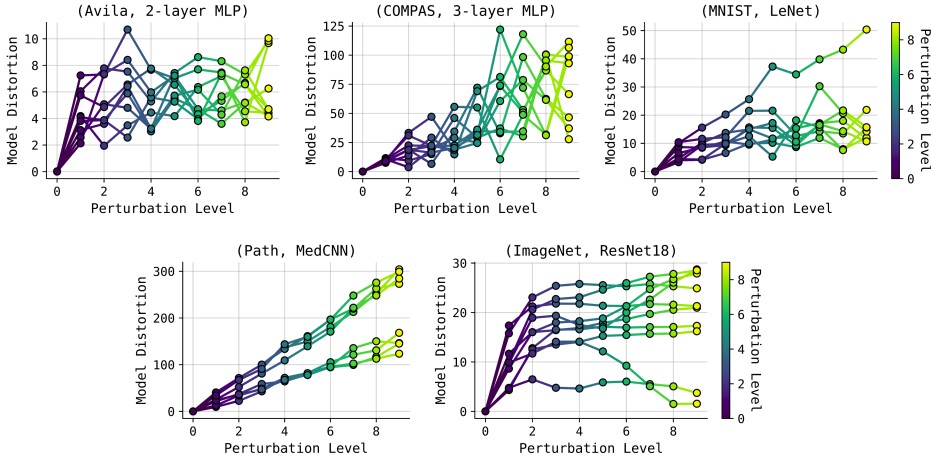

Figure A.5: Sample-wise trajectories of model distortion (*y-axis*) across different perturbation magnitudes of additive Gaussian noise. Each panel includes line plots across $N = 10$ samples, showing both correlated and uncorrelated outcomes.

tion fails to predictably dictate the degree of model distortion, undermining the assumption that significant model distortions will always occur in sensitivity evaluations.

**Model Faithfulness Under Cumulative Input Perturbation.** To investigate whether model distortion increases monotonically under cumulative input perturbation (Assumption 3), we measure $\boldsymbol{D}_f$ using a standard "pixel-flipping" faithfulness procedure (Bach et al., 2015). By randomising the perturbation order, the resulting faithfulness curve should reflect *only* the model's response; any deviation from a linear trend suggests that Assumption 3 is failed. Observing Figure A.3 (D), and (H), and Figure A.4 (C) and (F), we see that neither patch size (top) nor replacement strategy (bottom) induces monotonic non-decreasing model behaviour. While these results are expected due to the model's inherent nonlinearity, and OOD effects (Hase et al., 2021; Hesse et al., 2024), it is not accounted for in the faithfulness evaluation itself (Definition 4).

When genuine signals (*i.e.,* explanation quality) are not decoupled from noise (*i.e.,* non-monotonic model behaviour), interpretations may become biased (Hooker et al., 2019; Brocki & Chung, 2022; Brunke et al., 2020).

Together, these results reveal how easily, and systematically Assumptions 1-3 are violated by perturbation strategies commonly applied in practice (Section 2.1.1). Our findings are consequential as they demonstrate that the validity of existing robustness, sensitivity, and faithfulness evaluations (Definitions 2-4) are frequently undermined. As displayed in Figure A.3, there are many sample-wise exceptions where the perturbation magnitude, and the model distortion quantity are not strictly monotonically related, challenging the assumption that increased perturbations lead to proportionally greater distortions, and vice-versa.

### A.5.1 Issues with Cumulative Input Perturbation

If small changes in input parameters, cause large variations in evaluation outcomes, evaluation reliability is compromised. Corroborating previous studies (Brunke et al., 2020; Brocki & Chung, 2022; Rong et al., 2022), the varied faithfulness curves in Figure A.3 (G), and (H), and Figure A.4 (C) and (F), demonstrate how input parameter choices, such as patch size or pixel value, can drastically influence the evaluation outcomes across tasks, *i.e.,* act as evaluation confounds (*cf.* the same parameter for MNIST *solid* line vs. fMNIST *dotted* line). These variations between tasks expose a simple, yet systematically overlooked issue in faithfulness evaluations: that parameter choices to perturb the input inherently introduce task-specific biases to the evaluation. Attempts to mitigate these biases—using inverse curves (Blücher et al., 2024) or assessing the OOD impact of perturbations (Qiu et al., 2021; Haug et al., 2021)—fail to address the core problem: that evaluation methods (Section 2.1.2) that require input parameters to be tuned according to its task, are inherently biased, impeding impartial comparisons across tasks, and explanation approaches.

### A.6 Alignment Patterns, and Extended Results

Figure A.6 provides complementary results to Figure 3 in the main manuscript. The scatter points are coloured by perturbation magnitude up to $Z = 10$, using additive Gaussian noise. The varying but consistent overlaps of points of high and low perturbation magnitudes alongside the almost uniform distribution along the y-axis, illustrate that perturbation effects are not guaranteed to have a proportional effect on the model and explanation functions. Thus, using the perturbation magnitude as an indicator of the magnitude of which the explanation should change (as done in existing evaluations, see Definitions 2-3) is not reliable.

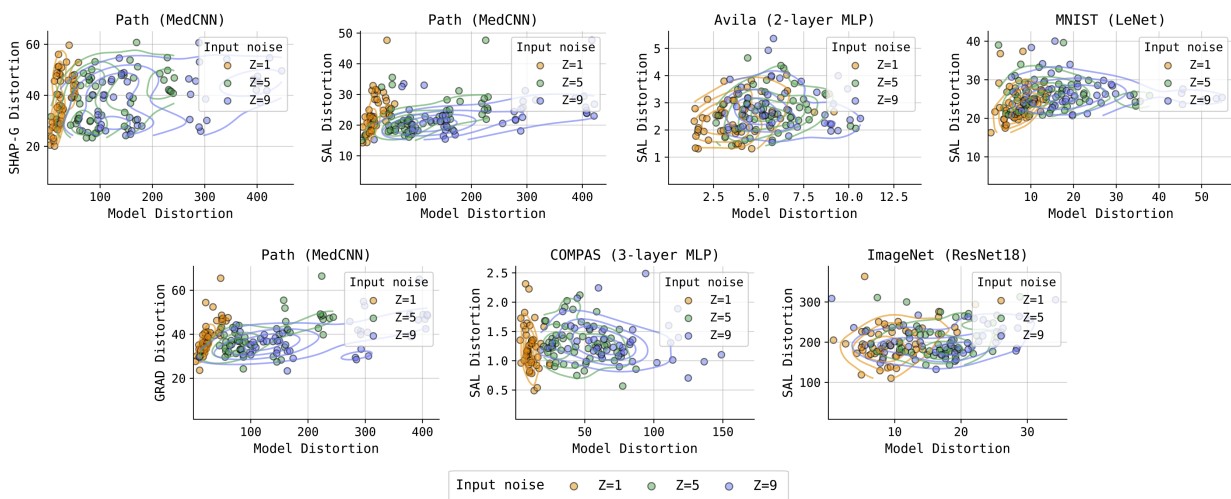

Figure A.6: Model (x-axis), and explanation distortions (y-axis) under varying levels of additive Gaussian input noise for vision and tabular tasks, as indicated in the titles. Scatter points represent individual samples, coloured by perturbation magnitude ($Z = 10$), with overlapping contours highlighting the relative alignment patterns. The individual plots contain SHAP-G, GRAD, and SAL explanations, as indicated by the y-axis labels.

Next, we show the change in the relationship between model distortion and explanation distortion when comparing the commonly used input perturbations with model perturbation (`Fast-GEF`) and when measuring explanation distortion and model distortion on the same geometry (`GEF`). Figure A.7 illustrates the relationship of model and explanation distortion across these three cases from left to right. Each contour plot includes $N = 10$ samples per perturbation magnitudes from $Z = 1$ to $Z = 5$. The scatter points are colored to one of three increasing perturbation magnitudes ($Z = 1$ to $Z = 3$). We can observe the distortion quantities across tasks with increasing model complexity, from a simpler tabular task (first row) to a highly parameterised model for a vision task (last row).

As expected, the first column (input perturbation) coincides with Figure A.6 and yields the same findings. However, in the following two columns, we can observe how `Fast-GEF` and `GEF` behave in practice. Most notably we observe that the alignment between model and explanation distortion changes from less complex to more complex tasks. Furthermore, we find that `Fast-GEF` tends to generate higher coherence compared to input perturbation, except for ImageNet, and that `GEF` yields the most coherent distortions. While these findings appear to support both our approaches, it is important to note that without access to ground truth, it is unclear whether the contour plots should necessarily show stronger coherence (*i.e.,* post-perturbed correlation), as it depends on the relationship between the explanation and model functions.

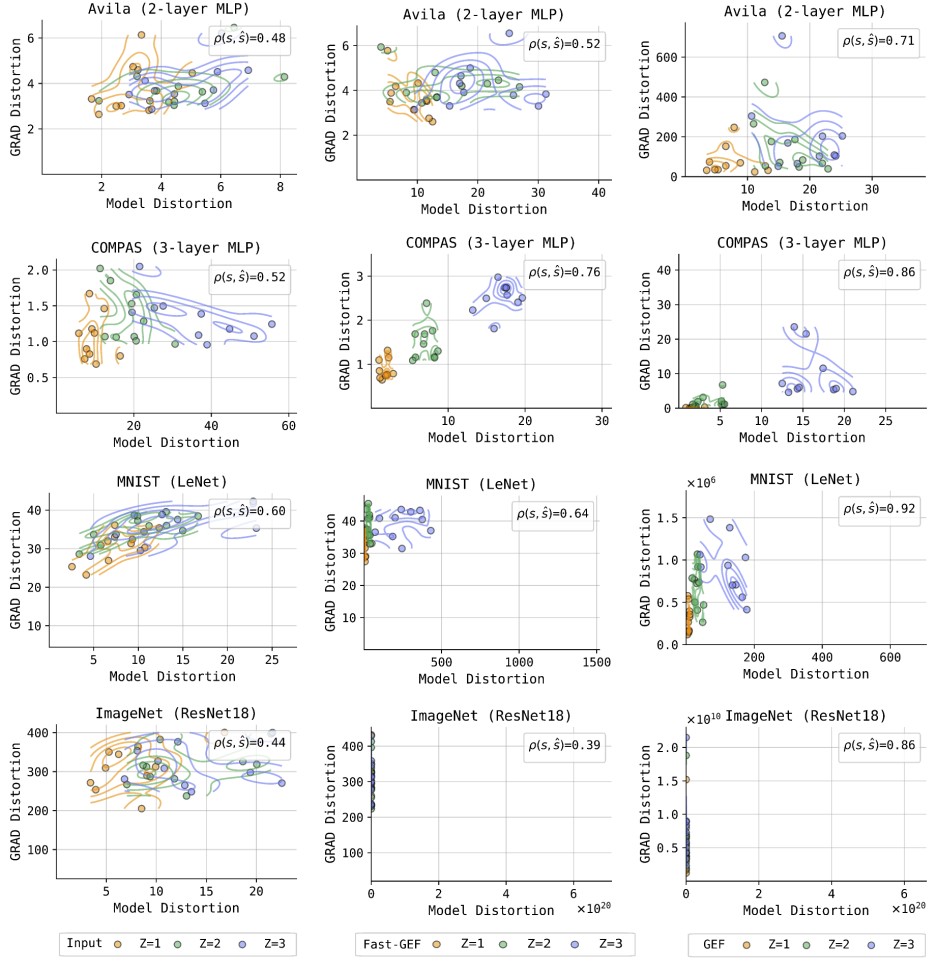

Figure A.7: Each plot shows the model (x-axis), and explanation distortions (y-axis) under different types of noise for *Gradient* (GRAD) explanation. The first column shows distortion outcomes after applying additive Gaussian input noise. The second and third columns show distortion outcomes after applying model parameter scaling (Section 5.1). The second column computes explanation distortion using `Fast-GEF` and the third column computes distortion using `GEF` (*i.e.,* with pullback mechanism). Scatter points represent individual samples, coloured by perturbation magnitude ($z$=1, $z$=2, $z$=3), with $Z = 5$ number of steps.

## A.7 Ablation Study

To better understand the influence of the hyperparameters, on the proposed `GEF` evaluation method, we conducted an ablation study. We employed two tasks, *i.e.,* a tabular dataset (Avila) using a 2-layer MLP model with SAL explanations and a vision dataset (MNIST) using a LeNet model with 250 random explanations, sampled from a uniform distribution, *i.e.,* $\hat{e}_i = \mathcal{U}(0, 1)$. For each hyperparameter, *i.e.,* the number of perturbed models $M$, the length of the perturbation path $Z$, the number of summation steps $T$, and the number of samples $K$, we enumerated over values from 0 to 20, while fixing the others at a default value of 10. For each configuration, we recorded mean (solid line) and standard deviation (shaded area) of the model distortion, explanation distortion, Jacobian quantity, and the mean computation time.

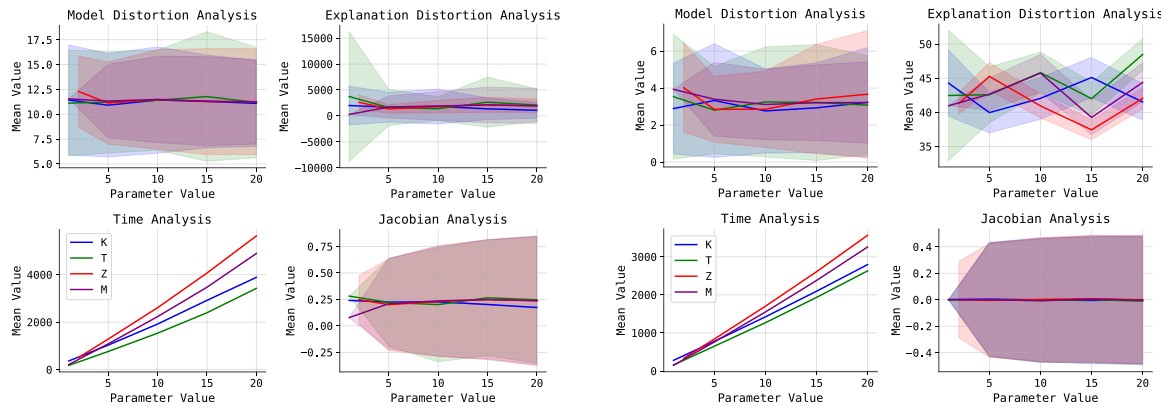

Figure A.8: Ablation study results across hyperparameters $M$, $Z$, $T$, and $K$ for two tasks: (*left*) Saliency explanation on Avila (2-layer MLP), and (*right*) Random explanation on MNIST (LeNet). The mean value (*solid* line) and variance (*shaded* area) are reported. The time analysis is measured in seconds.

Figure A.8 demonstrates the results for both the tabular (*left*) and the vision task (*right*). As can be observed by the converging values of the standard deviation and means, the hyperparameters are resilient to key parameter changes once parameter values reach 5 or higher. The Jacobian variance reflects the curvature captured in its estimate. While the variance increases for parameter values above 5, the mean stabilises, indicating diminishing returns in capturing additional curvature. At parameter values of 5, the majority of the curvature is already captured, providing a practical trade-off between computational efficiency and quality of approximation. Considering computational time, all parameters lead to a linear, non-negligible increase. Among them, $Z$ and $M$ are identified as the primary drivers of time. Based on these experimental findings, setting $K = T = Z = M = 5$ balances computational efficiency and stability of the quality estimate.

## A.8 Experiments, and Extended Results

This section provides descriptions of experimental setups, and extended results, including meta-evaluations, and agreement between different scoring methods. Additionally, we present further results for random control variant sanity checks, cross-domain benchmarking, and LLM-x methodology, and extended results.

### A.8.1 Meta-Evaluation

To employ the scoring methodology (Section A.3), we used the pre-existing test suite available in the `MetaQuantus` library[7] with their pre-defined hyperparameters.

**MetaQuantus Hyperparameters.** We applied these metrics over $K = 5$ perturbations, conducting 3 iterations with the test configurations specified in the library for two different sets of explanation methods, namely {GRAD, G-CAM}, and {SAL, SHAP-G}, which were evaluated by each metric. The explanation

---

[7]Find the library at `https://github.com/annahedstroem/MetaQuantus/`.

method groups were created by randomly selecting methods from the complete set of available methods, ensuring consistency across various experimental setups, such as dataset, and model combinations. In terms of choosing $K$, and the number of iterations, we followed the recommendations from the original study to keep the standard deviation between different sets relatively low. To ensure a fair comparison across metrics, all shared hyperparameters were assigned the same values.

**Metrics Hyperparameters.** All metrics have been implemented in `Quantus` (Hedström et al., 2023b). Different hyperparameters were chosen for the individual metrics based on the dataset. For the robustness metrics, we use 5 noisy samples, and employ additive Gaussian noise such that $\nu \sim \mathcal{N}(0, 0.001)$. For the faithfulness metrics, we use 28 features per perturbation step, and a patch size of 7 for the MNIST, and fMNIST datasets. For ImageNet, we set the number of features to 896, and the patch size to 28. For FC, similar to the robustness metrics, we let it run 5 times. For the sensitivity metrics, namely MPRT, and sMPRT, we use a noise magnitude of 0.01 for each sample, and sMPRT uses 5 samples in its calculation. For all sensitivity metrics, we use the Spearman rank correlation coefficient.

Table A.1: MC scores and standard deviation for unified, and faithfulness methods listed in A.4.5 for ImageNet, MNIST, and fMNIST datasets. The final row shows the mean score for each metric across the datasets. Values range between [0, 1], with higher values indicating better outcomes. Due to computational constraints, `GEF` scores are only computed for fMNIST, and MNIST datasets.

| | UNIFIED | | FAITHFULNESS | | |
|---|---|---|---|---|---|
| | GEF | Fast-GEF | PF | FC | RP |
| IMAGENET | NAN ± NAN | 0.78 ± 0.02 | 0.63 ± 0.01 | 0.51 ± 0.02 | 0.63 ± 0.06 |
| MNIST | 0.75 ± 0.07 | 0.74 ± 0.03 | 0.61 ± 0.04 | 0.63 ± 0.03 | 0.59 ± 0.03 |
| fMNIST | 0.71 ± 0.07 | 0.71 ± 0.03 | 0.63 ± 0.01 | 0.50 ± 0.04 | 0.58 ± 0.09 |
| **Mean** | **0.73 ± 0.07** | **0.74 ± 0.03** | **0.62 ± 0.02** | **0.56 ± 0.03** | **0.59 ± 0.06** |

Table A.2: MC scores and standard deviation for sensitivity, and robustness methods listed in A.4.5 for ImageNet, MNIST, and fMNIST datasets. The final row shows the mean score for each metric across the datasets. Values range between [0, 1], with higher values indicating better outcomes.

| | SENSITIVITY | | | ROBUSTNESS | | |
|---|---|---|---|---|---|---|
| | MPRT | sMPRT | eMPRT | RIS | ROS | RRS |
| IMAGENET | 0.71 ± 0.02 | 0.69 ± 0.04 | 0.71 ± 0.02 | 0.72 ± 0.06 | 0.76 ± 0.07 | 0.75 ± 0.04 |
| MNIST | 0.63 ± 0.02 | 0.66 ± 0.04 | 0.76 ± 0.03 | 0.73 ± 0.02 | 0.70 ± 0.09 | 0.74 ± 0.09 |
| fMNIST | 0.63 ± 0.01 | 0.67 ± 0.05 | 0.67 ± 0.05 | 0.70 ± 0.02 | 0.77 ± 0.06 | 0.70 ± 0.03 |
| **Mean** | **0.64 ± 0.02** | **0.67 ± 0.04** | **0.71 ± 0.03** | **0.72 ± 0.03** | **0.74 ± 0.07** | **0.73 ± 0.05** |

**Extended Results.** In Tables A.1, and A.2, we provide the corresponding results for Figure 6.

### A.8.2 Agreement between `GEF`, and `Fast-GEF`

To determine whether the simpler, computationally efficient `Fast-GEF` method can serve as an alternative to the more exact but computationally intensive `GEF` method, we compare the agreement between their respective faithfulness estimates. For a subset of explanation methods, and tasks (see Table 2), we thus compute scores, and rank explanation methods from R1 to RN. While it is expected that estimates from the two methods differ, a high agreement in a categorical ranking would make `Fast-GEF` a practical alternative in resource-constrained environments.

**Results.** Figure A.9 (A) visually compares how `GEF` and `Fast-GEF` ranks (x-axis) each explanation method in terms of increases (y-axis), highlighting the relative agreement between them. The explanations in the tabular, and text tasks show perfect ranking agreement. In the MNIST vision task, with minimal nominal differences, GRAD, and SHAP-G methods disagree in their ranking (R1, and R2), but such disagreement can be expected acknowledging the algorithmic similarity between these explanation methods. In the Derma vision task, the same pattern is observed, yet with a slightly larger difference for the global method FO-50. Interestingly, we observe that nominal differences are pronounced for global methods (DV-50, and FO-50), and that `Fast-GEF` tends to generate slightly lower faithfulness estimates *cf.* `GEF`.

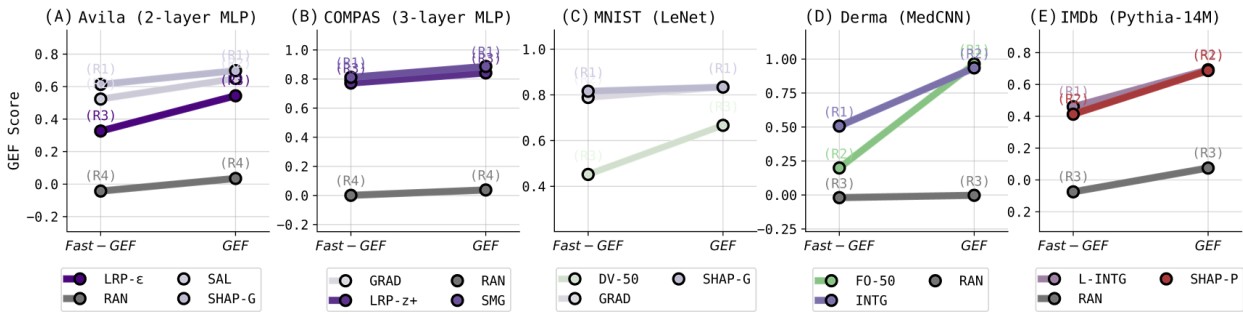

Figure A.9: (A) to (E) illustrates the `GEF` scores of `GEF` and `Fast-GEF` (with $M = 1$) for various explanation methods, and tasks. Explanation methods are ranked between R1 to RN, in descending order.

### A.8.3 Scoring Control Variants

Next, we validate that both `GEF` and `Fast-GEF` assign low faithfulness scores to different control variant explanations. In our sanity checks, we evaluate explanations generated by uniform sampling, *i.e.,* $\hat{e}_i \sim \mathcal{U}(0,1)$, a constant value, *i.e.,* $\hat{e}_i = \mathbf{0}$, and with a model-independent Sobel filter. For non-random reference, we evaluate GRAD explanations for the predicted class of the Derma task (see Table 2) (Sobel et al., 1968). For comparability, we extend this sanity check exercise to one metric per evaluative criteria, *i.e.,* `FC` (faithfulness), `MPRT` (sensitivity), and `RIS` (robustness). Hyperparameters are provided in Appendix A.8.1.

Table A.3: Evaluation scores of Derma (MedCNN) explanations for three random, and one regular (GRAD) explanation. The arrow ($\uparrow,\downarrow$) indicates whether higher or lower values are better. A nan value indicates that no score is produced.

| EXPLANATION | GEF ($\uparrow$) | FAST-GEF ($\uparrow$) | FC ($\uparrow$) | MPRT ($\downarrow$) | RIS ($\downarrow$) |
|---|---|---|---|---|---|
| CONTROL VAR. CONSTANT | NAN $\pm$ NAN | NAN $\pm$ NAN | NAN $\pm$ NAN | NAN $\pm$ NAN | 0.11 $\pm$ 0.29 |
| CONTROL VAR. RANDOM UNIFORM | -0.01 $\pm$ 0.30 | -0.01 $\pm$ 0.22 | -0.00 $\pm$ 0.51 | -0.00 $\pm$ 0.04 | 3.21 $\pm$ 2.89 |
| CONTROL VAR. SOBEL FILTER | NAN $\pm$ NAN | NAN $\pm$ NAN | -0.01 $\pm$ 0.50 | 1.00 $\pm$ 0.00 | 82197.21 $\pm$ 132718.26 |
| GRAD | 0.47 $\pm$ 0.23 | 0.48 $\pm$ 0.15 | -0.05 $\pm$ 0.49 | 0.01 $\pm$ 0.04 | 1764.60 $\pm$ 10007.26 |

**Results.** Table A.3 presents the results. Some metrics produce no values (nan), *e.g.,* when correlating identical vectors, and by that identify the unfaithful explanation. `Fast-GEF`, and `GEF` consistently assign low scores to random explanations, and high scores to non-random GRAD explanations, indicating their ability to identify the control explanations. Conversely, other metrics fail at least in one random test, either showing little discrepancy between regular, and control variants or even giving higher scores to the control. For instance, `MPRT`, and `RIS` score random uniform explanations as good or better than regular ones.

### A.8.4 Cross-Domain Benchmarking

**Extended Results.** We benchmark various local, and global explanation methods with `GEF` and `Fast-GEF`. In Figure A.10, we extend the results in Figure 8.

The results presented in Figure 7 are provided in Tables A.4, and A.5.

### A.8.5 LLM-x

In the following, we provided extended results of the LLM-x experiments.

**Extended Results.** The results presented in Figure 9 are provided in Tables A.6, and A.7.

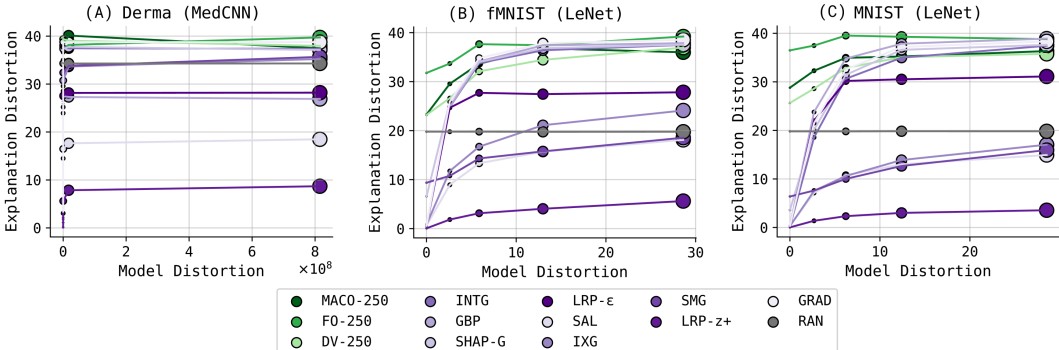

Figure A.10: `Fast-GEF` results for vision tasks. (A), (B), and (C) plot the model and explanation distortion for Derma (MedCNN), and fMNIST (LeNet), and MNIST (LeNet) along the perturbation path with $Z = 5$ perturbation steps. The size of the scatter point represents each perturbation steps, from 1 to 5.

Table A.4: `GEF` results on local methods for tabular tasks. Mean faithfulness scores, and standard errors are reported, with higher values indicating better quality.

| | TASK | ADULT (3-LAYER MLP) | ADULT LR | AVILA (2-LAYER MLP) | COMPAS (3-LAYER MLP) | COMPAS LR |
|---|---|---|---|---|---|---|
| | SMG | $0.86 \pm 0.00$ | $0.69 \pm 0.00$ | $0.72 \pm 0.01$ | $0.81 \pm 0.01$ | $0.73 \pm 0.01$ |
| | SHAP-G | $0.78 \pm 0.00$ | $0.84 \pm 0.01$ | $0.75 \pm 0.01$ | $0.84 \pm 0.00$ | $0.66 \pm 0.01$ |
| | SAL | $0.84 \pm 0.00$ | $0.76 \pm 0.00$ | $0.69 \pm 0.01$ | $0.75 \pm 0.01$ | $0.70 \pm 0.01$ |
| LOCAL METHODS | RAN | $-0.00 \pm 0.02$ | $0.00 \pm 0.02$ | $0.00 \pm 0.02$ | $0.02 \pm 0.02$ | $0.02 \pm 0.02$ |
| | LRP-$\varepsilon$ | $0.66 \pm 0.01$ | $0.61 \pm 0.01$ | $0.52 \pm 0.01$ | $0.74 \pm 0.01$ | $0.59 \pm 0.01$ |
| | LRP-$z^+$ | $0.79 \pm 0.00$ | $0.75 \pm 0.00$ | $0.66 \pm 0.01$ | $0.78 \pm 0.00$ | $0.70 \pm 0.01$ |
| | IXG | $0.84 \pm 0.00$ | $0.77 \pm 0.01$ | $0.69 \pm 0.01$ | $0.74 \pm 0.00$ | $0.72 \pm 0.00$ |
| | INTG | $0.82 \pm 0.00$ | $0.82 \pm 0.00$ | $0.69 \pm 0.01$ | $0.81 \pm 0.00$ | $0.80 \pm 0.00$ |
| | GRAD | $0.86 \pm 0.00$ | $0.74 \pm 0.00$ | $0.69 \pm 0.01$ | $0.81 \pm 0.01$ | $0.67 \pm 0.01$ |
| | GBP | $0.80 \pm 0.00$ | $0.60 \pm 0.01$ | $0.68 \pm 0.01$ | $0.78 \pm 0.01$ | $0.70 \pm 0.01$ |

Table A.5: `Fast-GEF` result on local methods for vision tasks. Mean faithfulness scores, and standard errors are reported, with higher values indicating better quality.

| | TASK | DERMA MEDCNN | FMNIST LENET | IMAGENET-1K RESNET18 | MNIST LENET | PATH MEDCNN |
|---|---|---|---|---|---|---|
| | SMG | $0.61 \pm 0.01$ | $0.69 \pm 0.01$ | $0.63 \pm 0.01$ | $0.73 \pm 0.01$ | $0.63 \pm 0.02$ |
| | SHAP-G | $0.49 \pm 0.01$ | $0.74 \pm 0.01$ | $0.69 \pm 0.01$ | $0.77 \pm 0.01$ | $0.60 \pm 0.01$ |
| | SAL | $0.67 \pm 0.01$ | $0.76 \pm 0.01$ | $0.71 \pm 0.01$ | $0.74 \pm 0.01$ | $0.65 \pm 0.01$ |
| LOCAL METHODS | RAN | $0.01 \pm 0.01$ | $0.00 \pm 0.01$ | $-0.00 \pm 0.01$ | $-0.00 \pm 0.01$ | $-0.01 \pm 0.01$ |
| | LRP-$\varepsilon$ | $0.45 \pm 0.01$ | $0.70 \pm 0.01$ | $0.35 \pm 0.01$ | $0.73 \pm 0.01$ | $0.66 \pm 0.01$ |
| | LRP-$z^+$ | $0.74 \pm 0.01$ | $0.73 \pm 0.01$ | $0.91 \pm 0.00$ | $0.73 \pm 0.01$ | $0.71 \pm 0.01$ |
| | IXG | $0.73 \pm 0.01$ | $0.72 \pm 0.01$ | $0.64 \pm 0.01$ | $0.73 \pm 0.01$ | $0.71 \pm 0.01$ |
| | INTG | $0.49 \pm 0.01$ | $0.73 \pm 0.01$ | $0.78 \pm 0.01$ | $0.77 \pm 0.01$ | $0.71 \pm 0.01$ |
| | GRAD | $0.54 \pm 0.01$ | $0.76 \pm 0.01$ | $0.71 \pm 0.01$ | $0.79 \pm 0.01$ | $0.66 \pm 0.01$ |
| | GBP | $0.63 \pm 0.01$ | $0.76 \pm 0.01$ | $0.85 \pm 0.01$ | $0.77 \pm 0.01$ | $0.64 \pm 0.01$ |
| | MACO-50 | $0.16 \pm 0.02$ | $0.47 \pm 0.02$ | $0.29 \pm 0.02$ | $0.29 \pm 0.02$ | $0.42 \pm 0.01$ |
| | MACO-250 | $0.30 \pm 0.02$ | $0.54 \pm 0.01$ | $0.31 \pm 0.01$ | $0.31 \pm 0.02$ | $0.40 \pm 0.01$ |
| | MACO-100 | $0.24 \pm 0.01$ | $0.52 \pm 0.01$ | $0.31 \pm 0.01$ | $0.41 \pm 0.01$ | $0.45 \pm 0.01$ |
| GLOBAL METHODS | FO-50 | $0.17 \pm 0.02$ | $0.42 \pm 0.01$ | $0.22 \pm 0.02$ | $0.26 \pm 0.02$ | $0.35 \pm 0.01$ |
| | FO-250 | $0.36 \pm 0.01$ | $0.38 \pm 0.02$ | $0.19 \pm 0.02$ | $0.15 \pm 0.02$ | $0.31 \pm 0.01$ |
| | FO-100 | $0.28 \pm 0.01$ | $0.36 \pm 0.02$ | $0.23 \pm 0.02$ | $0.21 \pm 0.02$ | $0.27 \pm 0.01$ |
| | DV-50 | $0.38 \pm 0.01$ | $0.54 \pm 0.02$ | $0.26 \pm 0.02$ | $0.36 \pm 0.02$ | $0.45 \pm 0.01$ |
| | DV-250 | $0.44 \pm 0.01$ | $0.50 \pm 0.01$ | $0.40 \pm 0.02$ | $0.40 \pm 0.02$ | $0.48 \pm 0.01$ |
| | DV-100 | $0.43 \pm 0.02$ | $0.49 \pm 0.02$ | $0.40 \pm 0.02$ | $0.43 \pm 0.02$ | $0.51 \pm 0.01$ |

Table A.6: `Fast-GEF` results on LLM-x, and local methods for top-$K$ tasks. Mean faithfulness scores, and standard errors are reported, with higher values indicating better quality.

| | Task | SMS Spam BERT-TINY FT | SST2 BERT-TINY FT |
|---|---|---|---|
| LOCAL METHODS | SHAP-P-5 | $0.62 \pm 0.01$ | $0.75 \pm 0.01$ |
| | SHAP-P-10 | $0.62 \pm 0.01$ | $0.75 \pm 0.01$ |
| | RAN-5 | $0.08 \pm 0.01$ | $-0.08 \pm 0.01$ |
| | RAN-10 | $0.03 \pm 0.01$ | $-0.10 \pm 0.01$ |
| | LLM-X-5 | $0.06 \pm 0.02$ | $0.05 \pm 0.02$ |
| | LLM-X-10 | $0.05 \pm 0.02$ | $0.08 \pm 0.02$ |
| | L-INTG-5 | $0.58 \pm 0.01$ | $0.77 \pm 0.01$ |
| | L-INTG-10 | $0.58 \pm 0.01$ | $0.77 \pm 0.01$ |

Table A.7: `Fast-GEF` results on LLM-x, and local methods for top-$K$ tasks. Mean faithfulness scores, and standard errors are reported, with higher values indicating better quality.

| Task | SMS Spam BERT-TINY FT | SST2 BERT-TINY FT |
|---|---|---|
| LLM-X | $-4.25 \pm 8.42$ | $-3.73 \pm 7.72$ |
| L-INTG | $195.49 \pm 2.91$ | $238.15 \pm 2.98$ |
| SHAP-P | $185.95 \pm 3.06$ | $230.27 \pm 3.75$ |

## A.9 Notation Tables

All notations used in this paper is provided in the following.

### Spaces, and Elements

| | |
|---|---|
| $\mathcal{X}, \boldsymbol{x}$ | The input space $\mathcal{X} \subseteq \mathbb{R}^D$ with a sample $\boldsymbol{x} \in \mathcal{X}$ |
| $\mathcal{F}, \theta$ | The model space $\mathcal{F} \subseteq \mathbb{R}^U$ with parameters $\theta \in \mathcal{F}$ |
| $\mathcal{Y}, \boldsymbol{y}$ | The function output space $\mathcal{Y} \subseteq \mathbb{R}^C$ with logits $\boldsymbol{y} \in \mathcal{Y}; \boldsymbol{y} = [y_1, \ldots, y_C]^T$ for $C$ classes $y_c \in \boldsymbol{y} \forall c \in [1, C]$ |
| $\mathcal{E}, \boldsymbol{e}$ | The explanation space $\mathcal{E} \subseteq \mathbb{R}^V$ with an explanation $\boldsymbol{e} \in \mathcal{F}$ |
| $\mathcal{Q}, q$ | The evaluation space $\mathcal{Q} \subseteq \mathbb{R}^M$ with a quality estimate $q \in \mathcal{Q}$ |
| $\mathcal{S}, \boldsymbol{s}$ | A set of spaces $S \subset \{\mathcal{X}, \mathcal{F}, \mathcal{Y}, \mathcal{E}, \mathcal{Q}\}$ where $\mathcal{S} \subseteq \mathbb{R}^S, S \in \mathbb{N}$ with $\boldsymbol{s} \in \mathcal{S}$ |
| $\mathcal{H}, \boldsymbol{h}$ | A subset of spaces $\mathcal{H} \subseteq \{\mathcal{F}, \mathcal{E}\}$ with $\boldsymbol{h} \in \mathcal{H}$ |
| $\hat{\boldsymbol{s}}, \hat{\boldsymbol{x}}, \hat{\theta}, \hat{y}, \hat{\boldsymbol{e}}$ | A sample, input, parameters, logit, explanation, post-perturbation. |

### Functions

| | |
|---|---|
| $f$ | A classifier function $f : \mathcal{X} \to \mathcal{Y}$ with $f(\boldsymbol{x}; \theta) = \boldsymbol{y}$ (we refer $f_\theta$ as $f$), parameterised by $\theta$ |
| $\phi_L$ | A local explanation function $\phi_L : \mathcal{F} \times \mathcal{X} \times \mathcal{Y} \to \mathbb{R}^V$ with $\phi_L(f, \boldsymbol{x}, y; \lambda) = \boldsymbol{e}$, parameterised by $\lambda$ |
| $\phi_G$ | A global explanation function $\phi_G : \mathcal{F} \times \mathcal{Y} \to \mathbb{R}^V$ with $\phi_G(f, y; \kappa) = \boldsymbol{e}$, parameterised by $\kappa$ |
| $\phi$ | Collectively, denoting $\phi_L$, and $\phi_G$ although they formally reside in different spaces |
| $\Psi$ | An evaluation function $\Psi : \mathcal{E} \times \mathcal{X} \times \mathcal{F} \times \mathcal{Y} \to \mathbb{R}$ with $\Psi(\boldsymbol{e}, \boldsymbol{x}, f, y; \tau) = \boldsymbol{q}$, parameterised by $\tau$ |
| $\mathcal{P}_\mathcal{S}$ | A perturbation function $\mathcal{P} : \mathcal{S} \to \mathcal{S}$ where $\mathcal{P}(\boldsymbol{s}; \omega)$ on space $\mathcal{S}$ |
| $\delta$ | A general discrepancy function $\delta : \mathcal{S} \times \mathcal{S} \to \mathbb{R}$ with $\delta(\boldsymbol{s}, \hat{\boldsymbol{s}}) = \xi$, parameterised by $\omega \in \mathbb{R}$ |
| $k$ | A separate mapping function $k : \mathcal{S} \to \mathcal{H}$ mapping $\boldsymbol{s}, \hat{\boldsymbol{s}}$ to a distinct space $\mathcal{H}$ |
| $\boldsymbol{D}_k$ | A functional distortion $\boldsymbol{D}_k : \mathcal{S} \times \mathcal{S} \to \mathbb{R}$ with $\boldsymbol{D}_k(\boldsymbol{s}, \hat{\boldsymbol{s}}) = \delta(k(\boldsymbol{s}), k(\hat{\boldsymbol{s}}))$ |
| $\rho$ | A correlation function with $\rho : \mathbb{R}^Z \times \mathbb{R}^Z \to \mathbb{R}$ |

## Constants

| | |
|---|---|
| $C$ | The number of classes |
| $D$ | The dimension of the input |
| $W$ | The dimension of the parameter vector |
| $V$ | The dimension of the explanation outputs |
| $Z$ | The number of perturbation steps |
| $K$ | The number of samples to approximate the Jacobian |
| $T$ | The number of integral steps between two points, $\boldsymbol{e}$, and $\hat{\boldsymbol{e}}$ |
| $M$ | The number of models to average over in `GEF`, and `Fast-GEF` |

## Variables

| | |
|---|---|
| $\xi$ | The perturbation magnitude defined as the discrepancy $\delta(\boldsymbol{s}, \hat{\boldsymbol{s}}) = \xi$ between $\hat{\boldsymbol{s}}$, and $\boldsymbol{s}$ |
| $\boldsymbol{D}_f$ | The model distortion $\boldsymbol{D}_f$ across parameter- $\boldsymbol{D}_f(\theta, \hat{\theta})$, and input perturbation $\boldsymbol{D}_f(\boldsymbol{x}, \hat{\boldsymbol{x}})$ |
| $\boldsymbol{D}_\phi$ | The explanation distortion $\boldsymbol{D}_\phi$ across parameter- $\boldsymbol{D}_\phi(\theta, \hat{\theta})$, and input perturbation $\boldsymbol{D}_\phi(\boldsymbol{x}, \hat{\boldsymbol{x}})$ |
| $\varepsilon_{\boldsymbol{D}_k}^{RO}$ | The implicit upper boundary value with $\varepsilon^{RO} \in \mathbb{R}^+$, and $k \in \{\phi, f\}$ used in robustness |
| $\varepsilon_{\boldsymbol{D}_k}^{SE}$ | The implicit lower boundary value with $\varepsilon^{SE} \in \mathbb{R}^+$, and $k \in \{\phi, f\}$ used in sensitivity |
| $\alpha$ | A boundary value for the perturbation magnitude, with $\alpha \in \mathbb{R}^+$ |
| $\eta_i$ | The Gaussian noise matrix with $\eta_i \sim \mathcal{N}(\boldsymbol{0}, \sigma_i^2 \mathbb{1})$ |
| $\sigma_z^2$ | The covariance scale of a Gaussian distribution with $\sigma_z^2 \in \mathbb{R}^+$ at $z^{th}$ perturbation |
| $J_f$ | The network Jacobian for fixed input $\boldsymbol{x}$, and model $f$, with $J_f \in \mathbb{R}^{V \times C}$, and elements $J_{i,j} = \frac{\partial e_i}{\partial f_j}$ |
| $\mathbf{g}$ | Pullback metric tensor based on the elementwise Jacobian with $\mathbf{g} \in \mathbb{R}^{V \times V}$ |
| $z$ | Index of perturbation steps with $z \in [1, Z]$ |
| $\boldsymbol{D}_f^z$ | The model distortion at perturbation step $z$ with $\boldsymbol{D}_f^z := \boldsymbol{D}_f^z(\theta, \hat{\theta}_z)$ |
| $\boldsymbol{D}_\phi^z$ | The explanation distortion at perturbation step $z$ with $\boldsymbol{D}_\phi^z := \boldsymbol{D}_f^z(\theta, \hat{\theta}_z)$ |
| $\boldsymbol{d}_f$ | The vector of model distortion with $Z$ steps, $\boldsymbol{d}_f = [\boldsymbol{D}_f^1, \boldsymbol{D}_f^2, \ldots, \boldsymbol{D}_f^Z]$ |
| $\boldsymbol{d}_\phi$ | The vector of explanation distortion with $Z$ steps, $\boldsymbol{d}_\phi = [\boldsymbol{D}_\phi^1, \boldsymbol{D}_\phi^2, \ldots, \boldsymbol{D}_\phi^Z]$ |

