# OpenReview forum: "Evaluating Interpretable Methods via Geometric Alignment of Functional Distortions"
_TMLR — Accepted by TMLR_

### Review · Reviewer_H3Wy · 2024-11-05

**Summary Of Contributions:**

The paper first systemises existing explainability evaluations, focusing on the three criterion of robustness, sensitivity and faithfulness. It proposes unifying them by viewing them as the correlation between model distortion and explanation distortion under perturbations. It highlights that directly comparing euclidean-distance measurements of model and explanation distortion is not enough, as they belong to different spaces. The paper proposes GEF, which fixes this by using a jacobian-based pullback operator, though computing this pullback is expensive so it also proposes Fast-GEF for practitioners which still uses euclidean distance. Using GEF, Fast-GEF, the paper evaluates local, global and llm-explanation methods across nlp, vision and tabular tasks. The high-level takeaways are that LLM-explanations are near-random, and local explanation methods are more faithful than global ones.

**Audience:**

Yes

**Broader Impact Concerns:**

None.

**Claims And Evidence:**

Yes

**Requested Changes:**

1. Can you plot Figure 3 but using the GEF, Fast-GEF metrics to see whether they lead to more coherent scatter plots between model and explanation distortion.

2. The code url in the abstract is broken. Please fix it.

3. Can more recent interpretability techniques on LLMs such as attention head ablations, causal patching, sparse auto-encoders etc. be added to the evaluations? This would make the methodology immediately interesting to the audience.

**Strengths And Weaknesses:**

**Strengths**
1. The paper is of great educational value in that it systemises existing approaches for evaluating explanations without access to ground-truth explanation labels.
2. It proposes a generalised criteria GEF which combines robustness, faithfulness and sensitivity scores into one metric.
3. It brings up the important fact that model and explanation distortion belong to different spaces and their euclidean distance measurements cannot be directly compared.
4. It tries to question existing assumptions in the field about how distortions should increase monotonically with perturbation amount.

**Weaknesses**
1. GEF is expensive, and it is unclear if Fast-GEF does not suffer from the same issue shown in Figure 3. While emphasis is made on the importance of the differential geometry intuition, it does not seem Fast-GEF uses it, so it is unclear how important this perspective is.

2.  I am not so sure I buy the evidence for how model distortions and perturbation don't align in practice. It's unclear what each point in the scatter plot in Fig A.2 A-C represents, making it difficult to assess whether the claim is justified. Are these the distortions on individual samples? if so, its still possible that for a fixed sample, increased perturbation leads to increased distortion.  Similarly, for layer-by-layer randomisation, it's unclear how many seeds the result is computed across, and whether the deviations are high just because of that. What is being varied in the randomisation? How can we be sure the setup is faithful to existing work? Mostly, I see an increasing trend in distortion as the perturbation level increases, with variance across samples as expected. I am not sure how this is evidence for the monotonicity assumption being violated.

3. The GEF metrics by combining three desiderata for explanations (faithfulness, sensitivity, robustness) leads to lesser insight about which of these three desiderata an explanation method underperforms in which can be valuable to both researchers developing new methods and practitioners using them.

---

> ### Author Response · Authors · 2024-12-02
>
> Thank you for your insightful review of our paper! It is great to hear that you find our work of educational value and share our opinion that it is important to account for geometric differences, and the often neglected fact that model and explanation functions reside in different spaces. In particular, your review has been helpful in extending our experiments to also include SAEs and clarifying the advantages and disadvantages of Fast-GEF vs GEF with several additions to the Appendix and updates to the main manuscript. Below, we address your concerns point by point.
>
> **Responses to Weaknesses**
>
> **(1) Comparison of Fast-GEF and GEF.** It is a correct observation that Fast-GEF does not use the geometric perspective (with the pullback technique) in its quality estimation. However, one of the main issues, which is presented in Fig 3 is that uniform perturbation of inputs (or parameters) often has a non-uniform effect on consequent functions — is precisely what we target with the GEF method and hence can be seen as an extension of Fast-GEF.
>
> **Shared benefits.** Both proposed methods — Fast-GEF and GEF — provide multiple advantages over existing metrics (see Defs. 2-4, with examples in Related works) such as:
>
> - **Task-Agnosticity**: By replacing input perturbations with model parameter scaling, they avoid dataset-specific thresholds, enabling consistent benchmarking across diverse tasks (Section 5.1).
> - **Unified, theoretically sound**: By exploring a spectrum of model conditions ($z=1$ at $y=y$ and $z=Z$ at $y \neq y$), comprehensive quality estimation is ensured (Section 3.3) — assigning high scores to faithful explanation (proof in Appendix A.1.3) and penalises constant or random explanations (proofs in Appendix A.1.1, empirical evidence A.8.3), which might pass traditional robustness/sensitivity tests.
> - **Metric Reliability**: Our meta-evaluation experiments (Section 6.1, Appendix A.6.3) confirm that Fast-GEF and GEF are generally more reliable compared to other metrics and comparable to each other.
>
> **Serves different needs.** Despite their shared benefits, they address different practical needs: GEF incorporates geometric considerations for precise measurement of distortions, while Fast-GEF offers computational efficiency by omitting use of the pullback. This makes them complementary methods with distinct trade-offs.
>
> To clarify **the relationship between GEF and Fast-GEF** — thanks to your suggested change (1) — we added a comparison (Appendix A.6, Fig. A.6) between Fast-GEF and GEF. It shows that while Fast-GEF and GEF often share categorical rankings (see, Appendix A.8.2), they may differ in their individual estimates. This divergence in magnitudes arises because Fast-GEF does not account for manifold-specific distortions. These differences can grow with model complexity (e.g., highly parameterised models for ImageNet tasks). These results, based on your requested analysis, highlight that while Fast-GEF serves as a practical approximation, it is not a substitute for GEF.
>
> **Practical recommendation.** Fast-GEF is computationally efficient, making it ideal for large-scale benchmarks or resource-constrained small ML labs. By contrast, GEF’s pullback mechanism provides precise distortion alignment but demands more compute. Due to GEF’s ability to measure geometric distortions, we recommend it wherever possible. For limited computing budgets and for the benefits (1-3) listed above, Fast-GEF remains a good alternative with consistent rankings to GEF in the examples we have tested.
> This question prompted us to add a focused discussion in Section 5.3, where we explicitly discuss the trade-offs of GEF vs Fast-GEF, with the hope of guiding readers in selecting the appropriate method based on their constraints.

---

> > ### Author Response · Authors · 2024-12-02
> >
> > **(2) Evidence for violation of monotonicity assumption.** Thank you for pointing out the need for additional evidence and clarity regarding the violation of monotonicity assumptions (Ass 1-3). Below, we summarise the enhancements made to address these concerns:
> > - **Additional analysis with violin plots (Fig. A.3 A-C, prev. A.2).** In Fig A.3, the goal was to provide varied statistical evidence for $N=50$, which makes details harder to convey. Each point in each violin in Fig. A.3 A-C represents a sample, overall showing the distribution across $N=50$ random samples. While the violin plots suggest the common presence of outliers, we agree that a sample-wise plot might be more helpful for our purpose. To complement these plots, we’ve now added line plots in Fig. A.5 (and revised Appendix A.5) to visualise the distortion trajectories of individual samples. These trajectories reveal both correlated and uncorrelated patterns between perturbation levels and model distortions. The new plots provide clearer evidence that model distortions do not always increase monotonically with perturbation levels, even when aggregated trends appear consistent.
> > - **Clarifications on line plots (Fig. A.2. D-E, prev. A.2).** Regarding your question about seeds and variability in layer-by-layer randomisation experiments — all experiments use a fixed random seed. This ensures that model parameters are the same across settings, with the standard deviation computed across samples. We follow the Model Parameter Randomisation Test (MPRT) [1] randomisation procedure, which randomises the parameters of individual layers — either in a top-down or bottom-up order. Although the general trend is monotonic — the variability with randomisation highlights that explanations are sensitive to choices like the order of perturbation (corroborating previous work [2, 3]), not just the perturbation.
> > - **Consistency with existing work.** For all our experiments, we rely on the standardised metric implementations provided in the Quantus [4] library. This ensures that our setup is consistent with existing work.

---

> > > ### Author Response · Authors · 2024-12-02
> > >
> > > **(3) Unified versus single evaluations.** You mention that unifying multiple metrics into GEF “leads to lesser insight” into specific underperformance areas — this is a great point, and we appreciate the opportunity to clarify.
> > >
> > > As mentioned frequently in recent findings [5,6,7,8] and as we report in our work, Defs. 2-4 have many drawbacks. To mention a few: (i) dependency on thresholds (Section 2.2.3), (ii) sample-wise violations of model monotonicity (Appendix A.5), (iii) overreliance on input perturbation, confounding evaluation and hindering cross-dataset and architecture comparison (Appendix A.5), and (iv) theoretical drawbacks of assigning perfect scores to intrinsically poor, random explanations (Appendix A.1). Thus, while we generally agree that more metrics and a holistic approach are preferred over a single-metric evaluation, it is also critical to note that these evaluations might not stand reliably on their own.
> > >
> > > GEF and Fast-GEF target each of these issues, offering a (i) threshold-free, (ii) model-anchored, (iii) general-purpose (due to its choice of perturbation strategy), and (iv) unified, theoretically sound evaluation. We think GEF and Fast-GEF are not substitutes for existing evaluations but provide a more reliable estimate of explanation quality overall.
> > >
> > > From a practical standpoint, researchers developing new explanation methods will likely optimise against a single target. Here, GEF becomes the preferred choice. Additionally, interdisciplinary researchers with fewer resources who spend time on evaluations can feel overwhelmed by the number of different evaluation methods and the correct choice of hyperparameters. This confusion leads to misuse of these evaluations and unreliable scores [9]. Again, here GEF becomes a preferable choice.

---

> > > > ### Author Response · Authors · 2024-12-02
> > > >
> > > > **Requested changes.**
> > > >
> > > > - (1) Thank you for suggesting this extension. We have added Fig. A.6 with a discussion to Appendix A.6, which extends Fig. 3 to include Fast-GEF and GEF. The results demonstrate that coherence between model and explanation distortions tends to improve with Fast-GEF and even more so with GEF, particularly for highly parameterised models (see $\rho(s, \hat{s})$ in Fig. A.6). However, coherence depends on the specific combination of explanation and model functions, as well as inherent nonlinearities and manifold structures. Without ground truth for explanation quality, higher coherence is not guaranteed and should not be directly interpreted as an indicator of better evaluation.
> > > > - (2) Thank you, concerning the broken URL — to preserve anonymity during the review process, we haven’t made it “clickable” so far. The link to the GitHub repository will be available as soon as the paper moves toward publication (i.e., in the camera-ready version).
> > > > - (3) This is a great suggestion! We’ve added a new Experiments subsection (see Section 6.4) to the main manuscript. In the experiment, we evaluated the faithfulness of SAEs on a Gemma-2-2B text classification task. Given the short rebuttal time, we ran it for Fast-GEF, but are nonetheless excited to report the results — e.g., that SAEs are generally faithful (on average, ~0.75 in Fast-GEF score) but when comparing widths of the SAEs (16K vs 65K), it does not seem that wider representations, i.e., explanations, result in higher faithfulness. In our results, there seems to be no trade-off. We hope that you find these results interesting and that the general audience of TMLR does as well. We’re grateful for this helpful suggestion!
> > > >
> > > > **References.**
> > > >
> > > > [1] [Adebayo et al., 2024](https://arxiv.org/pdf/1810.03292)
> > > >
> > > > [2] [Binder et al., 2022](https://arxiv.org/pdf/2211.12486)
> > > >
> > > > [3] [Hedström et al., 2024](https://link.springer.com/chapter/10.1007/978-3-031-63787-2_21)
> > > >
> > > > [4] [Quantus](https://pypi.org/project/quantus/)
> > > >
> > > > [5] [Zhang et al., 2024](https://openreview.net/pdf?id=Hftgajppmz)
> > > >
> > > > [6] [Brocki et al., 2024](https://openreview.net/pdf?id=nbqO93YTz-)
> > > >
> > > > [7] [Hase et al., 2024](https://openreview.net/pdf?id=HCrp4pdk2i)
> > > >
> > > > [8] [Agarwal et al., 2022](https://openreview.net/pdf?id=BfxZAuWOg9)
> > > >
> > > > [9] [Krishna et al., 2024](https://arxiv.org/pdf/2202.01602)

---

> > > > > ### Comment · Reviewer_H3Wy · 2024-12-12
> > > > >
> > > > > Thanks for the additional experiments, this clarifies my main concerns. I think this paper is quite interesting and recommend acceptance.

---

### Review · Reviewer_4myU · 2024-11-13

**Summary Of Contributions:**

The paper proposes a new evaluation criterion for explainable AI: "a threshold-free metric based on principles from differential geometry". The main argument is that XAI relies too much on task-specific evaluations. In the absence of a general theory, and considering the costs of subjective evaluations, this line of research is highly relevant.

Without ground-truth explanation labels, achieving faithfulness is arguably an impossible task. Early work in the area of neurosymbolic AI has instead focused on achieving fidelity of explanations to the trained model, rather than to the real-world problem being modelled. It could be argued that this is in practice how far one can go.

Now, if explanations can guide alignment and this process can be shown to take the model closer to solving the real-world problem, essentially a proxy to the ground truth explanation labels, then this brings XAI real close to neurosymbolic AI because it requires not only explanations but an ability to intervene in the model (re-training) based on the explanations and to show that the intervention is beneficial. This as far as I am aware was first proposed in https://neurosymbolic-ai-journal.com/system/files/nai-paper-729.pdf

**Audience:**

Yes

**Broader Impact Concerns:**

+ The way that the work is presented can help guide the field towards the key methodological questions that are discussed in the paper rather than to more and more application-specific approaches and algorithms that at the end are not very relevant.

- One limitation of the proposed approach is that, despite the claim of generality, it seems to work only for perturbation-based local approaches such as LIME and image-based aggregations approaches such as GradCAM. There are well-documented problems with both classes of XAI when it comes to fidelity, see e.g. https://arxiv.org/abs/1908.03020. There are other layerwise global explanation approaches that seek to extract high-fidelity logic statements from trained neural networks.

**Claims And Evidence:**

Yes

**Requested Changes:**

The claims initially confined to the provision of a better XAI metric supported by a geometric argument, extend without much justification to: unifying criterion for robustness, sensitivity, and faithfulness and fair measurements across non-linear mappings. Some of the claims could be toned down. As far as I can tell, robustness and fairness are not really evaluated empirically in the paper.

Fig.2 should be explained in a lot more detail. This is crucial to justify the idea that "diverse evaluation methods can be unified under a shared conceptual framework", and to convince the reader that having such a shared framework is desirable.

It wasn't clear to me what we're expected to conclude from Fig.3. I get it that different geometries will affect XAI results, but just like we don't know the ground truth, the geometry isn't known, or is it? In particular, I'd like to know better what defines an "explanation function" and how distortions are calculated in practice.

Also, if I understood correctly, Fig2(B) is not intended to split the space equally as shown with the yellow, green and red squares, since the goal is to avoid arbitrary thresholds. The figure could have reflected this better.

**Strengths And Weaknesses:**

+ The paper is clearly written and presentation is good.

- The figures could have been better and the captions could have been more informative.

+ The experiments are fine and the results with LLMs are also interesting, although not surprising: why should an LLM provide any better an explanation than another black-box model?

+ Overall, this is a very relevant area of research.

---

> ### Author Response · Authors · 2024-12-02
>
> We thank the reviewer for the time spent on the review! The feedback was especially insightful and helped us clarify and extend our contribution regarding the unification in Section 3 and the geometric view in Section 4. We’re glad to hear that you found our work is highly relevant, clearly written and presented, and helpful in guiding the field towards the key methodological questions. In the following, we address all concerns point by point.
>
> **Model intervention and neurosymbolic AI.** We thank the reviewer for pointing out the important connection to the field of neurosymbolic AI and agree regarding the importance of showing that model intervention (based on explanations) is beneficial. We really think that the evaluative approach of downstream task evaluation [1, 2] is critical and complementary to ours, and these are works that are now properly referenced in our paper (see Section 7).
>
> **Addressing weaknesses and general comments.** Regarding the concerns in Section 3: (i) lowered claims and (ii) more details on Fig. 2 — we’ve revised the entire section with reformulations for content. For Fig. 2, we’ve added a complementary illustration to our theoretical definition of GEF, which we hope will facilitate the reader’s understanding. We’ve reviewed each claim made in the paper, clarifying, adjusting, and making them more precise. We edited limited captions and figures throughout the main body and appendix material.
>
> Then, we thank you for letting us know that it was unclear **what to conclude from Fig. 3**. We agree that it could be improved. In the revised version of the manuscript, we’ve significantly updated Fig. 3 (simplified it and applied a consistent colour scheme) as well as extended it with complementary analysis in Appendix A.6, and included references to clarify which explanation functions are within the scope of our analysis (both in Sections 3.3 and 4).
>
> Also, on the comment that **LLM results are not surprising** — we agree in totality. Since previous studies [3] have suggested otherwise — i.e., that LLMs might be faithful as post- hoc explainers — we felt the need to contribute additional evidence to this area for the XAI and related communities. As reported in our paper (Section 6.3), we find that the faithfulness of LLMs as post- hoc explainers in text tasks are limited.
>
> Lastly, we too share the concern regarding the evidence on failure modes when it comes to the mentioned explanation methods, e.g., [4, 5] etc. It is helpful for us to note that we’ve missed **conveying the scope of GEF’s applicability**. It goes beyond just the perturbation-based local approaches such as LIME and image-based aggregation approaches like GradCAM, as you mentioned. GEF is general to the extent that it assumes an explanation function as defined in Section 2.1, i.e., global and local methods. We’ve made sure to add a clarifying sentence in Section 3.3 and reference to the Preliminaries (Section 2.1) to clarify the scope of GEF, which in actuality is much broader than what is in our experiments.
>
> **References.**
>
> [1] [Krishna et al., 2024](https://arxiv.org/pdf/2305.11426)
>
> [2] [Bareeva et al., 2024](https://openreview.net/pdf?id=IFk4bOA11Z)
>
> [3] [Kroeger et al., 2023](https://openreview.net/pdf?id=1G7n7LW3mF)
>
> [4] [Adebayo et al., 2018](https://arxiv.org/pdf/1810.03292)
>
> [5] [Zhang et al., 2024](https://openreview.net/pdf?id=Hftgajppmz)

---

### Review · Reviewer_FKvG · 2024-11-20

**Summary Of Contributions:**

The paper addresses the problem of explanation faithfulness. In particular, the absence of a general framework that unifies perturbation-based evaluative criterions. The paper first develops a unifying theory and then, within the theory, proposes a Generalized Explanation Faithfulness criterion that is both threshold-free and task agnostic. The proposed quality estimator is based on the correlation of distortion vectors, as collections of distortions for perturbations of increasing magnitude, to account for both robustness and sensitivity. The paper further develops GEF to account for geometric characteristics of the different functions spaces in which the model output and the explanation output functions live. They equip the model output space with use the pull-back metric tensor, rather than the Euclidean metric,  and compute distances accordingly. Finally, the paper provides extensive experiments in support of the claim that GEF, and Fast-GEF provide a sound faithfulness evaluator and test various explanation approaches such as the recently proposed LLM-as-explainer methods. Overall, the unifying theory seems coherent and helpful and the use of Riemannian geometry to account for different geometric properties of the functions spaces is ingenious. The proposed method based on correlation of distortion vectors can be helpful for the XAI community, even though, this not being my main field of expertise I cannot evaluate its novelty.

**Audience:**

Yes

**Broader Impact Concerns:**

The manuscript already has a broader impact statement that survives well the purpose.

**Claims And Evidence:**

Yes

**Requested Changes:**

As pointed out in the Strengths And Weaknesses section, the proposed approach to the computation of Jacobians can potentially yield very different results to the actual (perhaps intractable) Jacobian. Maybe investigating how numerical approximations are consistent with the true Jacobian in (big, and highly overparameterized) differentiable models, where it can be computed, would be interesting. Second, I am worried that the noise schedule used and the small number of points can fail to approximate well a smooth path. It would be interesting to see if this is the case, perhaps ablating over the number of steps and noise magnitudes.

**Strengths And Weaknesses:**

Strengths:
-The integration of robustness, sensitivity, and faithfulness under a single evaluative framework addresses fragmentation in the XAI literature.
-The paper provides extensive experiments that showcase the utility of GEF and Fast-GEF as faithfulness evaluators. By testing these evaluators across diverse datasets (vision, tabular, NLP), models, and explanation methods (including LLM-as-explainer methods). Correlation of distortion vectors is simple and flexible and makes GEF potentially useful.
-The recognition of the geometric differences between function spaces and the use of Riemannian geometry to account for these differences is innovative.


Weaknesses, such as computational complexity due to GEF's reliance on path integrals using the pull-back metric tensor, or the underlying assumptions that perturbations of model parameters effectively simulate realistic changes in model behavior, are partially addressed in the paper. One question to ask is: how much the faithfulness evaluation depends on the perturbation path, and therefore on the particular noise schedule used to construct the distortion vector? This seems like a key choice to make beforehand. Second question is: depending on the nature of the function, e.g., implicit or non-differentiable functions, it can be impossible to compute the Jacobian (and consequently g = J^TJ). To what extend can GEF be applied to popular explanation methods that have been proved useful in the XAI literature? If I understand correctly you approximate the Jacobian numerically as infinitesimal differences between explanations. I suspect this approach to potentially yield very different results to the actual (perhaps intractable) Jacobian. Perhaps investigating how numerical approximations are consistent with the true Jacobian in (big, and highly overparameterized) differentiable models would be interesting. Last, in A.2.1 you outline how the integral (eq. 11) is computed. You say you use only 5 points, I suspect because of the computational complexity. I am worried that the noise schedule used and the small number of points fail to approximate well a smooth path.

Overall, I find that the strengths outweigh the weaknesses.

---

> ### Author Response · Authors · 2024-12-02
>
> Thank you for your insightful and very detailed review. We greatly appreciate your positive feedback on the coherence of our unifying theory, our use of Riemannian geometry, and the practicality of correlating distortion vectors. Your observations regarding computational complexity (as discussed in Sections 5.3 and 7.1) and questions regarding the Jacobian approximation, and the noise schedule's impact on smooth path approximation have indeed been helpful in improving our paper. We address the individual points below.
>
> **Smoothness of path and ablation study.** We agree that your first question of whether the faithfulness evaluation depends on the perturbation path, is an important one. We also think that the type of perturbation is a key choice(*) in the evaluation. To complement the existing considerations on the influence of $Z$ in Appendix A.1.4 — as suggested in your requested changes, we’ve added an ablation study in Appendix A.7 which justify our hyperparameter choices. Empirically, in tested experiments, we found that setting $M, Z, T, K = 5$ strikes the balance between “stability and computational cost” and therefore this choice was propagated throughout our benchmarking experiments. For the agreement runs, we set $T, K=10$. Recall that, $M$ is the number of models, $Z$ is the number of perturbation steps, $T$ is the intermediate points in the integral, and $K$ is the number of samples at each integral step.
>
> That said, while we found empirical evidence that employing $M, Z, T, K = 5$ is sufficient to provide stable results in our experimental settings, it goes without saying that such hyperparameter findings cannot be claimed for universality (across tasks, NN models, and datasets). For this reason, we are very interested in improving our solution, e.g., exploring further performance enhancements of our sampling-based approximation or designing adaptive noise schedules. We added these proposals to our discussion under future work (Section 7.1).
>
> **Model scope.** With respect to the second question of the nature of the model function, your observation is absolutely correct: for implicit or non-differentiable functions, the Jacobian cannot be computed. In our preliminaries (Section 2.1), we write out the model scope — that model function $f$ is a “differentiable neural network (NN)”. This defines the scope for GEF. That said, since Fast-GEF does not rely on the pullback mechanism, differentiability is not a necessary criterion (unless the explanation function relies on it). As such, Fast-GEF evaluation could be a useful alternative under such conditions.
>
> **Explanation scope.** Next, you asked regarding the scope in which GEF can be applied to popular explanation methods. We are interested and would like to understand your question better. Which methods are you referring to? In the meantime, we can point you to the preliminaries (Section 2.1), where we provide examples of explanation methods within the scope of GEF. Also, experimentally, we include many method variants such as local (gradient-, backpropagation-, Shapley-based), global, and LLMs-as-explainer methods. In the revision, we also have experiments that evaluates sparse autoencoders (SAEs) as explanations.
>
> **Approximating the true Jacobian.** Lastly, you commented on our approach of approximating the Jacobian numerically as infinitesimal differences between explanations, with the hypothesis that our sampling-based method might yield different results compared to the actual (and perhaps intractable) Jacobian. We acknowledge this as a valid concern. As you rightly point out, in the general case, numerical approximation is often the only viable approach within the current computational frameworks and resources available to small ML labs (see our hardware details in Section A.4.3). Nevertheless, our methods are designed to be scalable in principle, and we share your interest in understanding how consistent these approximations are with the true Jacobian in large, highly over-parameterised differentiable models. Exploring this could yield valuable insights, but calculating the true Jacobian of large models is beyond our computational means. If you have suggestions for feasible techniques or models that might make such an investigation feasible, we would greatly appreciate your guidance. Your insights could help shape our future work for GEF.
>
> (*) While the choice of perturbation and its parameters is important to explore explanation behaviour across different model responses, by anchoring the distortion quantities, the issues are mostly alleviated — since we measure how well the explanation tracks the model's post-distortion behaviour, in a relative sense. Theoretically, any perturbation could be used (including input perturbations), but model-level perturbations ensure the evaluation remains task-agnostic, avoiding adjustments required for specific data types like images or text.

---

> > ### Comment · Reviewer_FKvG · 2024-12-06
> >
> > Thank you for the clarifications. With my question, perhaps poorly phrased, I was indeed asking to what extent the pull-back mechanism could be applied to XAI methods, without having to fall back on fast-GEF, as I am not very familiar with the XAI literature. I can say that you clarified my doubts and, therefore, I will recommend to accept the paper.

---

### Author Response · Authors · 2024-12-04

Hi,

First, we would like to thank all reviewers for their insightful and constructive comments. The feedback, requested changes and mentioned weaknesses, really gave us an opportunity to improve our work. We’ve now uploaded a new version of our manuscript with all the changes incorporated.

We’re happy to hear that all three reviewers agreed that our submission meets TMLR's critical acceptance criteria: (1) our claims are supported by "accurate, convincing, and clear evidence," and (2) the findings are of interest to TMLR’s audience. It is encouraging to hear that our usage of a pullback was seen as “innovative”, our work "highly relevant" and of "great educational value".

Given the mentioned weaknesses and requested changes, we made the following key changes across the manuscript:

- **(Section 1-3, Abstract, Introduction, Background).** Mentioning of new experiments. Fixed minor typos. Added relevant citations. Updates Figure 1.
- **(Section 3, A Unifying Perspective).** Revised the text, with a particular focus on refining claims, figure captions and remarks surrounding Definition 5. Updated Table 1 to include our proposal of a unified theory. Added another panel (C) to Figure 2 to illustrate Definition 5.
- **(Section 4, A Geometric Perspective).** Revised Figure 3 and its surrounding discussions — simplified the plot (moved some subfigures to Appendix), streamlined colour schemes and updated the caption. Revised caption of Figure 4 and surrounding text.
- **(Section 5, Method: From Theory to Practice).** Improved text in 5.1, revised Figure 5. Added subsection 5.3 to guide readers in choosing between GEF and Fast-GEF methods.
- **(Section 6, Experiments).** Added a new research question and experiment where we evaluate the faithfulness of sparse autoencoders (SAEs) across different model layers and SAE widths. New results with Figure 10 (A-G) and associated discussions are found in subsection 6.4.
- **(Appendix, Method: From Theory to Practice).** Added a paragraph to A.4.4 to explain how SAE explanations are generated. Added a discussion and new plot A.5 on sample-wise changes to model distortion across perturbation levels. Added a new subsection A.6 with an extended study of alignment patterns with Figure A.6 (to complement Figure 3 in the main manuscript) and discuss differences in individual estimations between GEF and Fast-GEF with Figure A.7, to complement existing analysis in A.8.2. Added a new ablation study of GEF hyperparameters in subsection A.7 with Figure A.8 motivating the empirical choices of Z, M, T, K = 5 and comment on the Jacobian approximation. General formatting and fixes of minor typos.

We hope the revised manuscript meets the expectations of the reviewers and are happy to address any further questions.

/Authors

---

### Decision · Action_Editor_e4Dp · 2024-12-31

**Recommendation:** Accept as is

**Comment:**

This paper focuses on the problem of measuring explanation faithfulness, i.e. whether an interpretability method's explanation for a model's prediction is actually reliable. The paper makes the case that existing evaluation procedures are scattered and insufficient, and therefore proposes a more principled and comprehensive evaluation procedure. A large-ish scale evaluation (across many modalities/settings) of existing explanation faithfulness methods is undertaken using the proposed evaluation, and interesting insights are shared. After some back-and-forth to improve the paper, all reviewers agreed the paper should be accepted.

**Audience:**

Certainly interpretability broadly and explanation faithfulness specifically are of interest to the TMLR community.

**Claims And Evidence:**

All reviewers felt this was a comprehensive survey with a well-motivated and justifiable evaluation procedure. The results were presented in a clear and convincing way.